# Asymptotic normality and confidence intervals for derivatives of 2-layers neural network in the random features model

**Yiwei Shen,**
Department of Statistics,
Rutgers University,
Piscataway, NJ 08854, USA.
ys573@stat.rutgers.edu

**Pierre C Bellec,**
Department of Statistics,
Rutgers University,
Piscataway, NJ 08854, USA.
pierre.bellec@rutgers.edu

## Abstract

This paper studies two-layers Neural Networks (NN), where the first layer contains random weights, and the second layer is trained using Ridge regularization. This model has been the focus of numerous recent works, showing that despite its simplicity, it captures some of the empirically observed behaviors of NN in the over-parametrized regime, such as the double-descent curve where the generalization error decreases as the number of weights increases to $+\infty$.

This paper establishes asymptotic distribution results for this 2-layers NN model in the regime where the ratios $\frac{p}{n}$ and $\frac{d}{n}$ have finite limits, where $n$ is the sample size, $p$ the ambient dimension and $d$ is the width of the first layer. We show that a weighted average of the derivatives of the trained NN at the observed data is asymptotically normal, in a setting with Lipschitz activation functions in a linear regression response with Gaussian features under possibly non-linear perturbations. We then leverage this asymptotic normality result to construct confidence intervals (CIs) for single components of the unknown regression vector.

The novelty of our results are threefold: (1) Despite the nonlinearity induced by the activation function, we characterize the asymptotic distribution of a weighted average of the gradients of the network after training; (2) It provides the first frequentist uncertainty quantification guarantees, in the form of valid $(1-\alpha)$-CIs, based on NN estimates; (3) It shows that the double-descent phenomenon occurs in terms of the length of the CIs, with the length increasing and then decreasing as $\frac{d}{n} \nearrow +\infty$ for certain fixed values of $\frac{p}{n}$. We also provide a toolbox to predict the length of CIs numerically, which lets us compare activation functions and other parameters in terms of CI length.

## 1 Model, contributions and related works

**Random features model as a 2-layers neural network.** Given $n$ observations $(\boldsymbol{x}_1, y_1), ...(\boldsymbol{x}_n, y_n)$ with $\boldsymbol{x}_i \in \mathbb{R}^p$ and $y_i \in \mathbb{R}$ for each $i = 1, ..., n$, the object of study of this paper is the estimate

$$\widehat{\boldsymbol{\alpha}} = \operatorname*{argmin}_{\boldsymbol{\alpha} \in \mathbb{R}^d} \Big( \frac{1}{2n} \sum_{i=1}^{n} \Big( y_i - \sum_{k=1}^{d} \alpha_k \sigma(\boldsymbol{x}_i^\top \boldsymbol{w}_k) \Big)^2 + \frac{\tau}{2} \|\boldsymbol{\alpha}\|_2^2 \Big), \tag{1}$$

where $\sigma : \mathbb{R} \to \mathbb{R}$ is a nonlinear activation function, $\boldsymbol{w}_1, ..., \boldsymbol{w}_d \in \mathbb{R}^p$ are vectors of random weights sampled by the practitioner. This estimate (1) corresponds to a two-layers Neural Network (NN), with random first layer weights $\boldsymbol{w}_1, ..., \boldsymbol{w}_d$, and second layer weights $\widehat{\boldsymbol{\alpha}} \in \mathbb{R}^d$ trained using a Ridge regularization penalty. The penalty parameter $\tau$ is chosen by $\tau = (d/p)\lambda$, where $\lambda > 0$

is a fixed parameter, so that the random features model is related to the kernel Ridge regression. Throughout, we define the matrix $\boldsymbol{A} \in \mathbb{R}^{n \times d}$ entrywise by $A_{ik} = \sigma(\boldsymbol{x}_i^\top \boldsymbol{w}_k)$. Equivalently, denoting by $\sigma : \mathbb{R}^{n \times d} \to \mathbb{R}^{n \times d}$ the entrywise application of $\sigma(\cdot)$ with a minor abuse of notation, if $\boldsymbol{W} \in \mathbb{R}^{d \times p}$ has rows $\boldsymbol{w}_1^\top, ..., \boldsymbol{w}_d^\top$, we have in matrix notation

$$\boldsymbol{W}^\top = [\boldsymbol{w}_1|...|\boldsymbol{w}_d], \quad \boldsymbol{A} = \sigma(\boldsymbol{X}\boldsymbol{W}^\top), \quad \widehat{\boldsymbol{\alpha}} = \operatorname*{argmin}_{\boldsymbol{\alpha} \in \mathbb{R}^d} \left( \frac{1}{2n} \|\boldsymbol{y} - \boldsymbol{A}\boldsymbol{\alpha}\|_2^2 + \frac{\tau}{2} \|\boldsymbol{\alpha}\|_2^2 \right). \quad (2)$$

Alternatively to the description of $\widehat{\boldsymbol{\alpha}}$ as the second layer weights of a two-layers NN, the above construction is the random features model of [Rahimi and Recht, 2008] where for each observation $(\boldsymbol{x}_i, y_i)$, a new feature vector $(\sigma(\boldsymbol{x}_i^\top \boldsymbol{w}_k))_{k=1,...,d}$ of size $d$ is constructed using the random weights $\boldsymbol{W}$ sampled independently of the data. Then (1) attempts to regress $\boldsymbol{y}$ on the new feature matrix $\boldsymbol{A} = \sigma(\boldsymbol{X}\boldsymbol{W}^\top)$. After training, a mapping $\hat{f} : \mathbb{R}^p \to \mathbb{R}$ and its gradient are available, given by

$$\hat{f}(\boldsymbol{\xi}) = \sigma(\boldsymbol{\xi}^\top \boldsymbol{W}^\top)\widehat{\boldsymbol{\alpha}} = \sum_{k=1}^d \sigma(\boldsymbol{\xi}^\top \boldsymbol{w}_k)\hat{\alpha}_k, \qquad (\partial/\partial \xi_j)\hat{f}(\boldsymbol{\xi}) = \sigma'(\boldsymbol{\xi}^\top \boldsymbol{W}^\top) \operatorname{diag}(\widehat{\boldsymbol{\alpha}})\boldsymbol{W}\boldsymbol{e}_j, \quad (3)$$

where $\boldsymbol{e}_j \in \mathbb{R}^p, j \in [p]$ is a canonical basis vector. The notation $\boldsymbol{\xi} \in \mathbb{R}^p$ used above for the argument of $\hat{f}$ is used to avoid confusion with the observed feature vectors $\boldsymbol{x}_1, ..., \boldsymbol{x}_n$ that are used in the construction of $\hat{f}$. The prediction function (3) can be used, for instance to provide predictions on a test set $\{\boldsymbol{x}_1^{\text{test}}, ..., \boldsymbol{x}_{n_{\text{test}}}^{\text{test}}\}$ of size $n_{\text{test}}$.

**Linear data generating process with non-linear perturbation.** We consider two models for the data-generating process of $(\boldsymbol{x}_i, y_i), i = 1, ..., n$. We will consider the linear model

$$\boldsymbol{y} = \boldsymbol{X}\boldsymbol{\beta} + \boldsymbol{\varepsilon}, \qquad (4)$$

where $\boldsymbol{\beta} \in \mathbb{R}^p$ is the unknown regression vector of interest, $\boldsymbol{y} \in \mathbb{R}^n$ is the observed response with components $y_1, ..., y_n$, $\boldsymbol{X} \in \mathbb{R}^{n \times p}$ is the feature or design matrix with rows $\boldsymbol{x}_1^\top, ..., \boldsymbol{x}_n^\top$ and $\boldsymbol{\varepsilon} \in \mathbb{R}^n$ is the additive noise. We will also consider a model with a non-linear perturbation $G$:

$$y_i = f(\boldsymbol{x}_i) + \varepsilon_i, \quad f(\boldsymbol{x}_i) = \beta_0 + \boldsymbol{x}_i^\top \boldsymbol{\beta} + G(\boldsymbol{x}_i), \qquad (5)$$

where $\boldsymbol{\beta} \in \mathbb{R}^p$ is again the parameter of interest, $\beta_0 \in \mathbb{R}$ is the intercept and $G : \mathbb{R}^p \to \mathbb{R}$ is a random non-linear perturbation function independent of $(\boldsymbol{x}_1, \epsilon_1, ..., \boldsymbol{x}_n, \epsilon_n)$ with mean zero (i.e., $\mathbb{E}_G[G(\boldsymbol{x})] = 0$ for all fixed $\boldsymbol{x} \in \mathbb{R}^p$). The linear model (4) is obtained as a special case when there is no intercept or perturbation in (5). We assume Gaussian features and Gaussian noises, with $x_{ij} \sim$ iid $N(0, 1)$ and $\varepsilon_j \sim$ iid $N(0, \theta_{\boldsymbol{\varepsilon}}^2)$ mutually independent. The signal-to-noise ratio (SNR) $\rho$ will be specified in § 2.4.

**Asymptotic setting in proportional limits.** We consider a high-dimensional asymptotic setting where $n, p, d \to +\infty$ so that any ratio of two integers among $n, p, d$ has a finite limit, namely

$$\psi_{p,n} := p/n, \quad \psi_{d,n} := d/n, \quad \lim_{n \to +\infty} \psi_{p,n} = \psi_p, \quad \lim_{n \to +\infty} \psi_{d,n} = \psi_d, \qquad (6)$$

where $\psi_p, \psi_d$ are positive finite. As will be made precise in the assumptions below, the mean and variance of each $y_i, i \in [n]$ are bounded uniformly in $n, p, d$ satisfying (6). The function $f(\cdot)$ in (5) is weakly differentiable in features $\boldsymbol{x}_i$, with gradients squared integrable, again uniformly in $n, p, d$.

## 1.1 Contributions

**Asymptotic normality and Confidence Intervals (CIs).** We establish Central Limit Theorems (CLT) for the derivatives of 2-layers NN models in (2) when $n, p, d \to +\infty$ in the proportional asymptotic regime (6). A weighted average of the gradients of the trained NN, up to an explicit additive correction, is proved to be asymptotically normal, where the variance of the limit can be estimated explicitly. The asymptotic normality result holds for pure linear models (4) and non-linear perturbation models (5) for a large class of random perturbations. Based on this asymptotic normality result, we construct CIs for single components $\beta_j$ of the unknown regression coefficients $\boldsymbol{\beta}$ of interest, as well as for linear contrast $\boldsymbol{u}_0^\top \boldsymbol{\beta}$ where $\boldsymbol{u}_0$ is a given direction satisfying $\|\boldsymbol{u}_0\|_2 = 1$. Motivated by recent evidence that the linear coefficients are the only parameters that RF models can learn in this proportional asymptotic limit [Ghorbani et al., 2019], our CIs are established only for the linear part $\boldsymbol{\beta}$. To our knowledge, this is the first method of frequentist uncertainty quantification, in the form of $1 - \alpha$ CIs, based on NN estimates (1).

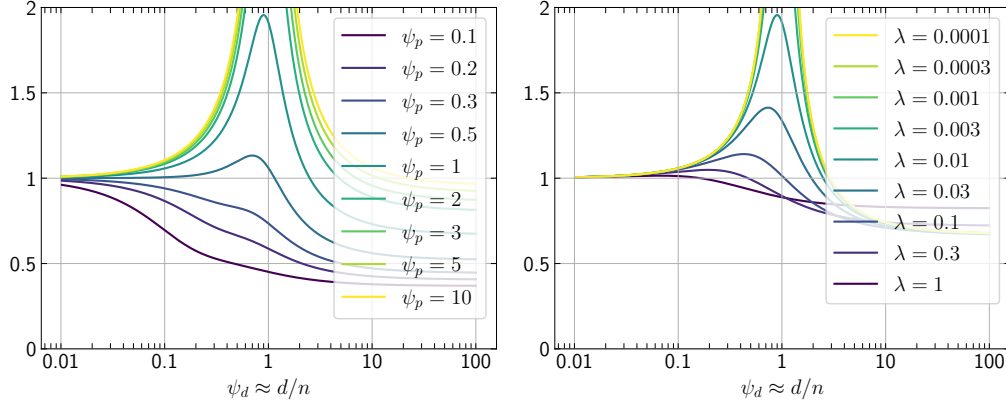

Figure 1: Theoretical squared length $V(\psi_d, \psi_p, \lambda, \rho, \sigma)$ (see (18) on p. 7) of our CI under various $\psi_d$, $\psi_p$ (left) and $\lambda$ (right). In the left the raw Ridge penalty is $\lambda = 0.01$. In the right the dimension ratio is $\psi_p = 1$. In both, we set SNR $\rho = 2$ and activation function $\sigma(x) = \max(x, 0)$.

**Double descent for CI length.** Mei and Montanari [2019] provided the precise asymptotics of the test error and the training error for the random features model and the NN estimate (1) in a setting comparable to ours, providing strong evidence of the double descent phenomenon for the generalization error. Other works that highlight the double descent phenomenon for the generalization error are discussed in § 1.2. Using some results from Mei and Montanari [2019], we are able to characterize the length of our CIs by solving certain quartic equations. This reveals a double descent phenomenon in the length of CIs: increasing model capacity beyond in the overparametrized regime decreases CI length (see Figure 2). Surprisingly, we report that the double descent phenomenon for the length of CIs based on the NN (1) matches the double descent curves from Mei and Montanari [2019] for the generalization error (see Figure 1): (i) It occurs in the interpolation regime when $\lambda$ is small enough and is significant when $p/n \approx \psi_p$ is bounded away from 0; (ii) The CI length is minimized in the infinite width limit (i.e., $\psi_d \to +\infty$) in many cases. This suggests that smaller generalization error leads to smaller CIs and that wider, overparametrized NN leads to smaller CIs.

**Comparing activation functions and other parameters in terms of confidence interval length.** In Figure 3 on p. 8, we report the behaviors of

$$R = R(\psi_p, \lambda, \rho, \sigma) = \lim_{\psi_d \to +\infty} \left[ \lim_{n,p,d \to +\infty \text{ as in (6)}} \left( \frac{nL^2}{\text{Var}(y_1)} \right) \right], \quad L^2 = \frac{\|\boldsymbol{y} - \boldsymbol{A}\widehat{\boldsymbol{\alpha}}\|_2^2}{(\text{trace}(\boldsymbol{I}_n - \boldsymbol{H}))^2}, \quad (7)$$

where $\rho$ is the signal-to-noise ratio (SNR), $\sigma$ is the activation function, and $L^2$ is proportional to the squared length of our $1 - \alpha$ CIs (See § 2.1 and 2.4). § 2.4 explains how to predict numerically the length $L$ for given values of $(\psi_d, \psi_p, \lambda, \rho, \sigma)$, and the length in the infinite width limit ($\psi_d \to \infty$) for given $(\psi_p, \lambda, \rho, \sigma)$. This lets us compare efficiently combinations of $(\psi_d, \psi_p, \lambda, \rho, \sigma)$ in terms of CI length, cf. § 3 where some comparisons are discussed. For instance, in the setting of Figure 3 we see that ReLU activations provide the smallest CI length in the infinite width limit.

## 1.2 Related literature

**Overparametrization and interpolation.** A surprising observation about deep neural networks is the combination of overparametrization (i.e., more weights than observation) and interpolation (zero or small training error), while performing well in terms of generalization error. Experimental results in [Zhang et al., 2016] and Belkin et al. [2018b] were the starting points of this line of research on interpolating learning methods. The good generalization properties of interpolating methods were the focus of many empirical and theoretical works in recent years, including Belkin et al. [2018a,b,c], Liang and Rakhlin [2018], Rakhlin and Zhai [2018], Belkin et al. [2019], Geiger et al. [2020], Muthukumar et al. [2020] among others.

**Double descent phenomenon.** Classically, models that exactly fit the training data are known to "over-fit" as exemplified by the bias-variance trade-off that features a U-shape curve. To reconcile the

performance of modern interpolating learning methods, Belkin et al. [2018a] describes a now well studied "double descent curve": as the number of parameters grow, the out-of-sample prediction error of certain machine learning methods would first increase up until a phase transition after which the generalization error would decrease. The "double descent" phenomenon refers to the shape of this curve, where increasing model capacity beyond interpolation leads to better prediction performance. A closely related phenomenon was discussed by Spigler et al. [2019] and referred to as a "jamming transition". Geiger et al. [2019, 2020] provided an empirical observation of this phenomenon for deep neural networks. Advani and Saxe [2017], Hastie et al. [2019], Bibas et al. [2019], Belkin et al. [2019], Mitra [2019], Muthukumar et al. [2020] studied the risk of minimum-norm interpolating methods and confirmed the double descent curve in several overparameterized linear regression models. Mei and Montanari [2019] computed the precise asymptotics of the test error and the training error for the random features model of Rahimi and Recht [2008] and showed that their model reproduces all the qualitative features of the double descent curve without assuming misspecification structures.

**Random features model.** The random features model can be regarded as a two-layer neural network with random weights $\boldsymbol{W}$ in the first layer. Despite its simplicity, the model captures some properties of deep neural networks. For instance, Hornik [1993] showed that neural networks with as few as a single hidden layer can uniformly approximate continuous functions in the limit of infinite width. Neal [1996], Williams [1997] showed that nets with a single hidden layer converge to a Gaussian process with analytic covariance function and that it is sensible to obtain good predictions with infinitely wide networks. Works connecting deeper neural networks with Gaussian processes include [Hazan and Jaakkola, 2015, Lee et al., 2017, Matthews et al., 2018, Novak et al., 2018, Garriga-Alonso et al., 2018, Yang, 2019]. The original paper Rahimi and Recht [2008] proposed the random features model as a randomized dimensionality reduction technique, for kernel Ridge regression, with randomized empirical kernel converging pointwise to the population kernel. Subsequent works on the random feature model as a kernel method include Bach [2013], Alaoui and Mahoney [2015], Rudi and Rosasco [2016], Bach [2017], Avron et al. [2017], Li et al. [2018], Zhang et al. [2019]. Kanagawa et al. [2018] reviewed the connections and equivalences between Gaussian processes and kernel methods. In Ghorbani et al. [2019], the approximation error and the generalization error of the random features model were studied and compared to that of neural tangent kernel models.

**Asymptotic normality and confidence intervals.** To prove the validity of our $1 - \alpha$ confidence intervals below, we employ techniques from the de-biasing literature for the Lasso in linear regression started Zhang and Zhang [2014], Van de Geer et al. [2014], Javanmard and Montanari [2014]. The subsequent work Bellec and Zhang [2019] provides the CLT that we use for our main result.

## 1.3  Notation

For any integer, $[n] = \{1, ..., n\}$. We will use index $i$ to loop/sum over $[n]$, index $j$ to loop over $[p]$ and $k$ to loop over $[d]$ for clarity. If clear from text, we will write only $n \to +\infty$ or only $p \to +\infty$ to indicate the asymptotic setting in (6). The big $O(\cdot)$, small $o(\cdot)$ notations and the convergence arrow $\to$ are endowed with the asymptotic setting (6) unless otherwise stated.

Let $\boldsymbol{0}_p$ denote the zero vector in $\mathbb{R}^p$ and $\boldsymbol{I}_p$ denote the identity matrix in $\mathbb{R}^{p \times p}$. For a matrix $\boldsymbol{X}$, the column vector $\boldsymbol{X}_j$ denotes the $j$-th column of $\boldsymbol{X}$, and $\boldsymbol{x}_i$ denotes the column vector corresponding to the $i$-th row. We let $\|\boldsymbol{x}_i\|_2$ denote the Euclidean norm of $\boldsymbol{x}_i$, and let $\|\boldsymbol{X}\|_{\mathrm{op}}$ denote the operator 2-norm (i.e., the spectral norm) of $\boldsymbol{X}$. Let $\mathbb{S}^{p-1}(r)$ be the Euclidean sphere in $\mathbb{R}^p$ with radius $r > 0$.

We let $\boldsymbol{e}_j$ denote the $j$-th canonical basis vector in $\mathbb{R}^p$, with 1 in the $j$-th coordinate and 0 elsewhere. The intercept is the scalar $\beta_0$ and $\beta_j$ is the $j$-th element of $\boldsymbol{\beta}$. For any vector $\boldsymbol{x} \in \mathbb{R}^p$, the matrix $\mathrm{diag}(\boldsymbol{x})$ is the diagonal matrix in $\mathbb{R}^{p \times p}$ with diagonal elements $x_1, \ldots, x_p$. We let $\mathbb{E}_G$ denote the conditional expectation given $\boldsymbol{X}, \boldsymbol{W}, \boldsymbol{\varepsilon}$. We let $\Phi(t) = (2\pi)^{-1/2} \int_{-\infty}^{t} e^{-x^2/2} dx$ denote the standard normal cumulative distribution function.

## 2  Asymptotic normality and confidence intervals based on NN estimates

**Central Limit Theorems (CLT) for derivatives of the trained neural network.** Our starting point is a search for CLT for the gradients of the trained neural network $\hat{f}(\boldsymbol{\xi})$ in (3) averaged over the training observations $\{\boldsymbol{x}_1, ..., \boldsymbol{x}_n\}$, i.e., whether for a fixed feature $j \in [p]$ the average (or some

weighted average) of $(\partial/\partial\xi_j)\hat{f}(\boldsymbol{x}_i)$ has asymptotically normal behavior. If $\hat{f}$ were independent from $\boldsymbol{x}_1, ..., \boldsymbol{x}_n$ this would follow from the classical CLT, however the training observations were used to construct $\hat{f}$ and this obvious dependence between $\hat{f}$ and $\boldsymbol{x}_1, ..., \boldsymbol{x}_n$ makes it a first glance hopeless to obtain CLT for averages of $(\partial/\partial\xi_j)\hat{f}(\boldsymbol{x}_i)$. Our results in Theorems 1 and 2 reveal that although classical CLT for $(\partial/\partial\xi_j)\hat{f}(\boldsymbol{x}_i)$ does not hold, a weighted average of $(\partial/\partial\xi_j)\hat{f}(\boldsymbol{x}_i)$ is asymptotically normal after an additive correction, in the sense that

$$\sum_{i=1}^{n} \Big( \frac{1 - H_{ii}}{\|\boldsymbol{y} - \boldsymbol{A}\widehat{\boldsymbol{\alpha}}\|_2} \Big) \frac{\partial \hat{f}}{\partial \xi_j}(\boldsymbol{x}_i) + [\text{additive correction}] \approx N(0,1), \tag{8}$$

where $(1 - H_{ii})/\|\boldsymbol{y} - \boldsymbol{A}\widehat{\boldsymbol{\alpha}}\|_2$ are weights of order $1/\sqrt{n}$ with $H_{ii}$ the diagonal elements of matrix

$$\boldsymbol{H} = \boldsymbol{A}(\boldsymbol{A}^\top \boldsymbol{A} + n\tau \boldsymbol{I}_d)^{-1} \boldsymbol{A}^\top \qquad \in \mathbb{R}^{n \times n}. \tag{9}$$

The validity of (8) is explained in the last paragraph preceding § 2.2. A first surprise of our results is that an asymptotic normality result of the form (8) holds for the derivatives of trained neural network. As we detail in the following subsections, a second surprise is that (8) can be used to construct CIs for the linear part of the models (4) and (5), even though neural networks are typically only used for predictions. A third surprise is that the length of these CIs features a double-descent phenomenon (cf. § 2.4 and 3). We make the following assumptions on the linear component of the data-generating process and the neural network weights $\boldsymbol{W}$.

**Assumption 1 (data-generating process)** *The entries of $\boldsymbol{X}$ and $\boldsymbol{\varepsilon}$ are mutually independent with $x_{ij} \sim^{iid} N(0,1)$ and $\varepsilon_i \sim^{iid} N(0, \theta_\varepsilon^2)$. The limiting magnitude of the signal $\|\boldsymbol{\beta}\|_2^2 \to \theta_\beta^2$ and the magnitude of the noise $\theta_\varepsilon^2$ are fixed constants independent of $n, d, p$.*

**Assumption 2 (NN estimation)** *The activation function $\sigma : \mathbb{R} \to \mathbb{R}$ is $L$-Lipschitz so that the derivative $\sigma' : \mathbb{R} \to \mathbb{R}$ exists almost everywhere. The weight matrix $\boldsymbol{W}$ is independent with $\boldsymbol{X}, G, \boldsymbol{\varepsilon}$ and satisfies (i) $\|\boldsymbol{W}\|_{op}$ is bounded uniformly over the asymptotic setting (6), or (ii) the entries of $\boldsymbol{W}$ are $w_{kj} \sim^{iid} N(0, 1/p)$.*

Assumption 2(i) is implied by Assumption 2(ii) with high-probability (c.f., Corollary 7.3.3 in Vershynin [2018]). For practical purposes, Assumption 2(i) is satisfied if the weight matrices $\boldsymbol{W}$ are trained on datasets $(\widetilde{\boldsymbol{y}}, \widetilde{\boldsymbol{X}}) \in \mathbb{R}^{n'} \times \mathbb{R}^{n' \times p}$ independent of $(\boldsymbol{y}, \boldsymbol{X})$ with $\|\boldsymbol{W}\|_{op} = 1$ (Arjovsky et al. [2015], Miyato et al. [2018]); or trained with respect to certain weight regularized loss function, e.g.,

$$\boldsymbol{W} = \text{argmin}_{\boldsymbol{W}' \in \mathbb{R}^{d \times p}} \big\{ \tfrac{1}{n'} \mathcal{L}(\widetilde{\boldsymbol{y}}, \widetilde{\boldsymbol{X}}, \boldsymbol{W}') + \tfrac{\kappa}{2} \|\boldsymbol{W}'\|_*^2 \big\}, \qquad \kappa > 0, \tag{10}$$

where the penalty norm $*$ is either the Frobenius norm (Krogh and Hertz [1992]) or the spectral norm (Yoshida and Miyato [2017]), provided that $(1/n')\mathcal{L}(\widetilde{\boldsymbol{y}}, \widetilde{\boldsymbol{X}}, \boldsymbol{0}_{d \times p})$ are uniformly bounded in large probability, for e.g., through the magnitude of $(1/n')\|\widetilde{\boldsymbol{y}}\|_2^2$.

## 2.1 Asymptotic normality and confidence intervals in the linear model

**Definition 1** *Following (2) and § 1.3, we define*

$$\zeta_L(\boldsymbol{e}_j) = \boldsymbol{X}_j^\top (\boldsymbol{y} - \boldsymbol{A}\widehat{\boldsymbol{\alpha}}) - \text{trace} \left[ \boldsymbol{T}_0(\boldsymbol{e}_j) + \boldsymbol{T}_1(\boldsymbol{e}_j) + \boldsymbol{T}_L(\boldsymbol{e}_j) \right], \tag{11}$$

*where $\boldsymbol{H}$ is given in (9) and*

$$\begin{aligned}
\boldsymbol{T}_0(\boldsymbol{e}_j) &= -(\boldsymbol{I}_n - \boldsymbol{H}) \text{diag} \left( \sigma'(\boldsymbol{X}\boldsymbol{W}^\top) \text{diag}(\widehat{\boldsymbol{\alpha}}) \boldsymbol{W} \boldsymbol{e}_j \right), \\
\boldsymbol{T}_1(\boldsymbol{e}_j) &= -\boldsymbol{A}(n\tau \boldsymbol{I}_d + \boldsymbol{A}^\top \boldsymbol{A})^{-1} \text{diag}(\boldsymbol{W}\boldsymbol{e}_j)\sigma'(\boldsymbol{W}\boldsymbol{X}^\top) \text{diag}(\boldsymbol{y} - \boldsymbol{A}\widehat{\boldsymbol{\alpha}}), \\
\boldsymbol{T}_L(\boldsymbol{e}_j) &= (\boldsymbol{I}_n - \boldsymbol{H})\boldsymbol{\beta}^\top \boldsymbol{e}_j.
\end{aligned} \tag{12}$$

**Theorem 1** *Let $t \in \mathbb{R}$. Under model (4), Assumption 1 and 2 with notation in Definition 1,*

$$\max_{j \in J_p} \Big| \mathbb{P}\Big( \frac{\zeta_L(\boldsymbol{e}_j)}{\|\boldsymbol{y} - \boldsymbol{A}\widehat{\boldsymbol{\alpha}}\|_2} \leq t \Big) - \Phi(t) \Big| \to 0 \tag{13}$$

*for some $J_p \subset [p]$ satisfying $|J_p|/p \geq 1 - \log(p)/p \to 1$.*

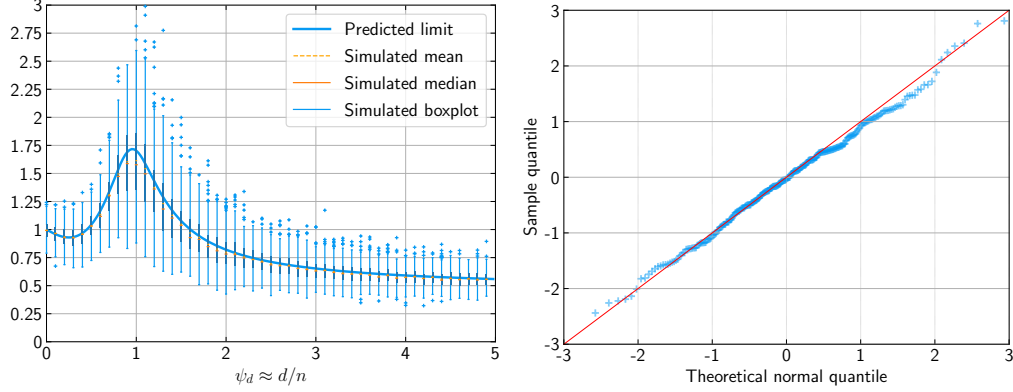

Figure 2: Simulated boxplots for squared length $nL^2$ and theoretical limit $V(\psi_d, \psi_p, \lambda, \rho, \sigma)$ (left). Quantile-Quantile plot for $\zeta_L(e_1)/\|y - A\widehat{\alpha}\|_2$ (right). Data are generated in a random quadratic model (Example 1), with dimensions $\psi_p = 1/3, p = 100, n = 300$, Ridge raw penalty $\lambda = 10^{-3}$, signal-noise level $\theta_0^2 = 0, \theta_{\beta}^2 = 2/3, \theta_{\mathrm{NL}}^2 = 1/6, \theta_{\varepsilon}^2 = 1/6$ thus SNR $\rho = 2$, activation function $\sigma(x) = \max(x, 0)$, signal $\beta \sim \mathrm{Unif}(\mathbb{S}^{p-1}(\theta_{\beta}))$ and the weights $w_{kj} \sim$ iid $N(0, 1/p)$. The boxplots are each based on 300 data points. The QQ-plot has sample size 300.

Theorem 1 provides the asymptotic normality result in the linear model (4). Asymptotic normality is confirmed in simulations with QQ-plots such as that in Figure 2. Asymptotic normality holds for a subset $J_p \subset [p]$ that contains "most covariates" in the sense except for at most $\log p$ indices in $[p]$, the quantity $\zeta_L(e_j)$ is asymptotically normal.

This main theorem provides us an asymptotically normal quantity $\zeta_L(e_j)/\|y - A\widehat{\alpha}\|_2$ affine in $\beta_j$ for most $j \in [p]$,

$$\frac{\zeta_L(e_j)}{\|y - A\widehat{\alpha}\|_2} = -\frac{\mathrm{trace}(I_n - H)}{\|y - A\widehat{\alpha}\|_2}\beta_j + \frac{X_j^\top(y - A\widehat{\alpha}) - \mathrm{trace}[T_0(e_j) + T_1(e_j)]}{\|y - A\widehat{\alpha}\|_2} \approx N(0, 1).$$

The only unobserved term in the above display is $\beta_j$. Consequently $1 - \alpha$ CIs for $\beta_j$ are obtained based on the $\frac{\alpha}{2}$-th quantile and the $1 - \frac{\alpha}{2}$-th quantile of the standard normal distribution, i.e.,

$$\max_{j \in J_p}\left|\mathbb{P}\left(\frac{\zeta_L(e_j)}{\|y - A\widehat{\alpha}\|_2} \in \left[\Phi^{-1}(\alpha/2), \Phi^{-1}(1 - \alpha/2)\right]\right) - (1 - \alpha)\right| \to 0. \tag{14}$$

Those CIs for $\beta_j$ have squared length in proportional to $L^2 := \|y - A\widehat{\alpha}\|_2^2/(\mathrm{trace}(I_n - H))^2$.

**Theorem 1 implies** (8). By (3), the weighed average in (8) is equal to $-\mathrm{trace}[T_0(e_j)]/\|y - A\widehat{\alpha}\|_2$ in Theorem 1. The term [additive correction] in (8) includes traces of $T_1, T_L$ and the first term in (11), thus the validity of (8) follows by Theorem 1.

## 2.2  Asymptotic normality for model under non-linear perturbation.

This section provides similar the asymptotic normality results for the model (5) under non-linear perturbation $G(\cdot)$. We require the following assumptions on the nonlinear perturbation $G(\cdot)$ in (5).

**Assumption 3** *As $n, p, d \to \infty$, the intercept $\beta_0 \to \theta_0$ and $\mathbb{E}\left[(G(x))^2\right] \to \theta_{\mathrm{NL}}^2$ for $x \sim N(0_p, I_p)$. Here, $\theta_0, \theta_{\mathrm{NL}}$ are fixed constant independent of $n, d, p$. For any realization of the random function $G$, $G$ is weakly differentiable in $x$ with mean zero and with squared integrable gradient uniformly bounded, i.e.,$\mathbb{E}_G[G(x)] = 0$ and $\mathbb{E}[\|\nabla G(x)\|_2^2] = O(1)$.*

**Assumption 4** *The nonlinear perturbation $G$ is generated once and for all and is the same for every $i = 1, ..., n$ so that conditionally on $G$, the observations $(x_i, y_i)_{i=1,...,n}$ are iid. The nonlinear perturbation $G$ is a centered random process indexed by $x \in \mathbb{R}^p$, satisfying*

$$\mathbb{E}_G[G(x)] = 0, \qquad \mathbb{E}_G[G(x_1)G(x_2)] = \Sigma_p(x_1^\top x_2/p), \tag{15}$$

*for all $x_1, x_2 \in \mathbb{R}^p$. The covariance function $\Sigma_p$ satisfies for some $\delta > 0$,*

*(i)* $\Sigma_p''(x)$ *exists for $x$ in $(1-\delta, 1+\delta)$ and $(-\delta, \delta)$. $\Sigma_p''(x)$ is $L_2$-Lipschitz in $(1-\delta, 1+\delta)$ and $(-\delta, \delta)$ for some constant $L_2 > 0$ independent of $n, d, p$.*

*(ii)* $\lim_{p \to +\infty} \Sigma_p'(0) = 0$ *and* $\max\left(\left|\Sigma_p'(1)\right|, \left|\Sigma_p''(0)\right|, \left|\Sigma_p''(1)\right|\right) = O(1)$.

The above assumption ensures that the covariance function $\Sigma_p$ is smooth enough in a neighborhood of 1 and 0. Similar locally smooth conditions are discussed in El Karoui [2010] to study the largest eigenvalue of kernel random matrices of the form $g(\boldsymbol{X}\boldsymbol{X}^\top/p)$ for locally smooth $g$. The reason that we introduce such local smoothness is to allow a Taylor expansion of $\Sigma_p$ around 1 and 0. The above assumptions include a large class of non-linear perturbations. A simple example is the following random quadratic perturbation model considered in Mei and Montanari [2019], which has $\Sigma_p(t) = \theta_{\mathrm{NL}}^2(t^2 - 1/p)$.

**Example 1 (random quadratic model)** *We assume that for $i \in [n]$,*

$$y_i = \beta_0 + \boldsymbol{x}_i^\top \boldsymbol{\beta} + G(\boldsymbol{x}_i) + \varepsilon_i, \quad G(\boldsymbol{x}_i) = (\theta_{\mathrm{NL}}/p)\left[\boldsymbol{x}_i^\top \boldsymbol{G}\boldsymbol{x}_i - \mathrm{trace}(\boldsymbol{G})\right], \tag{16}$$

*where $\boldsymbol{G} \in \mathbb{R}^{p \times p}$ has standard normal entries $g_{j_1, j_2} \sim^{iid} N(0,1)$, independent of $(\boldsymbol{X}, \boldsymbol{W}, \boldsymbol{\varepsilon})$.*

Under these assumptions, the following asymptotic normality result holds.

**Theorem 2** *Let $t \in \mathbb{R}$. Under model* (5)*, Assumption 1, 2, 3 and 4 and Definition 1,*

$$\max_{j \in J_p}\left|\mathbb{P}\left(\frac{\zeta_L(\boldsymbol{e}_j)}{\|\boldsymbol{y} - \boldsymbol{A}\widehat{\boldsymbol{\alpha}}\|_2} \leq t\right) - \Phi(t)\right| \to 0 \tag{17}$$

*for some $J_p \subset [p]$ satisfying $|J_p|/p \geq 1 - \log(p)/p \to 1$.*

The above theorem provides us the CI in § 2.1. When we assume no intercept nor perturbation, Theorem 1 is obtained as a special case of Theorem 2.

## 2.3 Asymptotic normality in general directions

The functions $\boldsymbol{T}_0, \boldsymbol{T}_1, \boldsymbol{T}_L, \zeta_L$ in (11) and (12) are linear in $\boldsymbol{e}_j$. As explained in the supplement, our asymptotic normality results still hold if one replaces $\boldsymbol{e}_j$ with a direction $\boldsymbol{u}_0 \in \mathbb{R}^p$ satisfying $\|\boldsymbol{u}_0\|_2 = 1$ in (13), (11), (12) and (17). The convergence will then be uniform over $\boldsymbol{u}_0 \in S_p$ where $S_p$ is a subset of the unit sphere $\mathbb{S}^{p-1}(1) \subset \mathbb{R}^p$ satisfying, in terms of relative volume, $|S_p|/|\mathbb{S}^{p-1}(1)| \to 1$ as $n, p, d \to +\infty$. Hence we can construct CIs for the unknown parameter $\boldsymbol{u}_0^\top \boldsymbol{\beta}$ instead of $\beta_j$, for most direction $\boldsymbol{u}_0$ in the unit sphere. We stick to canonical basis vectors $\boldsymbol{e}_j$ and $\beta_j$ as the unknown parameter in the main text in order to keep notation reasonably self-contained.

## 2.4 Predicting CI length given $(\psi_p, \psi_d)$ and the large width limit $(\psi_d \to \infty)$

Let $L^2 := \|\boldsymbol{y} - \boldsymbol{A}\widehat{\boldsymbol{\alpha}}\|_2^2/(\mathrm{trace}(\boldsymbol{I}_n - \boldsymbol{H}))^2$ denote the squared length of our CIs with confidence level $1 - \alpha = \mathbb{P}(|N(0,1)| \leq 1/2) \approx 0.383$. This quantile is chosen for convenience; different $\alpha$ leads to a length proportional to $L$ up to a constant depending on $\alpha$ only. As a consequence of the characterization of the generalization and training error for the NN estimate (1) in Mei and Montanari [2019], we find, as explained in the supplement, that under either (a) the linear model (4) with Assumption 1 and Assumption 2(ii), i.e., under iid normal entries of $\boldsymbol{W}$, or (b) the setting of Mei and Montanari [2019] which is comparable to ours,

$$\mathrm{Var}(y_1) \underset{n \to +\infty}{\to} \theta_{\boldsymbol{\beta}}^2 + \theta_{\boldsymbol{\varepsilon}}^2 + \theta_{\mathrm{NL}}^2, \qquad \frac{nL^2}{\mathrm{Var}(y_1)} \underset{n \to +\infty}{\overset{\mathbb{P}}{\to}} V, \qquad \lim_{\psi_d \to +\infty} V = R, \tag{18}$$

where $V := V(\psi_d, \psi_p, \lambda, \rho, \sigma)$ and $R := R(\psi_p, \lambda, \rho, \sigma)$ are non-random functions, the signal-to-noise ratio (SNR) is $\rho := \theta_{\boldsymbol{\beta}}^2/(\theta_{\boldsymbol{\varepsilon}}^2 + \theta_{\mathrm{NL}}^2)$, and where $\sigma$ represents the choice of activation function.

The quantities $V$ and $R$ are calculated by solving quartic polynomial equations. The specific forms of the quartic equations, which can be deduced from [Mei and Montanari, 2019], are given in the supplement. These equations can be solved numerically, with the code also found in the supplement. These predicted values for $V$ and $R$ obtained by solving the quartic equations reliably predict the observed squared lengths of our CIs in simulations as shown on the left of Figure 2. Solving these

quartic equations let us quickly predict the average CI length as shown in Figures 1 and 3. This enables comparing choices of $\lambda, \rho, \psi_d$ and activation function $\sigma$ in terms of the resulting CI length. For instance, it lets us answer which of several activation functions produces the smallest CIs.

We should note that Mei and Montanari [2019] assumes that the random $\boldsymbol{x}_i, \boldsymbol{w}_k$ are distributed uniformly on spheres of radius $\sqrt{p}$ and 1 respectively, and that the random process $G(\boldsymbol{x})$ is a centered Gaussian process satisfying additional mild conditions. Since random vectors uniformly distributed on the sphere $\mathbb{S}^{p-1}(\sqrt{p})$ are close to standard normal vectors (e.g., in the sense of § 3.3.3 in Vershynin [2018]), we expect the results of Mei and Montanari [2019] to be valid for Gaussian $\boldsymbol{x}_i, \boldsymbol{w}_k$ as well. The supplement provides a sketch of proof of the validity of the limits in (18) under the linear model (4) with Gaussian $w_{kj} \sim$ iid $N(0, 1/p)$ and $x_{ij} \sim$ iid $N(0, 1)$.

# 3   Numerical results and some observations

With the notation for $L, V, R$ in § 2.4, we display several numerical plots: Figure 1 on p. 3 which highlights the double-descent phenomenon for CI lengths as $\psi_d$ increases for fixed $(\psi_p, \lambda, \rho, \sigma)$; Figure 2 on p. 6 which highlights asymptotic normality and the accuracy of the numerical prediction for the CI length; and Figure 3 below which displays the squared length $R$ of CIs in the infinite width limit for several activation functions.

A common feature of these plots is the double-descent phenonemon in terms of CI length, suggesting that the length of CIs mimic the generalization error of the corresponding estimate (1). The fact that CI lengths would feature double-descent curves was a surprise to us.

The right plot in Figure 1 highlights the advantage of interpolation (small $\lambda$): for large networks (large $\psi_d$), the CI length is smaller for small $\lambda$. To our knowledge, this is a first instance of observing the advantage of interpolating methods for uncertainty quantification in the form of confidence intervals.

Pennington and Worah [2017] proposed a special class of activation functions satisfying $\gamma_1 := \mathbb{E}\left[\sigma'(Z)\right] = 0$ for $Z \sim N(0, 1)$. They showed that those activation functions maintain approximate isometry at initialization, that is, the eigenvalues of the data covariance matrix are unchanged in distribution after passing through a single nonlinear layer of the network. They conjectured that activation functions with $\gamma_1 = 0$ would improve the training time and performance.

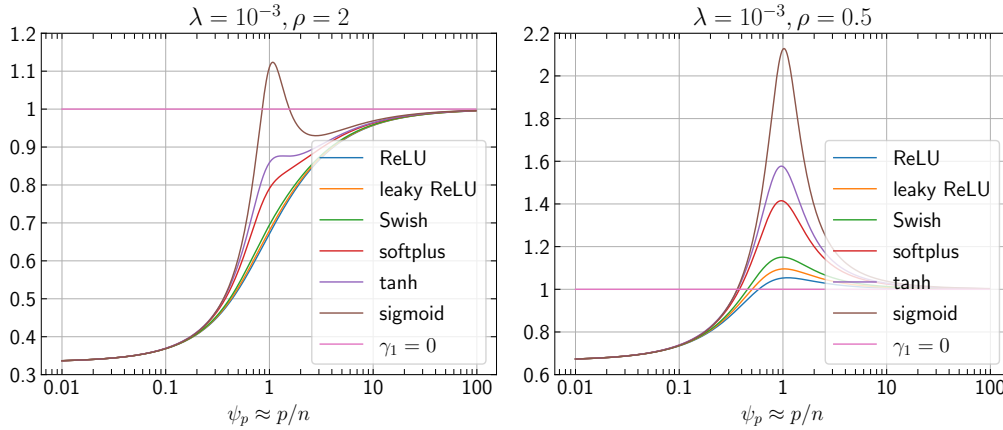

Figure 3: The indicator $R(\psi_p, \lambda, \rho, \sigma)$ (see (18) on p. 7) of the squared length in the infinite width. The activation functions are ReLU $\max(x, 0)$, leaky ReLU $\max(x, 0.05x)$, Swish $\frac{x}{1+e^{-x}}$, softplus $\ln(1 + \exp(x))$, tanh $\frac{e^x - e^{-x}}{e^x + e^{-x}}$ and sigmoid $\frac{1}{1+e^{-x}}$. Activation functions $\sigma(x)$ with $\gamma_1 := \mathbb{E}\left[\sigma'(Z)\right] = 0$ for $Z \sim N(0, 1)$ have limit $R = 1$ always. We consider Ridge penalty parameter $\lambda = 10^{-3}$ and signal-to-noise ratio $\rho \in \{2, 0.5\}$, corresponding to the high SNR interpolating regime (left) and low SNR interpolating regime (right).

Figure 3 shows that the activation functions with $\gamma_1 = 0$ maintain the constant limit $R(\psi_p, \lambda, \rho, \sigma) = 1$ for all $\psi_p, \lambda, \rho$. The activation functions commonly used in practice (e.g., ReLU, leaky ReLU, softplus) enable small $R < 1$ in the high SNR interpolating regime (SNR $\gg 1$, small $\lambda$); furthermore

ReLU activations consistently produce smaller CI lengths than other choices of $\sigma$. In the low SNR and interpolating regime (SNR $\ll 1$, small $\lambda$), $R$ can be larger than 1 for commonly used activation functions: here activation functions with $\gamma_1 = 0$ perform better in terms of CI lengths. The supplement provides further comparisons in terms of CI lengths, and plots comparable to Figures 1 and 3 in different regimes.

**Conclusion.** We have provided an explicit construction of valid $1 - \alpha$ CIs based on the NN estimate (1), as well as a toolbox to numerically predict the length of this CIs. More comparisons, in terms of CI length, can be easily plotted for different values of the parameters $\psi_p, \rho, \lambda$ and different activation functions using the code provided in the supplement. We hope that this toolbox will enable more insights on the role of activation functions and other network parameters for uncertainty quantification purposes.

## Broader Impact

Our work provides a first step towards frequentist uncertainty quantification, in the form of confidence intervals, for the NN estimate (1). Uncertainty quantification and confidence intervals are important for decision making in society, where decisions with long-lasting effects are made based on summary statistics computed on characteristics of individual human beings. As two concrete examples, consider the GPA awarded by Universities to their undergraduate students, or the credit score assigned to individuals in the US regarding their capacity to manage their debts. Graduate schools would refuse admissions to student applying with a GPA under a fixed threshold, and banks would refuse loans to individuals with a credit score under a threshold. In order to make fairer decisions, graduate schools and banks ought to consider the uncertainty of these summary statistics, for instance by comparing the admission threshold to some confidence interval [GPA $-$ STD, GPA $+$ STD] that takes into account the uncertainty of computing the GPA summary statistic. Similarly, banks should take into account confidence intervals when making loan decisions about an individual based on their credit score summary statistic. Although the neural network model (1) studied in the present paper is admittedly simple and far from the state-of-the-art networks used for predictions in practice, we hope that our results will provide useful ideas and inspire new results of frequentist uncertainty quantification for machine learning tools that are implemented in societal decision making and praised for their prediction properties, but, so far, lack provable uncertainty quantification guarantees.

## Acknowledgments and Disclosure of Funding

Y.S. would like to thank Linjun Zhang for helpful conversations. P.C.B. was partially supported by the NSF Grants DMS-1811976 and CAREER award DMS-1945428.

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
