[Supplementary Material]

SUPPLEMENTARY MATERIAL:

ASYMPTOTIC NORMALITY AND CONFIDENCE INTERVALS FOR DERIVATIVES OF 2-LAYER NEURAL NETWORK IN THE RANDOM FEATURES MODEL

We gather here the supporting propositions, the proofs and additional figures. The python code used to generate the figures is also attached. In Figure 4 on page 2 we plot $R(\psi_p, \lambda, \rho, \sigma)$ under various regimes as a supplement to Figure 3.

## CONTENTS

## APPENDIX A. OUTLINE OF THE PROOF OF THE MAIN RESULT

Since Theorem 1 can be regarded as a special case of Theorem 2, it suffices to prove Theorem 2. The formal proof of Theorem 2 is provided in Section D.1. The proof combines three supporting results using Slutsky's Theorem: the asymptotic results

$$(\text{A.1}) \quad \frac{\zeta(\boldsymbol{e}_j)}{(\text{Var}_j\,[\zeta(\boldsymbol{e}_j)])^{1/2}} \xrightarrow{d} N(0,1), \qquad \frac{\|\boldsymbol{y} - \boldsymbol{A}\widehat{\boldsymbol{\alpha}}\|_2}{(\text{Var}_j\,[\zeta(\boldsymbol{e}_j)])^{1/2}} \xrightarrow{\mathbb{P}} 1 \qquad \text{and} \qquad \frac{\zeta(\boldsymbol{e}_j) - \zeta_{\text{L}}(\boldsymbol{e}_j)}{\|\boldsymbol{y} - \boldsymbol{A}\widehat{\boldsymbol{\alpha}}\|_2} \xrightarrow{\mathbb{P}} 0.$$

The first two are proved in Proposition C.2.5 while the third is proved in Proposition C.2.6. The asymptotic results are shown to hold for most $j \in [p]$, in the sense that the current proof shows that (A.1) holds for all $j \in [p] \setminus J$ for some set $J$ with negligible cardinality compared to $p$. More specifically, Proposition C.2.5 leverages the flexible central limit theorems in Bellec and Zhang [2019] to obtain asymptotic normality of $\frac{\zeta(\boldsymbol{e}_j)}{(\text{Var}_j[\zeta(\boldsymbol{e}_j)])^{1/2}}$. Proposition C.2.6 shows that the nonlinear component (induced by the nonlinear perturbation) in $\zeta(\boldsymbol{e}_j) - \zeta_L(\boldsymbol{e}_j)$ is negligible.

APPENDIX B. FIGURE 4

FIGURE 4. The indicator $R(\psi_p, \lambda, \rho, \sigma)$ (see Definition C.3.2) of the squared length in the infinite width. The activation functions are ReLU $\max(x, 0)$, leaky ReLU $\max(x, 0.1x)$, Swish $\frac{x}{1+e^{-x}}$, softplus $\ln(1+\exp(x))$, tanh $\frac{e^x - e^{-x}}{e^x + e^{-x}}$ and sigmoid $\frac{1}{1+e^{-x}}$. Activation functions $\sigma(x)$ with $\gamma_1 := \mathbb{E}\left[\sigma'(Z)\right] = 0$ for $Z \sim N(0,1)$ have limit $R = 1$ always. We consider Ridge penalty parameter $\lambda \in \{10^{-3}, 1\}$ and signal-to-noise ratio $\rho \in \{5, 2, 0.5, 0.2\}$.

Appendix C. Supporting propositions

C.1. **Notation and constants.** The proof will require some further notation and constants.

C.1.1. *Notation.* Let $\|\boldsymbol{M}\|_F$ denote the Frobenius norm of a matrix $\boldsymbol{M}$ and $\|\boldsymbol{M}\|_{op}$ its operator norm.

For the functions $f : \mathbb{R}^p \to \mathbb{R}$ and $G : \mathbb{R}^p \to \mathbb{R}$, denote by $\partial_j$ the partial derivative with respect to the $j$-th coordinate for each $j \in [p]$, as well as

$$\boldsymbol{f} := (f(\boldsymbol{x}_i))^{i \in [n]} \in \mathbb{R}^n, \quad \boldsymbol{f}' := [\partial_j f(\boldsymbol{x}_i)]^{(i,j) \in [n] \times [p]} \in \mathbb{R}^{n \times p},$$

$$\boldsymbol{G} := (G(\boldsymbol{x}_i))^{i \in [n]} \in \mathbb{R}^n, \quad \boldsymbol{G}' := [\partial_j G(\boldsymbol{x}_i)]^{(i,j) \in [n] \times [p]} \in \mathbb{R}^{n \times p}$$

where $\boldsymbol{x}_1 \in \mathbb{R}^p, ..., \boldsymbol{x}_n \in \mathbb{R}^p$ are the observed feature vectors, i.e., the rows of $\boldsymbol{X}$.

Let $\mathbb{E}_j$ and $\mathrm{Var}_j$ be the conditional expectation and the conditional variance given $(\boldsymbol{X}_{-j}, G, \boldsymbol{W}, \boldsymbol{\varepsilon})$, where $\boldsymbol{X}_{-j}$ is the matrix $\boldsymbol{X}$ with $j$-th column removed. Let $\mathbb{E}_{\boldsymbol{\varepsilon}}$ be the conditional expectation given $\boldsymbol{X}, G, \boldsymbol{W}$. Let $\mathbb{P}$ and $\mathbb{E}$ be with respect to the total probability $\mathbb{P}_{\boldsymbol{X}, \boldsymbol{W}, \boldsymbol{\varepsilon}, G}$. We let $a_+$ denote $\max(a, 0)$.

We will also consider gradients with respect to columns of $\boldsymbol{X}$ or with respect to the noise $\boldsymbol{\varepsilon}$. Consider an expression $\boldsymbol{r} \in \mathbb{R}^q$ which is a function of $(\boldsymbol{X}, \boldsymbol{\varepsilon}, \boldsymbol{W}, G)$.

- For two given indices $(i, j) \in [n] \times [p]$, the column vector $\partial_{x_{ij}} \boldsymbol{r}$ has the same dimension as $\boldsymbol{r}$ and denotes the partial derivative of $\boldsymbol{r}$ with respect to $x_{ij}$ while

$$((x_{i',j'})_{i' \in [n], j' \in [p]:(i',j') \neq (i,j)}, \boldsymbol{\varepsilon}, \boldsymbol{W}, G)$$

  remain fixed. If the dimension $q$ of $\boldsymbol{r}$ is greater than 1, then $\nabla_{x_{ij}}$ acts componentwise on the components of $\boldsymbol{r}$.
- The matrix $\nabla_{\boldsymbol{X}_j} \boldsymbol{r}$ is the derivative (in the sense of the Frechet derivative) of $\boldsymbol{r}$ with respect to the $j$-th column $\boldsymbol{X}_j$ of $\boldsymbol{X}$ while $(\boldsymbol{X}_{-j}, \boldsymbol{\varepsilon}, \boldsymbol{W}, G)$ (which are random variables independent of $\boldsymbol{X}_j$) remain fixed; if the dimension $q$ of $\boldsymbol{r}$ is greater than 1, then $\nabla_{\boldsymbol{X}_j}$ acts componentwise on the components of $\boldsymbol{r}$. For instance, the derivative of the residuals $\boldsymbol{r} = \boldsymbol{y} - \boldsymbol{A}\widehat{\boldsymbol{\alpha}}$ with respect to $\boldsymbol{X}_j$ while $(\boldsymbol{X}_{-j}, \boldsymbol{\varepsilon}, \boldsymbol{W}, G)$ remain fixed is the $n \times n$ matrix $\nabla_{\boldsymbol{X}_j}(\boldsymbol{y} - \boldsymbol{X}\widehat{\boldsymbol{\alpha}})$ whose $i$-th column is $\partial_{x_{ij}}(\boldsymbol{y} - \boldsymbol{X}\widehat{\boldsymbol{\alpha}})$, the directional derivative with respect to the $(i, j)$-th entry of $\boldsymbol{X}$. We may refer to $\nabla_{\boldsymbol{X}_j}(\boldsymbol{y} - \boldsymbol{A}\widehat{\boldsymbol{\alpha}})$ as the Jacobian of the map $\boldsymbol{X}_j \mapsto \boldsymbol{y} - \boldsymbol{A}\widehat{\boldsymbol{\alpha}}$.
- Similarly, $\nabla_{\boldsymbol{\varepsilon}} \boldsymbol{r}$ is the derivative of $\boldsymbol{r}$ with respect to the noise vector $\boldsymbol{\varepsilon}$ while $(\boldsymbol{X}, \boldsymbol{W}, G)$ (which are random variables independent of $\boldsymbol{\varepsilon}$) remain fixed. If the dimension $q$ of $\boldsymbol{r}$ is greater than 1, then $\nabla_{\boldsymbol{\varepsilon}}$ acts componentwise on the components of $\boldsymbol{r}$. For instance, the derivative of the residuals $\boldsymbol{y} - \boldsymbol{A}\widehat{\boldsymbol{\alpha}}$ with respect to $\boldsymbol{\varepsilon}$ while $(\boldsymbol{X}, \boldsymbol{W}, G)$ remain fixed is the $n \times n$ matrix $\nabla_{\boldsymbol{\varepsilon}}(\boldsymbol{y} - \boldsymbol{X}\widehat{\boldsymbol{\alpha}})$ whose $i$-th column is the $i$-th entry of $\boldsymbol{\varepsilon}$. We may refer to $\nabla_{\boldsymbol{\varepsilon}}(\boldsymbol{y} - \boldsymbol{A}\widehat{\boldsymbol{\alpha}})$ as the Jacobian of the map $\boldsymbol{\varepsilon} \mapsto \boldsymbol{y} - \boldsymbol{A}\widehat{\boldsymbol{\alpha}}$.
- *(This is only used in Section C.4)* If $\boldsymbol{u}_0 \in \mathbb{R}^p$ has $\|\boldsymbol{u}_0\|_2 = 1$, we define $\boldsymbol{X}_0 = \boldsymbol{X}\boldsymbol{u}_0$. Then $\nabla_{\boldsymbol{X}_0} \boldsymbol{r}$ is the derivative with respect to $\boldsymbol{X}_0$ while $(\boldsymbol{X}(\boldsymbol{I}_p - \boldsymbol{u}_0\boldsymbol{u}_0^\top), \boldsymbol{\varepsilon}, \boldsymbol{W}, G)$, (which are random variables independent of $\boldsymbol{X}_0$) remain fixed. If $\boldsymbol{r} = \boldsymbol{y} - \boldsymbol{X}\widehat{\boldsymbol{\alpha}}$, the Jacobian $\nabla_{\boldsymbol{X}_0}(\boldsymbol{y} - \boldsymbol{X}\widehat{\boldsymbol{\alpha}})$ is the matrix whose $i$-th column is the derivative with respect to the $i$-th entry of $\boldsymbol{X}_0$.

C.1.2. *Constants.* The following positive finite constants $L, c_1, c_8, \delta, L_2$ are independent of $n, p, d$.

  (i) The activation function $\sigma$ is Lipschitz with constant $L$.
  (ii) $\boldsymbol{W}$ is deterministic with $\|\boldsymbol{W}\|_{op} < c_1$.
  (iii) For some $\delta > 0$, $\Sigma_p''(x)$ exists in $(1 - \delta, 1 + \delta)$ and $(-\delta, \delta)$ and is $L_2$-Lipschitz in $(1 - \delta, 1 + \delta)$ and $(-\delta, \delta)$.
  (iv) $\max\left(\left|\Sigma_p'(1)\right|, \left|\Sigma_p''(0)\right|, \left|\Sigma_p''(1)\right|\right) < c_8$.

For the proof of the asymptotic normality result in Theorem 2, we only consider the case when $\boldsymbol{W}$ is deterministic. The proof is applicable for $\boldsymbol{W}$ with entries iid $N(0, 1/p)$ by Corollary 7.3.3 in Vershynin [2018] by conditioning on $\boldsymbol{W}$.

C.2. **Asymptotic normality result.** In this section, we present supporting propositions for the asymptotic normality result in Theorem 2. Proposition C.2.1 provides the calculation of the Jacobian matrix $\nabla_{\boldsymbol{X}_j}(\boldsymbol{y} - \boldsymbol{A}\widehat{\boldsymbol{\alpha}})$.

**Proposition C.2.1** (Calculation of the Jacobian matrix). *Under model* (5),

$$
\begin{aligned}
\text{(C.1)} \qquad \nabla_{\boldsymbol{X}_j}(\boldsymbol{y} - \boldsymbol{A}\widehat{\boldsymbol{\alpha}}) = &- (\boldsymbol{I}_n - \boldsymbol{H}) \operatorname{diag}(\sigma'(\boldsymbol{X}\boldsymbol{W}^\top) \operatorname{diag}(\boldsymbol{W}\boldsymbol{e}_j)\widehat{\boldsymbol{\alpha}}) \\
&- \boldsymbol{A}(\boldsymbol{A}^\top\boldsymbol{A} + n\tau\boldsymbol{I}_n)^{-1} \operatorname{diag}(\boldsymbol{W}\boldsymbol{e}_j)\sigma'(\boldsymbol{W}\boldsymbol{X}^\top)\operatorname{diag}(\boldsymbol{y} - \boldsymbol{A}\widehat{\boldsymbol{\alpha}}) \\
&+ (\boldsymbol{I}_n - \boldsymbol{H}) \left[ (\boldsymbol{e}_j^\top\boldsymbol{\beta})\boldsymbol{I}_n + \operatorname{diag}(\boldsymbol{G}'\boldsymbol{e}_j) \right].
\end{aligned}
$$

Under Definitions 1 and C.2.1, we have

$$
\text{(C.2)} \qquad \nabla_{\boldsymbol{X}_j}(\boldsymbol{y} - \boldsymbol{A}\widehat{\boldsymbol{\alpha}}) = \boldsymbol{T}_0(\boldsymbol{e}_j) + \boldsymbol{T}_1(\boldsymbol{e}_j) + \boldsymbol{T}_{\mathrm{L}}(\boldsymbol{e}_j) + \boldsymbol{T}_{\mathrm{NL}}(\boldsymbol{e}_j).
$$

**Definition C.2.1.** *Let $\zeta_{\mathrm{L}}(\boldsymbol{e}_j)$ be in Definition 1. Let*

$$
\text{(C.3)} \qquad \zeta(\boldsymbol{e}_j) = \zeta_{\mathrm{L}}(\boldsymbol{e}_j) - \operatorname{trace}\left[\boldsymbol{T}_{\mathrm{NL}}(\boldsymbol{e}_j)\right],
$$

*where*

$$
\begin{aligned}
\text{(C.4)} \qquad \boldsymbol{T}_{\mathrm{NL}}(\boldsymbol{e}_j) &= (\boldsymbol{I}_n - \boldsymbol{H}) \operatorname{diag}\left(\boldsymbol{G}'\boldsymbol{e}_j\right), \\
\boldsymbol{G}' &= [\partial_j G(\boldsymbol{x}_i)]^{(i,j)\in[n]\times[p]} \in \mathbb{R}^{n\times p}.
\end{aligned}
$$

Proposition C.2.2 provides upper bounds on the components of the Jacobian matrix $\nabla_{\boldsymbol{X}_j}(\boldsymbol{y} - \boldsymbol{A}\widehat{\boldsymbol{\alpha}})$.

**Proposition C.2.2.** *Let $\boldsymbol{T}_0, \boldsymbol{T}_1, \boldsymbol{T}_{\mathrm{L}}, \boldsymbol{T}_{\mathrm{NL}}$ be as in Definitions 1 and C.2.1. Let $c_1, L, \boldsymbol{f}'$ be as in §C.1.*

*(i) $\|\boldsymbol{A}(n\tau\boldsymbol{I}_d + \boldsymbol{A}^\top\boldsymbol{A})^{-1}\|_{op} \leq 1/(2\sqrt{n\tau})$.*
*(ii) $\|\boldsymbol{I}_n - \boldsymbol{H}\|_{op} \leq 1$.*
*(iii) $\sum_{j\in[p]} \|\boldsymbol{T}_0(\boldsymbol{e}_j) + \boldsymbol{T}_{\mathrm{L}}(\boldsymbol{e}_j) + \boldsymbol{T}_{\mathrm{NL}}(\boldsymbol{e}_j)\|_F^2 \leq 2c_1^2 L^2 n\|\widehat{\boldsymbol{\alpha}}\|_2^2 + 2\|\boldsymbol{f}'\|_F^2$.*
*(iv) $\|\boldsymbol{T}_1(\boldsymbol{e}_j)\|_F^2 \leq L^2 c_1^2/(4n\tau) \cdot \|\boldsymbol{y} - \boldsymbol{A}\widehat{\boldsymbol{\alpha}}\|_2^2$.*
*(v) $\mathbb{E}\|\widehat{\boldsymbol{\alpha}}\|_2^2 = O(1)$.*

Proposition C.2.3 shows that the training error $\|\boldsymbol{y} - \boldsymbol{A}\widehat{\boldsymbol{\alpha}}\|_2^2$ is of order at least $n$ with large probability.

**Proposition C.2.3.** *There exists a large event $\Omega$ such that $\mathbb{P}(\Omega^c) \leq o(\exp(-c_6 n))$ for some $c_6 > 0$ and that on $\Omega$,*

*(i) $\|\boldsymbol{X}\|_F^2/n^2 \leq c_{4,n}$,*
*(ii) $1/n \cdot \min_{j\in[p]} \mathbb{E}_j\|\boldsymbol{y} - \boldsymbol{A}\widehat{\boldsymbol{\alpha}}\|_2^2 \geq c_{2,n}$,*
*(iii) $1/n \cdot \|\boldsymbol{y} - \boldsymbol{A}\widehat{\boldsymbol{\alpha}}\|_2^2 \geq c_{2,n}$.*

*The constant $c_6$ is independent of $n, d, p$. The constants $c_{4,n}$ and $c_{2,n}$ are given in Proposition C.2.4.*

**Proposition C.2.4.** *Let*

*(1) $c_{3,n} := \psi_{p,n}^{-1/2} + 1 + c^{-1/2} \to c_3 := \psi_p^{-1/2} + 1 + c^{-1/2}$ where $c > 0$ is some universal constant specified in the proof.*
*(2) $c_{4,n} := \psi_{p,n} + 2\psi_{p,n}^{1/2} + 2 \to c_4 := \psi_p + 2\psi_p^{1/2} + 2$.*
*(3) $F_n := 2c_1^2 L^2\|\boldsymbol{X}\|_F^2/n^2 + 2\psi_{d,n}(\sigma(0))^2$.*
*(4) $\overline{F}_n := 2c_1^2 L^2(n + \|\boldsymbol{X}\|_F^2)/n^2 + 2\psi_{d,n}(\sigma(0))^2$.*

(5) $\bar{c}_{2,n} := (1 + 2c_1^2 L^2 n^{-1}\tau^{-1} + 2c_1^2 L^2 c_{4,n}\tau^{-1} + 2\psi_{d,n}(\sigma(0))^2\tau^{-1})^{-1} \to \bar{c}_2 := (1 + 2c_1^2 L^2 c_4\tau^{-1} + 2\psi_d(\sigma(0))^2\tau^{-1})^{-1} > 0$.

(6) $c_{2,n} := \theta_{\boldsymbol{\varepsilon}}^2 (2\bar{c}_{2,n}/3 - n^{-1/2})_+^2 \to c_2 := \theta_{\boldsymbol{\varepsilon}}^2(2\bar{c}_2/3)^2 > 0$.

*Then*

(i) $\mathbb{P}\left(\left\|\boldsymbol{X}/\sqrt{p}\right\|_{op} \geq c_{3,n}\right) \leq 2\exp(-p)$.

(ii) $\mathbb{P}(\|\boldsymbol{X}\|_F^2/n^2 > c_{4,n}) \leq \exp(-n^2)$.

(iii) $\|\sigma(\boldsymbol{X}\boldsymbol{W}^\top)\|_F^2/n^2 \leq F_n$.

(iv) $\|\boldsymbol{I}_n - \boldsymbol{H}\|_F^2/n \geq (1 + F_n/\tau)^{-2}$.

(v) $\min_{j\in[p]}\left(n^{-1/2}\mathbb{E}_j\|\boldsymbol{I}_n - \boldsymbol{H}\|_F\right) \geq (1 + \overline{F}_n/\tau)^{-1}$.

Proposition C.2.5 helps us prove the asymptotic normality of $\zeta(\boldsymbol{e}_j)$ and helps us estimate the variance term in the asymptotic normality result with the training error: Propositions C.2.5(i) and C.2.5(iii) imply the asymptotic normality of $\zeta(\boldsymbol{e}_j)$ for most $j \in [p]$; Propositions C.2.5(ii) and C.2.5(iii) imply that we can estimate $\mathrm{Var}_j(\zeta(\boldsymbol{e}_j))$ using $\|\boldsymbol{y} - \boldsymbol{A}\widehat{\boldsymbol{\alpha}}\|_2^2$ for most $j \in [p]$.

**Proposition C.2.5.** *Let*

(i) $\zeta(\boldsymbol{e}_j)$ *be as in Definition C.2.1.*

(ii) $\boldsymbol{r} = \boldsymbol{y} - \boldsymbol{A}\widehat{\boldsymbol{\alpha}}$.

(iii) $\epsilon_j^2 := \mathbb{E}_j\left[\|\nabla_{\boldsymbol{X}_j}\boldsymbol{r}\|_F^2\right]/\left(\mathbb{E}_j\left[\|\boldsymbol{r}\|_2^2\right] + \mathbb{E}_j\left[\|\nabla_{\boldsymbol{X}_j}\boldsymbol{r}\|_F^2\right]\right)$.

*Then*

(i) $\mathbb{E}_j\left[\left(\frac{\zeta(\boldsymbol{e}_j)}{(\mathrm{Var}_j\zeta(\boldsymbol{e}_j))^{1/2}} - \frac{\boldsymbol{X}_j^\top\mathbb{E}_j[\boldsymbol{r}]}{\|\mathbb{E}_j[\boldsymbol{r}]\|_2}\right)^2\right] \leq 6\epsilon_j^2$.

(ii) $\mathbb{E}_j\left[\left|\frac{\|\boldsymbol{r}\|_2}{(\mathrm{Var}_j[\zeta(\boldsymbol{e}_j)])^{1/2}} - 1\right|\right] \leq \left(1 + \sqrt{2}\right)\epsilon_j/(1 - 2\epsilon_j^2)_+^{1/2}$.

(iii) $\mathbb{E}\left[\sum_{j\in[p]}\epsilon_j^2\right] = O(1)$.

Proposition C.2.6 shows that the nonlinear component $\mathrm{trace}(\boldsymbol{T}_{\mathrm{NL}}(\boldsymbol{e}_j))/\|\boldsymbol{y} - \boldsymbol{A}\widehat{\boldsymbol{\alpha}}\|_2$ of the asymptotic normality quantity $\zeta(\boldsymbol{e}_j)/\|\boldsymbol{y} - \boldsymbol{A}\widehat{\boldsymbol{\alpha}}\|_2$ is negligible. Since $\zeta_{\mathrm{L}}(\boldsymbol{e}_j) = \zeta(\boldsymbol{e}_j) + \mathrm{trace}\left[\boldsymbol{T}_{\mathrm{NL}}(\boldsymbol{e}_j)\right]$, we will then have the asymptotic normality of $\zeta_{\mathrm{L}}(\boldsymbol{e}_j)/\|\boldsymbol{y} - \boldsymbol{A}\widehat{\boldsymbol{\alpha}}\|_2$.

**Proposition C.2.6.** *Under Assumptions 1, 2, 3 and 4, we have the following convergence in probability: for all $\epsilon > 0$,*

$$(\mathrm{C.5}) \qquad \lim_{n\to+\infty}\mathbb{P}\left(\max_{j\in[p]}\left|\frac{\mathrm{trace}\left((\boldsymbol{I}_n - \boldsymbol{H})\mathrm{diag}(\boldsymbol{G}'\boldsymbol{e}_j)\right)}{\|\boldsymbol{y} - \boldsymbol{A}\widehat{\boldsymbol{\alpha}}\|_2}\right| > \epsilon\right) = 0.$$

Proposition C.2.7 characterizes the expected value of the derivatives of the nonlinear perturbation function $G$ in terms of $\Sigma_p'$ and $\Sigma_p''$.

**Proposition C.2.7.** *For all vector $\boldsymbol{v}_1, \boldsymbol{v}_2 \in \mathbb{R}^p$ such that the derivatives below exist,*

$$\mathbb{E}_G\left[\partial_{j_1}G(\boldsymbol{v}_1)\partial_{j_2}G(\boldsymbol{v}_2)\right] = \Sigma_p''(\boldsymbol{v}_1^\top\boldsymbol{v}_2/p)v_{1j_2}v_{2j_1}/p^2 + \Sigma_p'(\boldsymbol{v}_1^\top\boldsymbol{v}_2/p)\delta(j_1 = j_2)/p.$$

**C.3. The limits $V(\psi_d, \psi_p, \lambda, \rho, \sigma)$ and $R(\psi_p, \lambda, \rho, \sigma)$.** In this section we present the results related to the limits $V(\psi_d, \psi_p, \lambda, \rho, \sigma)$ and $R(\psi_p, \lambda, \rho, \sigma)$ in (18).

Assumption C.3.1 recalls the setting in Mei and Montanari [2019] that is comparable to our Gaussian setting.

**Assumption C.3.1.** *Let us assume that*

(i) $(\boldsymbol{x}_i)_{i\in[n]} \sim^{iid} \mathrm{Unif}(\mathbb{S}^{p-1}(\sqrt{p}))$.

(ii) $(\boldsymbol{w}_k)_{k\in[d]} \sim^{iid} \mathrm{Unif}(\mathbb{S}^{p-1}(1))$.

(iii) $G(\boldsymbol{x})$ *is a centered Gaussian process indexed by $\boldsymbol{x} \in \mathbb{S}^{p-1}(\sqrt{p})$, such that*

$\quad$ (i) $\mathbb{E}_G[G(\boldsymbol{x})] = 0$ *and* $\mathbb{E}_G[G(\boldsymbol{x}_1)G(\boldsymbol{x}_2)] = \Sigma_p(\boldsymbol{x}_1^\top\boldsymbol{x}_2/p)$,

(ii) $\mathbb{E}_{\boldsymbol{x}\sim\text{Unif}(\mathbb{S}^{p-1}(\sqrt{p}))}\left[\Sigma_p(x_1/\sqrt{p})\right]=0$, $\mathbb{E}_{\boldsymbol{x}\sim\text{Unif}(\mathbb{S}^{p-1}(\sqrt{p}))}\left[\Sigma_p(x_1/\sqrt{p})x_1\right]=0$ *and* $\lim_{p\to+\infty}\Sigma_p(1)=\theta_{\text{NL}}^2$.

(iv) $\mathbb{E}\left[\varepsilon_1\right]=0$, $\mathbb{E}\left[\varepsilon_1^2\right]=\theta_{\boldsymbol{\varepsilon}}^2$, $\mathbb{E}\left[\varepsilon_1^4\right]<+\infty$.

(v) $\beta_0\to\theta_0$, $\|\boldsymbol{\beta}\|_2^2\to\theta_{\boldsymbol{\beta}}^2$ *and* $\Sigma_p(1)\to\theta_{\text{NL}}^2$. *The signal-to-noise ratio* $\rho=\theta_{\boldsymbol{\beta}}^2/(\theta_{\boldsymbol{\varepsilon}}^2+\theta_{\text{NL}}^2)$.

Proposition C.3.1 provides the limiting squared length of our confidence intervals under Assumption C.3.1.

**Proposition C.3.1.** *Let* $L^2:=\|\boldsymbol{y}-\boldsymbol{A}\widehat{\boldsymbol{\alpha}}\|_2^2/(\text{trace}(\boldsymbol{I}_n-\boldsymbol{H}))^2$. *Under the model* (5) *in the asymptotic setting* (6)*, Assumption C.3.1 and Definition C.3.1,*

$$\text{Var}(y_1)\to\theta_{\boldsymbol{\beta}}^2+\theta_{\boldsymbol{\varepsilon}}^2+\theta_{\text{NL}}^2, \qquad nL^2/\text{Var}(y_1)\overset{\mathbb{P}_{\boldsymbol{X},\boldsymbol{W},G,\boldsymbol{\varepsilon}}}{\longrightarrow} V.$$

Definition C.3.1 defines $V(\psi_d,\psi_p,\lambda,\rho,\sigma)$.

**Definition C.3.1** (Definition 2 in Mei and Montanari [2019]). *Let* $Z\sim N(0,1)$, $\rho=\theta_{\boldsymbol{\beta}}^2/(\theta_{\boldsymbol{\varepsilon}}^2+\theta_{\text{NL}}^2)$,

$$\mu_1=\mathbb{E}\left[\sigma(Z)\right], \qquad \mu_2=\mathbb{E}\left[(\sigma(Z))^2\right], \qquad \gamma_1=\mathbb{E}\left[\sigma'(Z)\right],$$
$$\mu_*^2=\mu_2-\mu_1^2-\gamma_1^2, \qquad \varrho^2=\gamma_1^2/\mu_*^2, \qquad \overline{\lambda}=\lambda/\mu_*^2,$$
$$\psi_1=\psi_d/\psi_p, \qquad \psi_2=1/\psi_p, \qquad \bar{z}=\boldsymbol{i}(\psi_1\psi_2\lambda)^{1/2}/\mu_*.$$

*Let* $\Im(z)$ *be the imaginary part of complex number* $z$. *Let* $\mathbb{C}_+$ *be the set of complex numbers with positive imaginary parts. Let* $\nu_1,\nu_2\in\mathbb{C}_+$ *solves uniquely*

$$(\text{C.6}) \qquad \begin{aligned} \nu_1&=\psi_1\left(-\bar{z}-\nu_2-\frac{\varrho^2\nu_2}{1-\varrho^2\nu_1\nu_2}\right)^{-1}, \\ \nu_2&=\psi_2\left(-\bar{z}-\nu_1-\frac{\varrho^2\nu_1}{1-\varrho^2\nu_1\nu_2}\right)^{-1}, \end{aligned}$$

*under constraint* $|\nu_1|\le\psi_1/\Im(\bar{z})$ *and* $|\nu_2|\le\psi_2/\Im(\bar{z})$. *Let*

$$\chi=\nu_1\nu_2,$$
$$\mathscr{Q}=1+\psi_2^{-1}\left(\chi+\frac{\chi\varrho^2}{1-\chi\varrho^2}\right),$$
$$\mathscr{L}=\mathscr{Q}\cdot\left[\frac{\rho}{1+\rho}\cdot\frac{1}{1-\chi\varrho^2}+\frac{1}{1+\rho}\right],$$
$$\mathscr{A}_1=\frac{\rho}{1+\rho}\left[-\chi^2\left(\chi\varrho^4-\chi\varrho^2+\psi_2\varrho^2+\varrho^2-\chi\psi_2\varrho^4+1\right)\right]$$
$$\qquad+\frac{1}{1+\rho}\left[\chi^2\left(\chi\varrho^2-1\right)\left(\chi^2\varrho^4-2\chi\varrho^2+\varrho^2+1\right)\right],$$
$$\mathscr{A}_0=-\chi^5\varrho^6+3\chi^4\varrho^4+(\psi_1\psi_2-\psi_2-\psi_1+1)\chi^3\varrho^6-2\chi^3\varrho^4-3\chi^3\varrho^2,$$
$$\qquad+(\psi_1+\psi_2-3\psi_1\psi_2+1)\chi^2\varrho^4+2\chi^2\varrho^2+\chi^2+3\psi_1\psi_2\chi\varrho^2-\psi_1\psi_2,$$
$$\mathscr{A}=\mathscr{A}_1/\mathscr{A}_0,$$

$$V(\psi_d,\psi_p,\lambda,\rho,\sigma)=(\mathscr{L}-\psi_1\overline{\lambda}\mathscr{A})/\mathscr{Q}^2.$$

Random vectors uniformly distributed on the sphere $\mathbb{S}^{p-1}(\sqrt{p})$ are close to standard normal vectors (e.g., in the sense of § 3.3.3 in Vershynin [2018]). We expect the limit in Proposition C.3.1 to be valid for Gaussian $\boldsymbol{x}_i,\boldsymbol{w}_k$ as well. Proposition C.3.2 guarantees that the limit $V$ is valid for Gaussian vectors $(\boldsymbol{x}_i)_{i\in[n]}\sim^{iid}N(\boldsymbol{0}_p,\boldsymbol{I}_p)$, $(\boldsymbol{w}_k)_{k\in[d]}\sim^{iid}N(\boldsymbol{0}_p,(1/p)\boldsymbol{I}_p)$ for the pure linear model (4).

**Proposition C.3.2.** *Under the pure linear model* (4) *in the asymptotic setting* (6), *Assumption* 1 *and* 2 *with* (ii) *and Definition* C.3.1,

$$\mathrm{Var}(y_1) \to \theta_{\boldsymbol{\beta}}^2 + \theta_{\boldsymbol{\varepsilon}}^2, \qquad nL^2/\mathrm{Var}(y_1) \stackrel{\mathbb{P}_{\boldsymbol{X},\boldsymbol{W},\boldsymbol{\varepsilon}}}{\longrightarrow} V.$$

Proposition C.3.3 provides the $\psi_d \to +\infty$ limit of $V$.

**Proposition C.3.3.** *Let* $V, R$ *be in Definitions* C.3.1 *and* C.3.2. *Then*

$$\lim_{\psi_d \to +\infty} V(\psi_d, \psi_p, \lambda, \rho, \sigma) = R(\psi_p, \lambda, \rho, \sigma).$$

**Definition C.3.2.** *Let* $\varrho, \rho, \psi_1, \psi_2, \overline{\lambda}$ *be as in Definition* C.3.1. *Let*

$$R(\psi_p, \lambda, \rho, \sigma) = \begin{cases} 1 & \varrho = 0 \\ (\overline{\mathscr{L}} - \overline{\lambda \mathscr{A}_1 / \mathscr{A}_*}) / \overline{\mathscr{Q}}^2 & \varrho \neq 0 \end{cases},$$

*where*

$$\overline{\chi} = \begin{cases} -\dfrac{\psi_2}{1+\psi_2\overline{\lambda}} & \varrho = 0 \\[2ex] \dfrac{\left(\varrho^{-2} - \frac{\psi_2-1}{1+\psi_2\overline{\lambda}}\right) - \sqrt{\left(\varrho^{-2} - \frac{\psi_2-1}{1+\psi_2\overline{\lambda}}\right)^2 + 4\frac{\psi_2}{1+\psi_2\overline{\lambda}}\varrho^{-2}}}{2} & \varrho \neq 0 \end{cases},$$

$$\overline{\mathscr{Q}} = -\overline{\chi}\overline{\lambda},$$

$$\overline{\mathscr{L}} = \left(-\overline{\chi}\overline{\lambda}\right) \left[\frac{\rho}{1+\rho}\frac{1}{1-\overline{\chi}\varrho^2} + \frac{1}{1+\rho}\right],$$

$$\overline{\mathscr{A}_1} = \frac{\rho}{1+\rho}\left[-\overline{\chi}^2\left(\overline{\chi}\varrho^4 - \overline{\chi}\varrho^2 + \psi_2\varrho^2 + \varrho^2 - \overline{\chi}\psi_2\varrho^4 + 1\right)\right]$$

$$\qquad + \frac{1}{1+\rho}\left[\overline{\chi}^2\left(\overline{\chi}\varrho^2 - 1\right)\left(\overline{\chi}^2\varrho^4 - 2\overline{\chi}\varrho^2 + \varrho^2 + 1\right)\right],$$

$$\overline{\mathscr{A}_*} = (\psi_2 - 1)\overline{\chi}^3\varrho^6 + (1 - 3\psi_2)\overline{\chi}^2\varrho^4 + 3\psi_2\overline{\chi}\varrho^2 - \psi_2.$$

## C.4. Asymptotic normality result for a general direction.

In this section we consider a general $\boldsymbol{u}_0 \in \mathrm{Unif}\left(\mathbb{S}^{p-1}(1)\right)$ instead of a canonical basis $\boldsymbol{e}_j$. Defined in Definitions 1 and C.2.1, the functions $\boldsymbol{T}_0, \boldsymbol{T}_1, \boldsymbol{T}_{\mathrm{L}}, \boldsymbol{T}_{\mathrm{NL}}, \zeta_L$ and $\zeta$ are linear in $\boldsymbol{e}_j$. So we can naturally extend the functions for a general direction $\boldsymbol{u}_0 \in \mathrm{Unif}\left(\mathbb{S}^{p-1}(1)\right)$ by linear combinations, for e.g., $\zeta_{\mathrm{L}}(\boldsymbol{u}_0) = \boldsymbol{u}_0^\top(\zeta_{\mathrm{L}}(\boldsymbol{e}_j))^{j\in[p]}$. Extending the functions by linear combinations is equivalent to replacing $\boldsymbol{e}_j$ with $\boldsymbol{u}_0$ in the definitions of the functions.

Theorem C.4.1 provides the asymptotic normality of $\zeta_{\mathrm{L}}(\boldsymbol{u}_0)$ for a general $\boldsymbol{u}_0$ satisfying $\|\boldsymbol{u}_0\|_2 = 1$.

**Theorem C.4.1.** *Let* $t \in \mathbb{R}$. *Under model* (5), *Assumption* 1, 2, 3 *and* 4, *Definition* 1 *and a further assumption that* $\Sigma_p'(0) = O(1/p)$, *we have*

$$(\text{C.7}) \qquad \sup_{\boldsymbol{u}_0 \in S_p} \left|\mathbb{P}_{\boldsymbol{X},\boldsymbol{W},\boldsymbol{\varepsilon},G|\boldsymbol{u}_0}\left(\frac{\zeta_{\mathrm{L}}(\boldsymbol{u}_0)}{\|\boldsymbol{y} - \boldsymbol{A}\widehat{\boldsymbol{\alpha}}\|_2} \leq t\right) - \Phi(t)\right| \to 0$$

*for some* $S_p \subset \mathbb{S}^{p-1}(1)$ *satisfying* $|S_p|/|\mathbb{S}^{p-1}(1)| \geq 1 - \log(p)/p \to 1$.

The operations of taking expectations in this section are defined as follows:

(i) Let $\mathbb{E}_0, \mathrm{Var}_0$ denote the conditional expectation and the conditional variance given $\boldsymbol{X}\boldsymbol{Q}_0, \boldsymbol{\varepsilon}, \boldsymbol{W}, G, \boldsymbol{u}_0$.

(ii) Let $\mathbb{E}_{\boldsymbol{u}_0,\boldsymbol{X},\boldsymbol{W},\boldsymbol{\varepsilon},G}$ or $\mathbb{E}$ denote the expectation with respect to the total probability.

(iii) Let $\mathbb{E}_{\boldsymbol{X},\boldsymbol{W},\boldsymbol{\varepsilon},G|\boldsymbol{u}_0}$ or $\mathbb{E}_{\boldsymbol{X},\boldsymbol{W},\boldsymbol{\varepsilon},G}$ denote the conditional expectation given $\boldsymbol{u}_0$.

(iv) Let $\mathbb{E}_{\boldsymbol{u}_0}$ denote the conditional expectation given $\boldsymbol{X}, \boldsymbol{W}, \boldsymbol{\varepsilon}, G$.

Proposition C.4.1 is comparable to Proposition C.2.5 but for a general direction $\boldsymbol{u}_0$.

**Proposition C.4.1.** *Let*

(i) $\boldsymbol{u}_0 \sim \mathrm{Unif}\left(\mathbb{S}^{p-1}(1)\right)$ *independent with* $\boldsymbol{X}, \boldsymbol{W}, G, \boldsymbol{\varepsilon}$.

(ii) $\boldsymbol{X}_0 := \boldsymbol{X}\boldsymbol{u}_0$.

(iii) $\zeta(\boldsymbol{u}_0) := \boldsymbol{u}_0^\top \left(\zeta(\boldsymbol{e}_j)\right)^{j\in[p]}$ where $\zeta(\boldsymbol{e}_j)$ is defined in Definition C.2.1.

(iv) $\boldsymbol{r} = \boldsymbol{y} - \boldsymbol{A}\widehat{\boldsymbol{\alpha}}$.

(v) $\epsilon_0^2 := \mathbb{E}_0\left[\|\nabla_{\boldsymbol{X}_0}\boldsymbol{r}\|_F^2\right] / \left(\mathbb{E}_0\left[\|\boldsymbol{r}\|_2^2\right] + \mathbb{E}_0\left[\|\nabla_{\boldsymbol{X}_0}\boldsymbol{r}\|_F^2\right]\right)$.

*Then*

(i) $\mathbb{E}_0\left[\left(\frac{\zeta(\boldsymbol{u}_0)}{(\mathrm{Var}_0\zeta(\boldsymbol{u}_0))^{1/2}} - \frac{\boldsymbol{X}_0^\top \mathbb{E}_0[\boldsymbol{r}]}{\|\mathbb{E}_0[\boldsymbol{r}]\|_2}\right)^2\right] \le 6\epsilon_0^2$.

(ii) $\mathbb{E}_0\left[\left|\frac{\|\boldsymbol{r}\|_2}{(\mathrm{Var}_0[\zeta(\boldsymbol{u}_0)])^{1/2}} - 1\right|\right] \le \left(1 + \sqrt{2}\right)\epsilon_0/(1 - 2\epsilon_0^2)_+^{1/2}$.

(iii) $\mathbb{E}_{\boldsymbol{u}_0,\boldsymbol{X},\boldsymbol{W},\boldsymbol{\varepsilon},G}\left[\epsilon_0^2\right] = O(1/p)$.

Proposition C.4.2 is comparable to Proposition C.2.6 but for a general direction $\boldsymbol{u}_0$.

**Proposition C.4.2.** *Let $\Omega$ be as in Proposition C.2.3. Let $\delta_{i,j} = 1$ if $i = j$, $0$ otherwise. Let $\overline{\Omega}_3 := \Omega \cap \left\{\max_{i_1,i_2\in[n]}|\boldsymbol{x}_{i_1}^T\boldsymbol{x}_{i_2}/p - \delta_{i_1,i_2}| < \delta\right\}$ where $\delta$ is a fixed positive defined in Assumption 4. Let $\boldsymbol{u}_0 \sim \mathrm{Unif}\left(\mathbb{S}^{p-1}(1)\right)$ be independent of $\boldsymbol{X},\boldsymbol{W},\boldsymbol{\varepsilon},G$. Under Assumptions 1, 2, 3 and 4 and a further assumption that $\Sigma_p'(0) = O(1/p)$, we have that*

(C.8)
$$\mathbb{E}_{\boldsymbol{u}_0,\boldsymbol{X},\boldsymbol{W},\boldsymbol{\varepsilon},G}\left[\left(\frac{\mathrm{trace}\left((\boldsymbol{I}_n - \boldsymbol{H})\,\mathrm{diag}(\boldsymbol{G}'\boldsymbol{u}_0)\right)}{\|\boldsymbol{y} - \boldsymbol{A}\widehat{\boldsymbol{\alpha}}\|_2}\right)^2 I_{\Omega_3}\right] = O(1/p).$$

## APPENDIX D. PROOFS

In this section we provide the proofs of our theorems and the supporting propositions.

D.1. **Proofs of Theorem 1 and 2.** Theorem 1 is a special case of Theorem 2 when there is no intercept nor perturbation. We prove in this section Theorem 2 based on supporting propositions.

D.1.1. *Proof of Theorem 2.*

*Proof.* Let $\zeta$ be as in Definition C.2.1. As we explain in the next paragraphs, Propositions C.2.5 and C.2.6 imply that, for a large subset $J_p \subset [p]$, for all $j \in J_p$ we have

(i) $\frac{\zeta(\boldsymbol{e}_j)}{(\mathrm{Var}_j[\zeta(\boldsymbol{e}_j)])^{1/2}} \xrightarrow{d} N(0,1)$.

(ii) $\frac{\|\boldsymbol{y}-\boldsymbol{A}\widehat{\boldsymbol{\alpha}}\|_2}{(\mathrm{Var}_j[\zeta(\boldsymbol{e}_j)])^{1/2}} \xrightarrow{\mathbb{P}} 1$.

(iii) $\frac{\zeta(\boldsymbol{e}_j)-\zeta_{\mathrm{L}}(\boldsymbol{e}_j)}{\|\boldsymbol{y}-\boldsymbol{A}\widehat{\boldsymbol{\alpha}}\|_2} \xrightarrow{\mathbb{P}} 0$.

The above convergence are uniform over $j \in J_p$, where $J_p$ is a large subset of $[p]$. So by Slutsky's Theorem we obtain the convervence in distribution of $\zeta(\boldsymbol{e}_j)/\|\boldsymbol{y} - \boldsymbol{A}\widehat{\boldsymbol{\alpha}}\|_2$ to $N(0,1)$ and the convergence is uniform over all $j \in J_p$.

We first specify the subset $J_p \subset [p]$. Notice that (iii) in Proposition C.2.5 provides the existence of a constant $c_{11} > 0$ independent of $n, p, d$ such that $\sum_{j\in[p]} \mathbb{E}\left[\epsilon_j^2\right] \le c_{11}$. We can specify the large volume index set $J_p \subset [p]$ as

$$J_p := \left\{j \in [p] : \mathbb{E}\left[\epsilon_j^2\right] \le \frac{c_{11}}{\log(p)}\right\}.$$

Since

$$\frac{1}{p}\#\left\{j \in [p] : \mathbb{E}\left[\epsilon_j^2\right] \ge \frac{c_{11}}{\log(p)}\right\} \le \frac{\frac{1}{p}\sum_{j\in[p]}\mathbb{E}\left[\epsilon_j^2\right]}{c_{11}/\log(p)} \le \frac{\log(p)}{p},$$

we have $|J_p|/p \ge 1 - \frac{\log(p)}{p}$.

We claim that (i) is provided by Proposition C.2.5 and (iii) is provided by Proposition C.2.6 directly. Now let us explain (ii) rigorously based on Proposition C.2.5 (ii). From the definition of $J_p$ and Chebyshev's inequality, we can see that for all $\bar{\epsilon} > 0$,

$$\max_{j \in J_p} \mathbb{P}(|\epsilon_j| > \bar{\epsilon}) \leq \frac{c_{11}}{\bar{\epsilon}^2 \log(p)}.$$

Let $\boldsymbol{r} = \boldsymbol{A}\widehat{\boldsymbol{\alpha}} - \boldsymbol{y}$, $V_j := \mathrm{Var}_j\left[\zeta(\boldsymbol{e}_j)\right]$ and $U_j := \left|\|\boldsymbol{r}\|_2 / V_j^{1/2} - 1\right|$. If $\epsilon_j \leq \frac{1}{2}$, by Proposition C.2.5 and simple algebra,

$$\mathbb{E}_j\left[U_j\right] \leq \left(1 + \sqrt{2}\right)\epsilon_j / (1 - 2\epsilon_j^2)_+^{1/2} \leq (2 + \sqrt{2})\epsilon_j.$$

Let us consider $\bar{\epsilon} \leq \frac{1}{2}$. We let $\Omega_j(\bar{\epsilon}) := \left\{\mathbb{E}_j\left[U_j\right] < (2 + \sqrt{2})\bar{\epsilon}\right\}$. Then

$$\mathbb{P}(\Omega_j(\bar{\epsilon})) \geq \mathbb{P}(|\epsilon_j| \leq \bar{\epsilon}) \geq 1 - \frac{c_{11}}{\bar{\epsilon}^2 \log(p)}.$$

Then, letting $\mathbb{I}(\cdot) := I_{\{\cdot\}}$ be the indicator function, we have

$$\mathbb{P}(U_j > \epsilon) := \mathbb{E}\left[\mathbb{I}(U_j > \epsilon)\right]$$

$$= \mathbb{E}\left[\mathbb{E}_j\left[\mathbb{I}(U_j > \epsilon)\right]I_{\Omega_j(\bar{\epsilon})}\right] + \mathbb{E}\left[\mathbb{E}_j\left[\mathbb{I}(U_j > \epsilon)\right]I_{\Omega_j^c(\bar{\epsilon})}\right]$$

$$\leq \mathbb{E}\left[\frac{\mathbb{E}_j\left[U_j\right]}{\epsilon}I_{\Omega_j(\bar{\epsilon})}\right] + \mathbb{P}(\Omega_j^c(\bar{\epsilon}))$$

$$\leq \left(2 + \sqrt{2}\right)\bar{\epsilon}/\epsilon + \frac{c_{11}}{\bar{\epsilon}^2 \log(p)}.$$

Choosing $\bar{\epsilon} := \min\left(\frac{1}{\log\log(p)}, \frac{1}{2}\right)$, we have that, for all $\epsilon > 0$,

$$\lim_{p \to +\infty} \max_{j \in J_p} \mathbb{P}(U_j > \epsilon) = 0.$$

Thus we have (ii). $\qquad\square$

D.1.2. *Proof of Proposition C.2.1*. Propositions are restated before their proofs for convenience.

**Proposition C.2.1** (Calculation of the Jacobian matrix)**.** *Under model* (5),

$$\nabla_{\boldsymbol{X}_j}(\boldsymbol{y} - \boldsymbol{A}\widehat{\boldsymbol{\alpha}}) = -(\boldsymbol{I}_n - \boldsymbol{H})\mathrm{diag}(\sigma'(\boldsymbol{X}\boldsymbol{W}^\top)\mathrm{diag}(\boldsymbol{W}\boldsymbol{e}_j)\widehat{\boldsymbol{\alpha}})$$

(C.1)
$$- \boldsymbol{A}(\boldsymbol{A}^\top\boldsymbol{A} + n\tau\boldsymbol{I}_n)^{-1}\mathrm{diag}(\boldsymbol{W}\boldsymbol{e}_j)\sigma'(\boldsymbol{W}\boldsymbol{X}^\top)\mathrm{diag}(\boldsymbol{y} - \boldsymbol{A}\widehat{\boldsymbol{\alpha}})$$

$$+ (\boldsymbol{I}_n - \boldsymbol{H})\left[(\boldsymbol{e}_j^\top\boldsymbol{\beta})\boldsymbol{I}_n + \mathrm{diag}(\boldsymbol{G}'\boldsymbol{e}_j)\right].$$

*Proof.* The calculation of $\nabla_{\boldsymbol{X}_j}\boldsymbol{y}$ follows directly by

(D.1)
$$\nabla_{\boldsymbol{X}_j}\boldsymbol{y} = \nabla_{\boldsymbol{X}_j}\boldsymbol{f} = \mathrm{diag}(\boldsymbol{f}'\boldsymbol{e}_j) = (\boldsymbol{e}_j^\top\boldsymbol{\beta})\boldsymbol{I}_n + \mathrm{diag}(\boldsymbol{G}'\boldsymbol{e}_j).$$

For the calculation of $\nabla_{\boldsymbol{X}_j}(\boldsymbol{y} - \boldsymbol{A}\widehat{\boldsymbol{\alpha}})$, we notice that by the KKT condition,

$$\boldsymbol{y} - \boldsymbol{A}\widehat{\boldsymbol{\alpha}} = (\boldsymbol{I}_n - \boldsymbol{H})\boldsymbol{y}.$$

Proposition C.2.1 follows by the following intermediate steps:

(i) $\nabla_{\boldsymbol{X}_j}\left[(\boldsymbol{I}_n - \boldsymbol{H})\boldsymbol{y}\right] = (-1)\left[\left((\partial_{x_{i_2 j}}\boldsymbol{H})\boldsymbol{y}\right)_{i_1}\right]^{i_1, i_2 \in [n]} + (\boldsymbol{I}_n - \boldsymbol{H})\nabla_{\boldsymbol{X}_j}\boldsymbol{y}.$

(ii)
$$\partial_{x_{i_2 j}}\boldsymbol{H} = (\boldsymbol{I}_n - \boldsymbol{H})(\partial_{x_{i_2 j}}\boldsymbol{A})(\boldsymbol{A}^\top\boldsymbol{A} + n\tau\boldsymbol{I}_n)^{-1}\boldsymbol{A}^\top$$
$$+ \boldsymbol{A}(\boldsymbol{A}^\top\boldsymbol{A} + n\tau\boldsymbol{I}_n)^{-1}(\partial_{x_{i_2 j}}\boldsymbol{A})^\top(\boldsymbol{I}_n - \boldsymbol{H}).$$

(iii) $\partial_{x_{i_2 j}}\boldsymbol{A} = \mathrm{diag}(\boldsymbol{e}_{i_2})\sigma'(\boldsymbol{X}\boldsymbol{W}^\top)\mathrm{diag}(\boldsymbol{W}\boldsymbol{e}_j).$

The above intermediate steps can be seen from the followings.

(i) By the chain rule for multiplication.

(ii) Notice that $\boldsymbol{H} := \boldsymbol{A}(\boldsymbol{A}^\top\boldsymbol{A} + n\tau\boldsymbol{I}_n)^{-1}\boldsymbol{A}$. For the inverse matrix, we use the fact that

$$\partial_{x_{i_2 j}}\left[(\boldsymbol{A}^\top\boldsymbol{A} + n\tau\boldsymbol{I}_n)^{-1}\right] = (-1)(\boldsymbol{A}^\top\boldsymbol{A} + n\tau\boldsymbol{I}_n)^{-1}(\partial_{x_{i_2 j}}(\boldsymbol{A}^\top\boldsymbol{A}))(\boldsymbol{A}^\top\boldsymbol{A} + n\tau\boldsymbol{I}_n)^{-1}.$$

(iii) We notice that $\boldsymbol{A} = \sigma(\boldsymbol{X}\boldsymbol{W}^\top)$. So that by the chain rule,

$$\partial_{x_{i_2 j}}\boldsymbol{A} = \left[\delta_{i=i_2}\sigma'(\boldsymbol{x}_i^\top\boldsymbol{w}_k)w_{kj}\right]^{i\in[n],k\in[d]} = \mathrm{diag}(\boldsymbol{e}_{i_2})\sigma'(\boldsymbol{X}\boldsymbol{W}^\top)\mathrm{diag}(\boldsymbol{W}\boldsymbol{e}_j).$$

Combining the intermediate steps above, noticing that

$$\left[\left((\boldsymbol{I}_n - \boldsymbol{H})\mathrm{diag}(\boldsymbol{e}_{i_2})\sigma'(\boldsymbol{X}\boldsymbol{W}^\top)\mathrm{diag}(\boldsymbol{W}\boldsymbol{e}_j)(\boldsymbol{A}^\top\boldsymbol{A} + n\tau\boldsymbol{I}_n)^{-1}\boldsymbol{A}^\top\boldsymbol{y}\right)_{i_1}\right]^{i_1,i_2\in[n]}$$
$$= (\boldsymbol{I}_n - \boldsymbol{H})\mathrm{diag}\left(\sigma'(\boldsymbol{X}\boldsymbol{W}^\top)\mathrm{diag}(\boldsymbol{W}\boldsymbol{e}_j)\widehat{\boldsymbol{\alpha}}\right),$$

and

$$\left[\left(\boldsymbol{A}(\boldsymbol{A}^\top\boldsymbol{A} + n\tau\boldsymbol{I}_n)^{-1}\mathrm{diag}(\boldsymbol{W}\boldsymbol{e}_j)\sigma'(\boldsymbol{W}\boldsymbol{X}^\top)\mathrm{diag}(\boldsymbol{e}_{i_2})(\boldsymbol{I}_n - \boldsymbol{H})\boldsymbol{y}\right)_{i_1}\right]^{i_1,i_2\in[n]}$$
$$= \boldsymbol{A}(\boldsymbol{A}^\top\boldsymbol{A} + n\tau\boldsymbol{I}_n)^{-1}\mathrm{diag}(\boldsymbol{W}\boldsymbol{e}_j)\sigma'(\boldsymbol{W}\boldsymbol{X}^\top)\mathrm{diag}(\boldsymbol{y} - \boldsymbol{A}\widehat{\boldsymbol{\alpha}}),$$

we have our calculation. □

D.1.3. *Proof of Proposition C.2.2.*

**Proposition C.2.2.** *Let $\boldsymbol{T}_0, \boldsymbol{T}_1, \boldsymbol{T}_\mathrm{L}, \boldsymbol{T}_\mathrm{NL}$ be as in Definitions 1 and C.2.1. Let $c_1, L, \boldsymbol{f}'$ be as in §C.1.*

   *(i)* $\|\boldsymbol{A}(n\tau\boldsymbol{I}_d + \boldsymbol{A}^\top\boldsymbol{A})^{-1}\|_{op} \leq 1/(2\sqrt{n\tau})$.
   *(ii)* $\|\boldsymbol{I}_n - \boldsymbol{H}\|_{op} \leq 1$.
   *(iii)* $\sum_{j\in[p]}\|\boldsymbol{T}_0(\boldsymbol{e}_j) + \boldsymbol{T}_\mathrm{L}(\boldsymbol{e}_j) + \boldsymbol{T}_\mathrm{NL}(\boldsymbol{e}_j)\|_F^2 \leq 2c_1^2 L^2 n\|\widehat{\boldsymbol{\alpha}}\|_2^2 + 2\|\boldsymbol{f}'\|_F^2$.
   *(iv)* $\|\boldsymbol{T}_1(\boldsymbol{e}_j)\|_F^2 \leq L^2 c_1^2/(4n\tau) \cdot \|\boldsymbol{y} - \boldsymbol{A}\widehat{\boldsymbol{\alpha}}\|_2^2$.
   *(v)* $\mathbb{E}\|\widehat{\boldsymbol{\alpha}}\|_2^2 = O(1)$.

*Proof.*      (i) Let $\boldsymbol{A} = \boldsymbol{U}\boldsymbol{\Sigma}\boldsymbol{V}^\top$ be the SVD of $\boldsymbol{A}$ where $\boldsymbol{\Sigma} \in \mathbb{R}^{n\times d}$ has diagonal elements the singular values $\sigma_l$ for $l \in [\min(n,d)]$. Then the singular values of $\boldsymbol{A}(n\tau\boldsymbol{I}_d + \boldsymbol{A}^\top\boldsymbol{A})^{-1}$ are either $\sigma_l/(n\tau + \sigma_l^2)$ or 0. We notice that

$$\sigma_l/(n\tau + \sigma_l^2) \leq 1/(2\sqrt{n\tau}).$$

So that the operator norm is no more than $1/(2\sqrt{n\tau})$.

(ii) Let $\boldsymbol{A} = \boldsymbol{U}\boldsymbol{\Sigma}\boldsymbol{V}^\top$ be the SVD of $\boldsymbol{A}$ where $\boldsymbol{\Sigma} \in \mathbb{R}^{n\times d}$ has diagonal elements the singular values $\sigma_l$ for $l \in [\min(n,d)]$. Then the singular values of $\boldsymbol{I}_n - \boldsymbol{H}$ are either $(n\tau)/(n\tau + \sigma_l^2)$ or 1, no more than 1.

(iii)

$$\sum_{j\in[p]}\|\boldsymbol{T}_0(\boldsymbol{e}_j) + \boldsymbol{T}_\mathrm{L}(\boldsymbol{e}_j) + \boldsymbol{T}_\mathrm{NL}(\boldsymbol{e}_j)\|_F^2 \leq \sum_{j\in[p]}\|\boldsymbol{I}_n - \boldsymbol{H}\|_{op}^2\left\|\left(\sigma'(\boldsymbol{X}\boldsymbol{W}^\top)\mathrm{diag}(\widehat{\boldsymbol{\alpha}})\boldsymbol{W} - \boldsymbol{f}'\right)\boldsymbol{e}_j\right\|_2^2$$
$$\leq \sum_{j\in[p]}\left\|\left(\sigma'(\boldsymbol{X}\boldsymbol{W}^\top)\mathrm{diag}(\widehat{\boldsymbol{\alpha}})\boldsymbol{W} - \boldsymbol{f}'\right)\boldsymbol{e}_j\right\|_2^2$$
$$= \left\|\sigma'(\boldsymbol{X}\boldsymbol{W}^\top)\mathrm{diag}(\widehat{\boldsymbol{\alpha}})\boldsymbol{W} - \boldsymbol{f}'\right\|_F^2$$
$$\leq 2\left\|\sigma'(\boldsymbol{X}\boldsymbol{W}^\top)\mathrm{diag}(\widehat{\boldsymbol{\alpha}})\boldsymbol{W}\right\|_F^2 + 2\left\|\boldsymbol{f}'\right\|_F^2.$$
$$\leq 2\left\|\boldsymbol{W}\right\|_{op}^2\left\|\sigma'(\boldsymbol{X}\boldsymbol{W}^\top)\mathrm{diag}(\widehat{\boldsymbol{\alpha}})\right\|_F^2 + 2\left\|\boldsymbol{f}'\right\|_F^2$$
$$\leq 2c_1^2 L^2 n\left\|\widehat{\boldsymbol{\alpha}}\right\|_2^2 + 2\left\|\boldsymbol{f}'\right\|_F^2.$$

We used $\|\boldsymbol{I}_n - \boldsymbol{H}\|_{\mathrm{op}} \leq 1$ in the above display.

(iv)

$$
\begin{aligned}
\|\boldsymbol{T}_1(\boldsymbol{e}_j)\|_F^2 &\leq \|\boldsymbol{A}(n\tau\boldsymbol{I}_d + \boldsymbol{A}^\top\boldsymbol{A})^{-1}\|_{\mathrm{op}}^2 \cdot \|\operatorname{diag}(\boldsymbol{W}\boldsymbol{e}_j)\sigma'(\boldsymbol{W}\boldsymbol{X}^\top)\operatorname{diag}(\boldsymbol{y} - \boldsymbol{A}\widehat{\boldsymbol{\alpha}})\|_F^2 \\
&\leq 1/(4n\tau) \cdot L^2 \cdot \|(\boldsymbol{W}\boldsymbol{e}_j)(\boldsymbol{y} - \boldsymbol{A}\widehat{\boldsymbol{\alpha}})^\top\|_F^2 \\
&= L^2/(4n\tau) \cdot \|\boldsymbol{W}\boldsymbol{e}_j\|_2^2\|\boldsymbol{y} - \boldsymbol{A}\widehat{\boldsymbol{\alpha}}\|_2^2 \\
&\leq L^2/(4n\tau) \cdot \|\boldsymbol{W}\|_{\mathrm{op}}^2\|\boldsymbol{y} - \boldsymbol{A}\widehat{\boldsymbol{\alpha}}\|_2^2. \\
&\leq L^2 c_1^2/(4n\tau) \cdot \|\boldsymbol{y} - \boldsymbol{A}\widehat{\boldsymbol{\alpha}}\|_2^2.
\end{aligned}
$$

We used $\left\|\boldsymbol{A}(n\tau\boldsymbol{I}_d + \boldsymbol{A}^\top\boldsymbol{A})^{-1}\right\|_{\mathrm{op}}^2 \leq 1/(4n\tau)$ in the above display.

(v) From the KKT condition, we have

$$
n\tau\|\widehat{\boldsymbol{\alpha}}\|_2^2 = (\boldsymbol{A}\widehat{\boldsymbol{\alpha}})^\top(\boldsymbol{y} - \boldsymbol{A}\widehat{\boldsymbol{\alpha}}) = \boldsymbol{y}^\top\boldsymbol{H}^\top(\boldsymbol{I}_n - \boldsymbol{H})\boldsymbol{y}.
$$

Let the singular values of $\boldsymbol{A}$ be $\sigma_l$ for $l \in [\min(n,d)]$, then the singular value of $\boldsymbol{H}^\top(\boldsymbol{I}_n - \boldsymbol{H})$ are $\frac{\sigma_l^2}{n\tau + \sigma_l^2} \cdot \frac{n\tau}{n\tau + \sigma_l^2} \in [0, 1/4]$. So that $\|\widehat{\boldsymbol{\alpha}}\|_2^2 \leq 1/(4n\tau) \cdot \|\boldsymbol{y}\|_2^2$. Taking expectation we have

$$
\begin{aligned}
\mathbb{E}\|\widehat{\boldsymbol{\alpha}}\|_2^2 &\leq 1/(4n\tau) \cdot \mathbb{E}\|\beta_0\boldsymbol{1}_n + \boldsymbol{X}\boldsymbol{\beta} + \boldsymbol{G} + \boldsymbol{\varepsilon}\|_2^2 \\
&= 1/(4\tau) \cdot \left(\beta_0^2 + \|\boldsymbol{\beta}\|_2^2 + \mathbb{E}\left[\Sigma_p(\|\boldsymbol{x}_1\|_2^2/p)\right] + \theta_\varepsilon^2\right) = O(1).
\end{aligned}
$$

$\square$

### D.1.4. Proof of Proposition C.2.3.

**Proposition C.2.3.** *There exists a large event $\Omega$ such that $\mathbb{P}(\Omega^c) \leq o(\exp(-c_6 n))$ for some $c_6 > 0$ and that on $\Omega$,*

(i) $\|\boldsymbol{X}\|_F^2/n^2 \leq c_{4,n}$,
(ii) $1/n \cdot \min_{j \in [p]} \mathbb{E}_j\|\boldsymbol{y} - \boldsymbol{A}\widehat{\boldsymbol{\alpha}}\|_2^2 \geq c_{2,n}$,
(iii) $1/n \cdot \|\boldsymbol{y} - \boldsymbol{A}\widehat{\boldsymbol{\alpha}}\|_2^2 \geq c_{2,n}$.

*The constant $c_6$ is independent of $n, d, p$. The constants $c_{4,n}$ and $c_{2,n}$ are given in Proposition C.2.4.*

*Proof.* We first notice that by the KKT condition for the ridge regression type estimator $\widehat{\boldsymbol{\alpha}}$, we have

$$
\mathbb{E}_{\boldsymbol{\varepsilon}}\|\boldsymbol{y} - \boldsymbol{A}\widehat{\boldsymbol{\alpha}}\|_2^2 = \mathbb{E}_{\boldsymbol{\varepsilon}}\|(\boldsymbol{I}_n - \boldsymbol{H})\boldsymbol{y}\|_2^2 \geq \mathbb{E}_{\boldsymbol{\varepsilon}}\|(\boldsymbol{I}_n - \boldsymbol{H})\boldsymbol{\varepsilon}\|_2^2 = \theta_\varepsilon^2\|\boldsymbol{I}_n - \boldsymbol{H}\|_F^2.
$$

Next we show that $\|\boldsymbol{y} - \boldsymbol{A}\widehat{\boldsymbol{\alpha}}\|$ is concentrated around $\mathbb{E}_{\boldsymbol{\varepsilon}}\|\boldsymbol{y} - \boldsymbol{A}\widehat{\boldsymbol{\alpha}}\|$: By KKT condition, $\nabla_{\boldsymbol{\varepsilon}}(\boldsymbol{y} - \boldsymbol{A}\widehat{\boldsymbol{\alpha}}) = \boldsymbol{I}_n - \boldsymbol{H}$. Since $\|\boldsymbol{I}_n - \boldsymbol{H}\|_{\mathrm{op}} \leq 1$, $\|\boldsymbol{y} - \boldsymbol{A}\widehat{\boldsymbol{\alpha}}\|$ is 1-Lipschitz in $\boldsymbol{\varepsilon}$ (see also Bellec and Tsybakov [2017] for general results of this kind). By the triangle inequality and the independence between $\boldsymbol{X}_j$ and $\boldsymbol{\varepsilon}$, if $u(\boldsymbol{\varepsilon}, \boldsymbol{X}_j)$ is a function that is 1-Lipschitz with respect to $\boldsymbol{\varepsilon}$ for every value of $\boldsymbol{X}_j$, then

$$
|\mathbb{E}_j u(\boldsymbol{\varepsilon}, \boldsymbol{X}_j) - \mathbb{E}_j u(\tilde{\boldsymbol{\varepsilon}}, \boldsymbol{X}_j)| \leq \mathbb{E}_j|u(\boldsymbol{\varepsilon}, \boldsymbol{X}_j) - u(\tilde{\boldsymbol{\varepsilon}}, \boldsymbol{X}_j)| \leq \|\boldsymbol{\varepsilon} - \tilde{\boldsymbol{\varepsilon}}\|_2\mathbb{E}_j[1] = \|\boldsymbol{\varepsilon} - \tilde{\boldsymbol{\varepsilon}}\|_2.
$$

This implies that $\mathbb{E}_j\|\boldsymbol{y} - \boldsymbol{A}\widehat{\boldsymbol{\alpha}}\|_2$ is also 1-Lipschitz in $\boldsymbol{\varepsilon}$. By the concentration inequality for Lipschitz functions of a standard normal random vector (See Theorem 5.2.2 in Vershynin [2018] or Theorem 5.5 in Boucheron et al. [2013]) applied to the mappings $\boldsymbol{\varepsilon} \mapsto \|\boldsymbol{y} - \boldsymbol{A}\widehat{\boldsymbol{\alpha}}\|_2$ and $\boldsymbol{\varepsilon} \mapsto \mathbb{E}_j\|\boldsymbol{y} - \boldsymbol{A}\widehat{\boldsymbol{\alpha}}\|_2$, we have that for some universal constant $c_5 > 0$ and for all $t > 0$,

$$
\mathbb{P}(\mathbb{E}_j\|\boldsymbol{y} - \boldsymbol{A}\widehat{\boldsymbol{\alpha}}\|_2 \geq \mathbb{E}_{\boldsymbol{\varepsilon}}\mathbb{E}_j\|\boldsymbol{y} - \boldsymbol{A}\widehat{\boldsymbol{\alpha}}\|_2 - \sqrt{n}\theta_\varepsilon t) \geq 1 - 2\exp(-c_5 nt^2),
$$

$$
\mathbb{P}(\|\boldsymbol{y} - \boldsymbol{A}\widehat{\boldsymbol{\alpha}}\|_2 \geq \mathbb{E}_{\boldsymbol{\varepsilon}}\|\boldsymbol{y} - \boldsymbol{A}\widehat{\boldsymbol{\alpha}}\|_2 - \sqrt{n}\theta_\varepsilon t) \geq 1 - 2\exp(-c_5 nt^2).
$$

By the union bound, the following event occurs with probability at least $1 - 2(p+1)\exp(-c_5 nt^2)$:

$$
\bigcap_{j \in [p]} \left\{ \mathbb{E}_j\|\boldsymbol{y} - \boldsymbol{A}\widehat{\boldsymbol{\alpha}}\|_2 \geq \mathbb{E}_{\boldsymbol{\varepsilon}}\mathbb{E}_j\|\boldsymbol{y} - \boldsymbol{A}\widehat{\boldsymbol{\alpha}}\|_2 - \sqrt{n}\theta_\varepsilon t, \qquad \|\boldsymbol{y} - \boldsymbol{A}\widehat{\boldsymbol{\alpha}}\|_2 \geq \mathbb{E}_{\boldsymbol{\varepsilon}}\|\boldsymbol{y} - \boldsymbol{A}\widehat{\boldsymbol{\alpha}}\|_2 - \sqrt{n}\theta_\varepsilon t \right\}.
$$

Since $\boldsymbol{X}_j, \boldsymbol{X}_{-j}, \boldsymbol{\varepsilon}$ are independent, we can exchange the order of expectation by Fubini's Theorem,

$$\mathbb{E}_{\boldsymbol{\varepsilon}}\mathbb{E}_j\|\boldsymbol{y} - \boldsymbol{A}\widehat{\boldsymbol{\alpha}}\|_2 - \sqrt{n}\theta_{\boldsymbol{\varepsilon}}t = \mathbb{E}_j\mathbb{E}_{\boldsymbol{\varepsilon}}\|\boldsymbol{y} - \boldsymbol{A}\widehat{\boldsymbol{\alpha}}\|_2 - \sqrt{n}\theta_{\boldsymbol{\varepsilon}}t,$$

$$= \mathbb{E}_j\left[\left(\mathbb{E}_{\boldsymbol{\varepsilon}}\|\boldsymbol{y} - \boldsymbol{A}\widehat{\boldsymbol{\alpha}}\|_2^2 - \operatorname{Var}_{\boldsymbol{\varepsilon}}\|\boldsymbol{y} - \boldsymbol{A}\widehat{\boldsymbol{\alpha}}\|_2\right)^{1/2}\right] - \sqrt{n}\theta_{\boldsymbol{\varepsilon}}t,$$

$$\geq \theta_{\boldsymbol{\varepsilon}}\mathbb{E}_j\left[\left(\|\boldsymbol{I}_n - \boldsymbol{H}\|_F^2 - 1\right)_+^{1/2}\right] - \sqrt{n}\theta_{\boldsymbol{\varepsilon}}t$$

$$\geq \theta_{\boldsymbol{\varepsilon}}\mathbb{E}_j\|\boldsymbol{I}_n - \boldsymbol{H}\|_F - (\theta_{\boldsymbol{\varepsilon}} + \sqrt{n}\theta_{\boldsymbol{\varepsilon}}t)$$

$$= \theta_{\boldsymbol{\varepsilon}}\left[\mathbb{E}_j\|\boldsymbol{I}_n - \boldsymbol{H}\|_F - \sqrt{n}t - 1\right].$$

The first inequality above is due to the fact that $\mathbb{E}_{\boldsymbol{\varepsilon}}\|\boldsymbol{y} - \boldsymbol{A}\widehat{\boldsymbol{\alpha}}\|_2^2 \geq \theta_{\boldsymbol{\varepsilon}}^2\|\boldsymbol{I}_n - \boldsymbol{H}\|_F^2$ and the fact that $\operatorname{Var}_{\boldsymbol{\varepsilon}}\|\boldsymbol{y} - \boldsymbol{A}\widehat{\boldsymbol{\alpha}}\|_2 \leq \theta_{\boldsymbol{\varepsilon}}^2$ by the Gaussian Poincaré Inequality [Boucheron et al., 2013, Theorem 3.20] with respect to the 1-Lipschitz mapping $\boldsymbol{\varepsilon} \mapsto \|\boldsymbol{y} - \boldsymbol{A}\widehat{\boldsymbol{\alpha}}\|_2$. The second last inequality above is due to the fact that for any two positive real number $a, b$, $(a^2 - b^2)_+^{1/2} \geq a - b$. By a similar argument,

$$\mathbb{E}_{\boldsymbol{\varepsilon}}\|\boldsymbol{y} - \boldsymbol{A}\widehat{\boldsymbol{\alpha}}\|_2 - \sqrt{n}\theta_{\boldsymbol{\varepsilon}}t \geq \theta_{\boldsymbol{\varepsilon}}[\|\boldsymbol{I}_n - \boldsymbol{H}\|_F - \sqrt{n}t - 1].$$

From Proposition C.2.4 we can have for all $t > 0$ and some universal constant $c_5 > 0$,

$$\mathbb{P}\left(1/n \cdot \min_{j \in [p]}\mathbb{E}_j\|\boldsymbol{y} - \boldsymbol{A}\widehat{\boldsymbol{\alpha}}\|_2^2 \geq \theta_{\boldsymbol{\varepsilon}}^2\left((1 + \overline{F}_n/\tau)^{-1} - t - 1/\sqrt{n}\right)_+^2\right) \geq 1 - 2p\exp(-c_5 nt^2).$$

$$\mathbb{P}\left(1/n \cdot \|\boldsymbol{y} - \boldsymbol{A}\widehat{\boldsymbol{\alpha}}\|_2^2 \geq \theta_{\boldsymbol{\varepsilon}}^2\left((1 + \overline{F}_n/\tau)^{-1} - t - 1/\sqrt{n}\right)_+^2\right) \geq 1 - \exp(-c_5 nt^2).$$

Consider the intersection of the above events with the event $\{\|\boldsymbol{X}\|_F^2/n^2 \leq c_{4,n}\}$. We notice that on that intersection,

$$\overline{F}_n \leq 2c_1^2 L^2/n + 2c_1^2 L^2 c_{4,n} + 2\psi_{d,n}(\sigma(0))^2$$

$$\leq 2c_1^2 L^2 c_4 + 2\psi_d(\sigma(0))^2 + o(1),$$

$$(1 + \overline{F}_n/\tau)^{-1} \geq \bar{c}_{2,n} = \bar{c}_2 + o(1).$$

Taking $t = \bar{c}_{2,n}/3$, we obtain

$$\mathbb{P}\left(1/n \cdot \min_{j \in [p]}\mathbb{E}_j\|\boldsymbol{y} - \boldsymbol{A}\widehat{\boldsymbol{\alpha}}\|_2^2 \geq \theta_{\boldsymbol{\varepsilon}}^2\left(2\bar{c}_{2,n}/3 - 1/\sqrt{n}\right)_+^2\right) \geq 1 - 2p\exp(-9^{-1}c_5\bar{c}_{2,n}^2 n)$$

$$- \exp(-n^2)$$

as well as $\quad \mathbb{P}\left(1/n \cdot \|\boldsymbol{y} - \boldsymbol{A}\widehat{\boldsymbol{\alpha}}\|_2^2 \geq \theta_{\boldsymbol{\varepsilon}}^2\left(2\bar{c}_{2,n}/3 - 1/\sqrt{n}\right)_+^2\right) \geq 1 - \exp(-9^{-1}c_5\bar{c}_{2,n}^2 n).$

$$- \exp(-n^2).$$

$\square$

### D.1.5. *Proof of Proposition C.2.4.*

**Proposition C.2.4.** *Let*

(1) $c_{3,n} := \psi_{p,n}^{-1/2} + 1 + c^{-1/2} \to c_3 := \psi_p^{-1/2} + 1 + c^{-1/2}$ *where $c > 0$ is some universal constant specified in the proof.*

(2) $c_{4,n} := \psi_{p,n} + 2\psi_{p,n}^{1/2} + 2 \to c_4 := \psi_p + 2\psi_p^{1/2} + 2.$

(3) $F_n := 2c_1^2 L^2\|\boldsymbol{X}\|_F^2/n^2 + 2\psi_{d,n}(\sigma(0))^2.$

(4) $\overline{F}_n := 2c_1^2 L^2(n + \|\boldsymbol{X}\|_F^2)/n^2 + 2\psi_{d,n}(\sigma(0))^2.$

(5) $\bar{c}_{2,n} := (1 + 2c_1^2 L^2 n^{-1}\tau^{-1} + 2c_1^2 L^2 c_{4,n}\tau^{-1} + 2\psi_{d,n}(\sigma(0))^2\tau^{-1})^{-1} \to \bar{c}_2 := (1 + 2c_1^2 L^2 c_4\tau^{-1} + 2\psi_d(\sigma(0))^2\tau^{-1})^{-1} > 0.$

(6) $c_{2,n} := \theta_{\boldsymbol{\varepsilon}}^2(2\bar{c}_{2,n}/3 - n^{-1/2})_+^2 \to c_2 := \theta_{\boldsymbol{\varepsilon}}^2(2\bar{c}_2/3)^2 > 0.$

*Then*

(i) $\mathbb{P}\left(\left\|\boldsymbol{X}/\sqrt{p}\right\|_{op} \geq c_{3,n}\right) \leq 2\exp(-p)$.

(ii) $\mathbb{P}(\|\boldsymbol{X}\|_F^2/n^2 > c_{4,n}) \leq \exp(-n^2)$.

(iii) $\|\sigma(\boldsymbol{X}\boldsymbol{W}^\top)\|_F^2/n^2 \leq F_n$.

(iv) $\|\boldsymbol{I}_n - \boldsymbol{H}\|_F^2/n \geq (1 + F_n/\tau)^{-2}$.

(v) $\min_{j \in [p]}\left(n^{-1/2}\mathbb{E}_j\|\boldsymbol{I}_n - \boldsymbol{H}\|_F\right) \geq (1 + \overline{F}_n/\tau)^{-1}$.

*Proof.* (i) Corollary 7.3.3 in Vershynin [2018] provides the high probability upper bound for the operator norm of random matrix,

$$\mathbb{P}(\|\boldsymbol{X}\|_{op} \geq \sqrt{n} + \sqrt{p} + t) \leq 2\exp(-ct^2)$$

for some universal constant $c > 0$. Taking $ct^2 = p$, we have

$$\mathbb{P}\left(\|\boldsymbol{X}/\sqrt{p}\|_{op} \geq \psi_{p,n}^{-1/2} + 1 + c^{-1/2}\right) \leq 2\exp(-p).$$

(ii) By Lemma 1 in Laurent and Massart [2000], we have the concentration for $\|\boldsymbol{X}\|_F^2$, the chi-square random-variable with degree of freedom $np$, as follows: For any $x > 0$,

$$\mathbb{P}(\|\boldsymbol{X}\|_F^2 - np \geq 2\sqrt{xnp} + 2x) \leq \exp(-x).$$

Here we take $x = n^2$ for simplicity of proof.

(iii) If $\sigma$ is $L$-Lipschitz, then

$$|\sigma(x)| \leq L|x| + |\sigma(0)|,$$
$$\implies (\sigma(x))^2 \leq 2L^2 x^2 + 2(\sigma(0))^2.$$

Taking $x = \boldsymbol{x}_i^\top \boldsymbol{w}_k$ and summing over $(i,k) \in [n] \times [d]$ we have

$$\|\sigma(\boldsymbol{X}\boldsymbol{W}^\top)\|_F^2/n^2 \leq 2L^2\|\boldsymbol{X}\boldsymbol{W}^\top\|_F^2/n^2 + 2\psi_{d,n}(\sigma(0))^2$$
$$\leq 2c_1^2 L^2\|\boldsymbol{X}\|_F^2/n^2 + 2\psi_{d,n}(\sigma(0))^2.$$

(iv) Let $\sigma_l$, $l \in [\min(n,d)]$ be the singular values of $\boldsymbol{A}$. If $n > d$, we define $\sigma_i = 0$ for $i \in (d,n]$. We notice that by the SVD of $\boldsymbol{H}$,

$$\|\boldsymbol{I}_n - \boldsymbol{H}\|_F^2 = \sum_{i \in [n]}\left(\frac{1}{1 + \sigma_i^2/(n\tau)}\right)^2$$

$$\geq n\left(\frac{1}{1 + \sum_{l \in \min(n,d)}\sigma_l^2/(n^2\tau)}\right)^2$$

$$= n\left(1 + \|\boldsymbol{A}\|_F^2/(n^2\tau)\right)^{-2}$$

where the inequality is due to Jensen's Inequality.

(v) This is by the convexity of $x \mapsto \frac{1}{1+x}$ on $\mathbb{R}^+$ and Jensen's Inequality, $\mathbb{E}_j(\frac{1}{1+X}) \geq \frac{1}{1+\mathbb{E}_j X}$ for any random variable $X$ supported on $\mathbb{R}^+$. We then notice that $\mathbb{E}_j[\|\boldsymbol{X}\|_F^2] \leq n + \|\boldsymbol{X}\|_F^2$. $\square$

### D.1.6. *Proof of Proposition C.2.5.*

**Proposition C.2.5.** *Let*

(i) $\zeta(\boldsymbol{e}_j)$ *be as in Definition C.2.1.*

(ii) $\boldsymbol{r} = \boldsymbol{y} - \boldsymbol{A}\widehat{\boldsymbol{\alpha}}$.

(iii) $\epsilon_j^2 := \mathbb{E}_j\left[\|\nabla_{\boldsymbol{X}_j}\boldsymbol{r}\|_F^2\right] / \left(\mathbb{E}_j\left[\|\boldsymbol{r}\|_2^2\right] + \mathbb{E}_j\left[\|\nabla_{\boldsymbol{X}_j}\boldsymbol{r}\|_F^2\right]\right)$.

*Then*

(i) $\mathbb{E}_j\left[\left(\frac{\zeta(\boldsymbol{e}_j)}{(\mathrm{Var}_j\zeta(\boldsymbol{e}_j))^{1/2}} - \frac{\boldsymbol{X}_j^\top \mathbb{E}_j[\boldsymbol{r}]}{\|\mathbb{E}_j[\boldsymbol{r}]\|_2}\right)^2\right] \leq 6\epsilon_j^2$.

(ii) $\mathbb{E}_j \left[ \left| \frac{\|\boldsymbol{r}\|_2}{(\operatorname{Var}_j[\zeta(\boldsymbol{e}_j)])^{1/2}} - 1 \right| \right] \le \left(1 + \sqrt{2}\right) \epsilon_j / (1 - 2\epsilon_j^2)_+^{1/2}.$

(iii) $\mathbb{E} \left[ \sum_{j \in [p]} \epsilon_j^2 \right] = O(1).$

*Proof.* (i) The first inequality follows directly from Theorem 2.1 in Bellec and Zhang [2019].

(ii) By Second Order Stein's lemma applied to the mapping $\boldsymbol{X}_j \mapsto \boldsymbol{r}$ (cf. Theorem 2.1 in Bellec and Zhang [2018]), we have

$$V := \operatorname{Var}_j \left[ \zeta(\boldsymbol{e}_j) \right] = \mathbb{E}_j \left[ \|\boldsymbol{r}\|_2^2 \right] + \mathbb{E}_j [\operatorname{trace}[(\nabla_{\boldsymbol{X}_j} \boldsymbol{r})^2]].$$

By Second Order Stein's lemma applied to the mapping $\boldsymbol{X}_j \mapsto \boldsymbol{r} - \mathbb{E}_j \left[ \boldsymbol{r} \right]$, we have

$$\begin{aligned}
\overline{V} &:= \operatorname{Var}_j \left( \boldsymbol{X}_j^\top (\boldsymbol{r} - \mathbb{E}_j \left[ \boldsymbol{r} \right]) - \operatorname{trace} \left[ \nabla_{\boldsymbol{X}_j} \boldsymbol{r} \right] \right) \\
&= \mathbb{E}_j \left[ \|\boldsymbol{r} - \mathbb{E}_j \boldsymbol{r}\|_2^2 + \operatorname{trace}[(\nabla_{\boldsymbol{X}_j} \boldsymbol{r})^2] \right] \\
&= \operatorname{Var}_j \left[ \zeta(\boldsymbol{e}_j) \right] - \|\mathbb{E}_j \left[ \boldsymbol{r} \right]\|_2^2.
\end{aligned}$$

By the fact that $2ab \in [-a^2 - b^2, a^2 + b^2]$ for two real $a$ and $b$,

$$\left| \mathbb{E}_j \left[ \operatorname{trace}[(\nabla_{\boldsymbol{X}_j} \boldsymbol{r})^2] \right] \right| \le \mathbb{E}_j \left[ \|\nabla_{\boldsymbol{X}_j} \boldsymbol{r}\|_F^2 \right].$$

By Gaussian Poincaré Inequality [Boucheron et al., 2013, Theorem 3.20] applied to the mapping $\boldsymbol{X}_j \mapsto r_i(\boldsymbol{X}_j)$ for each $i \in [n]$,

$$\mathbb{E}_j \left[ \|\boldsymbol{r} - \mathbb{E}_j \left[ \boldsymbol{r} \right]\|_2^2 \right] \le \mathbb{E}_j \left[ \|\nabla_{\boldsymbol{X}_j} \boldsymbol{r}\|_F^2 \right].$$

So that $\overline{V} \le 2\mathbb{E}_j \left[ \|\nabla_{\boldsymbol{X}_j} \boldsymbol{r}\|_F^2 \right].$ We also notice that

$$\begin{aligned}
\mathbb{E}_j \left[ \left| \frac{\|\boldsymbol{r}\|_2}{V^{1/2}} - 1 \right| \right] &= V^{-1/2} \mathbb{E}_j \left[ \left| \|\boldsymbol{r}\|_2 - V^{1/2} \right| \right] \\
&\le V^{-1/2} \mathbb{E}_j \left[ \left| \|\boldsymbol{r}\|_2 - \|\mathbb{E}_j \left[ \boldsymbol{r} \right]\|_2 \right| \right] + V^{-1/2} \left| \|\mathbb{E}_j \left[ \boldsymbol{r} \right]\|_2 - V^{1/2} \right|.
\end{aligned}$$

By Jensen's Inequality,

$$\begin{aligned}
\mathbb{E}_j \left[ \|\|\boldsymbol{r}\|_2 - \|\mathbb{E}_j \left[ \boldsymbol{r} \right]\|_2 \| \right] &\le \left( \mathbb{E}_j \left[ (\|\mathbb{E}_j \left[ \boldsymbol{r} \right]\| - \|\boldsymbol{r}\|)^2 \right] \right)^{1/2} \\
&\le \left( \mathbb{E}_j \left[ \|\boldsymbol{r} - \mathbb{E}_j \left[ \boldsymbol{r} \right]\|_2^2 \right] \right)^{1/2} \\
&\le \sqrt{\mathbb{E}_j [\|\nabla_{\boldsymbol{x}_j} \boldsymbol{r}\|_F^2]}.
\end{aligned}$$

The last inequality above follows by the Gaussian Poincaré Inequality [Boucheron et al., 2013, Theorem 3.20] applied to the mapping $\boldsymbol{X}_j \mapsto r_i(\boldsymbol{X}_j)$ for each $i \in [n]$. By $|a - b| \le \sqrt{a^2 + b^2}$ for two real $a, b > 0$,

$$\left| \|\mathbb{E}_j \left[ \boldsymbol{r} \right]\|_2 - V^{1/2} \right| \le \overline{V}^{1/2} \le \sqrt{2} \sqrt{\mathbb{E}_j [\|\nabla_{\boldsymbol{x}_j} \boldsymbol{r}\|_F^2]}.$$

So that

$$\begin{aligned}
\mathbb{E}_j \left[ \left| \frac{\|\boldsymbol{r}\|}{V_j^{1/2}} - 1 \right| \right] &\le \left(1 + \sqrt{2}\right) \left( \frac{\mathbb{E}_j \left[ \|\nabla_{\boldsymbol{X}_j} \boldsymbol{r}\|_F^2 \right]}{V} \right)^{1/2} \\
&\le \left(1 + \sqrt{2}\right) \left( \frac{\mathbb{E}_j [\|\nabla_{\boldsymbol{X}_j} \boldsymbol{r}\|_F^2]}{(\mathbb{E}_j [\|\boldsymbol{r}\|_2^2] - \mathbb{E}_j [\|\nabla_{\boldsymbol{X}_j} \boldsymbol{r}\|_F^2])_+} \right)^{1/2} \\
&\le \left(1 + \sqrt{2}\right) \epsilon_j / (1 - 2\epsilon_j^2)_+^{1/2},
\end{aligned}$$

where $\epsilon_j^2 = \mathbb{E}_j \|\nabla_{\boldsymbol{X}_j} \boldsymbol{r}\|_F^2 / \left( \mathbb{E}_j \|\boldsymbol{r}\|_2^2 + \mathbb{E}_j \|\nabla_{\boldsymbol{X}_j} \boldsymbol{r}\|_F^2 \right)$. In the above we let $a/0 = +\infty$ by convention.

(iii) We first recall a large probability event $\Omega$, defined in Proposition C.2.3. Since $1 = I_\Omega + I_{\Omega^c}$ and $\sum_{j\in[p]} \epsilon_j^2 \le p$,

$$\mathbb{E}\left[\sum_{j\in[p]} \epsilon_j^2\right] \le \mathbb{E}\left[\sum_{j\in[p]} \epsilon_j^2 I_\Omega\right] + o(p\exp(-c_6 n)).$$

Let $\boldsymbol{T}_0, \boldsymbol{T}_1, \boldsymbol{T}_{\mathrm{L}}, \boldsymbol{T}_{\mathrm{NL}}$ be as in Proposition C.2.1. Then $\nabla_{\boldsymbol{X}_j}\boldsymbol{r} = \boldsymbol{T}_0 + \boldsymbol{T}_1 + \boldsymbol{T}_{\mathrm{L}} + \boldsymbol{T}_{\mathrm{NL}}$,

$$\|\nabla_{\boldsymbol{X}_j}\boldsymbol{r}\|_F^2 \le 2\|\boldsymbol{T}_0 + \boldsymbol{T}_{\mathrm{L}} + \boldsymbol{T}_{\mathrm{NL}}\|_F^2 + 2\|\boldsymbol{T}_1\|_F^2.$$

We have

$$\mathbb{E}\left[\sum_{j\in[p]} \epsilon_j^2 I_\Omega\right] \le \mathbb{E}\left[\sum_{j\in[p]} \frac{\mathbb{E}_j\|\boldsymbol{T}_0 + \boldsymbol{T}_{\mathrm{L}} + \boldsymbol{T}_{\mathrm{NL}}\|_F^2 + \mathbb{E}_j\|\boldsymbol{T}_1\|_F^2}{\mathbb{E}_j\|\boldsymbol{r}\|_2^2 + \mathbb{E}_j\|\nabla_{\boldsymbol{X}_j}\boldsymbol{r}\|_F^2} I_\Omega\right]$$

By Proposition C.2.2 and Proposition C.2.3,

$$\mathbb{E}\left[\sum_{j\in[p]} \frac{\mathbb{E}_j\|\boldsymbol{T}_1(\boldsymbol{e}_j)\|_F^2}{\mathbb{E}_j\|\boldsymbol{r}\|_2^2 + \mathbb{E}_j\|\nabla_{\boldsymbol{X}_j}\boldsymbol{r}\|_F^2} I_\Omega\right] \le \mathbb{E}\left[\sum_{j\in[p]} \frac{L^2 c_1^2/(4n\tau)\cdot\mathbb{E}_j\|\boldsymbol{r}\|_2^2}{\mathbb{E}_j\|\boldsymbol{r}\|_2^2 + \mathbb{E}_j\|\nabla_{\boldsymbol{X}_j}\boldsymbol{r}\|_F^2}\right]$$
$$\le (1/4)L^2 c_1^2 \psi_{p,n}\tau^{-1},$$
$$= (1/4)L^2 c_1^2 \psi_p \tau^{-1} + o(1).$$

$$\mathbb{E}\left[\sum_{j\in[p]} \frac{\mathbb{E}_j\|\boldsymbol{T}_0(\boldsymbol{e}_j) + \boldsymbol{T}_{\mathrm{L}}(\boldsymbol{e}_j) + \boldsymbol{T}_{\mathrm{NL}}(\boldsymbol{e}_j)\|_F^2}{\mathbb{E}_j\|\boldsymbol{r}\|_2^2 + \mathbb{E}_j\|\nabla_{\boldsymbol{X}_j}\boldsymbol{r}\|_F^2} I_\Omega\right] \le 1/(c_{2,n}n)\cdot\mathbb{E}\left[\sum_{j\in[p]}\|\boldsymbol{T}_0 + \boldsymbol{T}_{\mathrm{L}} + \boldsymbol{T}_{\mathrm{NL}}\|_F^2\right]$$
$$\le 1/(c_{2,n}n)\cdot\left(2c_1^2 L^2 n\mathbb{E}\left[\|\widehat{\boldsymbol{\alpha}}\|_2^2\right] + 2\mathbb{E}\left[\|\boldsymbol{f}'\|_F^2\right]\right)$$
$$\le 2c_1^2 c_{2,n}^{-1} L^2 \mathbb{E}[\|\widehat{\boldsymbol{\alpha}}\|_2^2] + 2c_{2,n}^{-1}\mathbb{E}\left[\|\boldsymbol{\beta} + \nabla G(\boldsymbol{x}_1)\|_2^2\right].$$

Proposition C.2.2 and Assumptions 1 and 3 provide us $\mathbb{E}\left[\|\widehat{\boldsymbol{\alpha}}\|_2^2\right] = O(1)$ and $\mathbb{E}\left[\|\boldsymbol{\beta} + \nabla G(\boldsymbol{x}_1)\|_2^2\right] = O(1)$. Combining the above we have $\mathbb{E}\left[\sum_{j\in[p]}\epsilon_j^2\right] \le O(1)$.

□

### D.1.7. *Proof of Proposition C.2.6.*

**Proposition C.2.6.** *Under Assumptions 1, 2, 3 and 4, we have the following convergence in probability: for all $\epsilon > 0$,*

(C.5)
$$\lim_{n\to+\infty} \mathbb{P}\left(\max_{j\in[p]}\left|\frac{\operatorname{trace}\left((\boldsymbol{I}_n - \boldsymbol{H})\operatorname{diag}(\boldsymbol{G}'\boldsymbol{e}_j)\right)}{\|\boldsymbol{y} - \boldsymbol{A}\widehat{\boldsymbol{\alpha}}\|_2}\right| > \epsilon\right) = 0.$$

*Proof.* Let us first define

(i) $\boldsymbol{v}_n = \boldsymbol{1}_n - (H_{11}, H_{22}, \cdots, H_{nn})^\top$.
(ii) $q_j := \operatorname{trace}(\boldsymbol{T}_{\mathrm{NL}}(\boldsymbol{e}_j)) = \operatorname{trace}\left[(\boldsymbol{I}_n - \boldsymbol{H})\operatorname{diag}(\boldsymbol{G}'\boldsymbol{e}_j)\right] = \boldsymbol{v}_n^\top[\boldsymbol{G}']_j$.
(iii) $\boldsymbol{r} := \boldsymbol{A}\widehat{\boldsymbol{\alpha}} - \boldsymbol{y}$.
(iv) $\Omega$ be in Proposition C.2.3 such that
    (i) $\Omega := \left\{1/n\cdot\|\boldsymbol{y} - \boldsymbol{A}\widehat{\boldsymbol{\alpha}}\|_2^2 \ge c_{2,n}\right\}$.
    (ii) $\mathbb{P}(\Omega^c) \le o(\exp(-c_6 n))$ for some constant $c_6 > 0$.
(v) $\overline{\Omega}_3$ be such that (cf. Corollary 2.8.3 and Proposition 2.5.2 (i) in Vershynin [2018])

(i)

$$\overline{\Omega}_3 = \left\{ \max_{i_1, i_2 \in [n]} |\boldsymbol{x}_{i_1}^T \boldsymbol{x}_{i_2}/p - \delta_{i_1, i_2}| < \delta \right\} \cap \left\{ \max_{j_1, j_2 \in [p]} |\boldsymbol{X}_{j_1}^\top \boldsymbol{X}_{j_2}/n - \delta_{j_1, j_2}| < \delta \right\}$$
$$\cap \left\{ \max_{i \in [n], j \in [p]} |x_{ij}| \le p^{1/4} \right\}$$

(ii) $\mathbb{P}(\overline{\Omega}_3^c) \le o(\exp(-c_{10} n^{1/2}))$ for some universal constant $c_{10} > 0$.

(vi) $\Omega_3 := \Omega \cap \overline{\Omega}_3$.

By Chebyshev's inequality,

$$\mathbb{P}\left( \max_{j \in [p]} \frac{|q_j|}{\|\boldsymbol{r}\|_2} > \epsilon \right) \le \mathbb{P}\left( \left\{ \max_{j \in [p]} \frac{|q_j|}{\|\boldsymbol{r}\|_2} > \epsilon \right\} \cap \Omega_3 \right) + \mathbb{P}(\Omega_3^c)$$
$$\le \mathbb{P}\left( \left\{ \max_{j \in [p]} \frac{|q_j|}{n^{-1/2} c_{2,n}^{1/2}} > \epsilon \right\} \cap \Omega_3 \right) + \mathbb{P}(\Omega_3^c)$$
$$\le \mathbb{P}\left( \left\{ \max_{j \in [p]} \frac{|q_j|}{n^{-1/2} c_{2,n}^{1/2}} > \epsilon \right\} \cap \overline{\Omega}_3 \right) + \mathbb{P}(\Omega_3^c)$$
$$\le \mathbb{P}\left( \left\{ \max_{j \in [p]} \frac{|q_j|}{n^{-1/2} c_{2,n}^{1/2}} I_{\overline{\Omega}_3} > \epsilon \right\} \right) + \mathbb{P}(\Omega_3^c)$$
$$\le \frac{\mathbb{E}\left[ \max_{j \in [p]} q_j^2 I_{\overline{\Omega}_3} \right]}{n \epsilon^2 c_{2,n}} + o(1).$$

So to show our proposition, it suffices to show that

$$\mathbb{E}\left[ \max_{j \in [p]} q_j^2 I_{\overline{\Omega}_3} \right] = o(n).$$

Letting $\boldsymbol{v}_n = \mathbf{1}_n - (H_{11}, H_{22}, \cdots, H_{nn})^\top$, we notice that by some algebra,

$$\mathbb{E}\left[ \max_{j \in [p]} q_j^2 I_{\overline{\Omega}_3} \right] = \mathbb{E}\left[ \max_{j \in [p]} \left( \text{trace} \left[ (\boldsymbol{I}_n - \boldsymbol{H}) \text{diag}(\boldsymbol{G}' \boldsymbol{e}_j) \right] \right)^2 I_{\overline{\Omega}_3} \right]$$
$$= \mathbb{E}\left[ \max_{j \in [p]} \left( \boldsymbol{v}_n^\top [\boldsymbol{G}']_j \right)^2 I_{\overline{\Omega}_3} \right]$$
$$= \mathbb{E}\left[ \max_{j \in [p]} \boldsymbol{v}_n^\top \mathbb{E}_G \left[ [\boldsymbol{G}']_j ([\boldsymbol{G}']_j)^\top \right] \boldsymbol{v}_n I_{\overline{\Omega}_3} \right]$$

Let

$$\boldsymbol{M}(\boldsymbol{e}_j) := \mathbb{E}_G \left[ [\boldsymbol{G}']_j ([\boldsymbol{G}']_j)^\top \right].$$

From Proposition C.2.7, on $\overline{\Omega}_3$, the $(i_1, i_2)$-th element of the above matrix is

$$m_{i_1, i_2}(\boldsymbol{e}_j) = \Sigma_p''(\boldsymbol{x}_{i_1}^\top \boldsymbol{x}_{i_2}/p) x_{i_1 j} x_{i_2 j}/p^2 + \Sigma_p'(\boldsymbol{x}_{i_1}^\top \boldsymbol{x}_{i_2}/p)/p.$$

Let $\delta_{i_1,i_2} = 1$ if $i_1 = i_2$, 0 otherwise. We look at Taylor expansions around $\delta_{i_1,i_2}$, and do some arrangement as following: for $i_1, i_2 \in [n]$ and $\left|\boldsymbol{x}_{i_1}^\top \boldsymbol{x}_{i_2}/p - \delta_{i_1,i_2}\right| \leq \delta$,

$$
\begin{aligned}
\Sigma_p'(\boldsymbol{x}_{i_1}^\top \boldsymbol{x}_{i_2}/p) &= \Sigma_p'(\delta_{i_1,i_2}) + \Sigma_p''(\kappa_{i_1,i_2})(\boldsymbol{x}_{i_1}^\top \boldsymbol{x}_{i_2}/p - \delta_{i_1,i_2}), \\
&= \Sigma_p'(\delta_{i_1,i_2}) + \Sigma_p''(0)(\boldsymbol{x}_{i_1}^\top \boldsymbol{x}_{i_2}/p - \delta_{i_1,i_2}) \\
&\quad + \left(\Sigma_p''(\kappa_{i_1,i_2}) - \Sigma_p''(0)\right)(\boldsymbol{x}_{i_1}^\top \boldsymbol{x}_{i_2}/p - \delta_{i_1,i_2})(1 - \delta_{i_1,i_2}) \\
&\quad + \left(\Sigma_p''(\kappa_{i_1,i_2}) - \Sigma_p''(0)\right)(\boldsymbol{x}_{i_1}^\top \boldsymbol{x}_{i_2}/p - \delta_{i_1,i_2})\delta_{i_1,i_2}, \\
\Sigma_p''(\boldsymbol{x}_{i_1}^\top \boldsymbol{x}_{i_2}/p)x_{i_1j}x_{i_2j} &= \Sigma_p''(0)x_{i_1j}x_{i_2j} + \left(\Sigma_p''(\boldsymbol{x}_{i_1}^\top \boldsymbol{x}_{i_2}/p) - \Sigma_p''(0)\right)x_{i_1j}x_{i_2j}(1 - \delta_{i_1,i_2}) \\
&\quad + \left(\Sigma_p''(\boldsymbol{x}_{i_1}^\top \boldsymbol{x}_{i_2}/p) - \Sigma_p''(0)\right)x_{i_1j}x_{i_2j}\delta_{i_1,i_2},
\end{aligned}
$$

where $\kappa_{i_1,i_2}$ satisfies $|\kappa_{i_1,i_2} - \delta_{i_1,i_2}| \leq \left|\boldsymbol{x}_{i_1}^\top \boldsymbol{x}_{i_2}/p - \delta_{i_1,i_2}\right|$. From this we have decomposition of $\boldsymbol{M}(\boldsymbol{e}_j) := \mathbb{E}_G\left[[\boldsymbol{G}']_j[\boldsymbol{G}']_j^\top\right]$ into several matrices with small operator norm easy to calculate. With a slight abuse of notations $\boldsymbol{A}, \boldsymbol{B}, \boldsymbol{C}, \boldsymbol{D}, \boldsymbol{E}, \boldsymbol{F}, \boldsymbol{G}, \boldsymbol{H}$, we have

$$
\boldsymbol{M} = \boldsymbol{A} + \boldsymbol{B} + \boldsymbol{C} + \boldsymbol{D} + \boldsymbol{E} + \boldsymbol{F} + \boldsymbol{G} + \boldsymbol{H},
$$

where

$$
\begin{aligned}
\boldsymbol{A} &= \left(\Sigma_p'(1) - \Sigma_p'(0)\right)\boldsymbol{I}_n/p, \\
\boldsymbol{B} &= \Sigma_p'(0)\mathbf{1}_n\mathbf{1}_n^\top/p, \\
\boldsymbol{C} &= \Sigma_p''(0)(\boldsymbol{X}\boldsymbol{X}^T/p)/p, \\
\boldsymbol{D}_{i_1,i_2} &= \left(\Sigma_p''(\kappa_{i_1,i_2}) - \Sigma_p''(0)\right)(\boldsymbol{x}_{i_1}^\top \boldsymbol{x}_{i_2}/p)\delta_{i_1\neq i_2}/p, \\
\boldsymbol{E}_{i_1,i_2} &= \left[\Sigma_p''(\kappa_{i_1,i_2})\boldsymbol{x}_{i_1}^\top \boldsymbol{x}_{i_2}/p - \Sigma_p''(0)\boldsymbol{x}_{i_1}^\top \boldsymbol{x}_{i_2}/p - \Sigma_p''(\kappa_{i_1,i_2})\right]\delta_{i_1=i_2}/p, \\
\boldsymbol{F} &= \Sigma_p''(0)\boldsymbol{X}_j\boldsymbol{X}_j^T/p^2, \\
\boldsymbol{G}_{i_1,i_2}(\boldsymbol{e}_j) &= \left(\Sigma_p''(\boldsymbol{x}_{i_1}^\top \boldsymbol{x}_{i_2}/p) - \Sigma_p''(0)\right)x_{i_1j}x_{i_2j}\delta_{i_1\neq i_2}/p^2, \\
\boldsymbol{H}_{i_1,i_2}(\boldsymbol{e}_j) &= \left(\Sigma_p''(\boldsymbol{x}_{i_1}^\top \boldsymbol{x}_{i_2}/p) - \Sigma_p''(0)\right)x_{i_1j}x_{i_2j}\delta_{i_1=i_2}/p^2.
\end{aligned}
$$

It suffices to show that $q(\boldsymbol{N}) := \mathbb{E}\left[\max_{j\in[p]} \boldsymbol{v}_n^\top \boldsymbol{N}(\boldsymbol{e}_j)\boldsymbol{v}_n I_{\overline{\Omega}_3}\right] = o(p)$ for $\boldsymbol{N}$ being from $\boldsymbol{A}$ to $\boldsymbol{H}$. We notice that $\|\boldsymbol{I}_n - \boldsymbol{H}\|_{\mathrm{op}} \leq 1$ implies $|v_{n,i}| \leq 1$ for all $i \in [n]$. Then we have

(i) $q(\boldsymbol{A}) = o(p)$ provided that $\Sigma_p'(1), \Sigma_p'(0) = o(p)$.
(ii) $q(\boldsymbol{B}) = o(p)$ provided that $\Sigma_p'(0) = o(1)$.
(iii) $q(\boldsymbol{C}) = o(p)$ provided that $\mathbb{E}\left[\left\|\boldsymbol{X}/\sqrt{p}\right\|_{\mathrm{op}}\right] = O(1)$ and $\Sigma_p''(0) = o(p)$.
(iv) $q(\boldsymbol{E}) = o(p)$ provided that $\sup_{x\in[1-\delta,1+\delta]} \Sigma_p''(x), \Sigma_p''(0) = o(p)$.
(v) $q(\boldsymbol{F}) = o(p)$ provided that $\Sigma_p''(0) = o(p)$.
(vi) $q(\boldsymbol{H}) = o(p)$ provided that $\sup_{x\in[1-\delta,1+\delta]} \Sigma_p''(x), \Sigma_p''(0) = o(p)$.

We notice that the above are true by assumptions on $\Sigma_p$ and $\boldsymbol{X}$. For $\boldsymbol{D}$ and $\boldsymbol{G}$, we notice the following:

$$
|q(\boldsymbol{D})| := \left|\mathbb{E}\left[\max_{j\in[p]} \boldsymbol{v}_n^\top \boldsymbol{D}(\boldsymbol{e}_j)\boldsymbol{v}_n I_{\overline{\Omega}_3}\right]\right| \leq \mathbb{E}\left[|\boldsymbol{v}_n|^\top |\boldsymbol{D}||\boldsymbol{v}_n| I_{\overline{\Omega}_3}\right] \leq \mathbf{1}_n^\top \mathbb{E}[|\boldsymbol{D}| I_{\overline{\Omega}_3}]\mathbf{1}_n,
$$

where the absolute value operation is taken element-wise for the vector $\boldsymbol{v}_n$ and matrix $\boldsymbol{D}$. By the Lipschitz assumption of $\Sigma_p''$ around 0, for $i_1 \neq i_2$,

$$
\mathbb{E}\left[|d_{i_1,i_2}| I_{\overline{\Omega}_3}\right] \leq \mathbb{E}\left[\left|\left(\Sigma_p''(\kappa_{i_1,i_2}) - \Sigma_p''(0)\right)\right| |\boldsymbol{x}_{i_1}^\top \boldsymbol{x}_{i_2}/p| I_{\overline{\Omega}_3}/p\right] \leq L_2/p \cdot \mathbb{E}\left[(\boldsymbol{x}_{i_1}^\top \boldsymbol{x}_{i_2}/p)^2\right] = L_2/p^2.
$$

This implies $|q(\boldsymbol{D})| = O(1)$. For $|q(\boldsymbol{G})|$, we notice that

$$
|q(\boldsymbol{G})| := \left|\mathbb{E}\left[\max_{j\in p} \boldsymbol{v}_n^\top \boldsymbol{G}(\boldsymbol{e}_j)\boldsymbol{v}_n I_{\overline{\Omega}_3}\right]\right| \leq \mathbb{E}\left[\max_{j\in[p]} |\boldsymbol{v}_n|^\top |\boldsymbol{G}(\boldsymbol{e}_j)| |\boldsymbol{v}_n| I_{\overline{\Omega}_3}\right] \leq \mathbb{E}\left[\max_{j\in[p]} \mathbf{1}_n^\top |\boldsymbol{G}(\boldsymbol{e}_j)| \mathbf{1}_n I_{\overline{\Omega}_3}\right]
$$

where the absolute value operation is taken element-wise for the vector $\boldsymbol{v}_n$ and matrix $\boldsymbol{G}(\boldsymbol{e}_j)$. By the Lipschitz assumption of $\Sigma_p''$ around 0, for $i_1 \neq i_2$,

$$\mathbb{E}\left[\max_{j \in [p]} |g_{i_1, i_2}(\boldsymbol{e}_j)| I_{\overline{\Omega}_3}\right] \leq L_2 \mathbb{E}\left[\max_{j \in [p]} |(\boldsymbol{x}_{i_1}^\top \boldsymbol{x}_{i_2}/p) x_{i_1 j} x_{i_2 j}| I_{\overline{\Omega}_3}\right]/p^2$$

$$\leq L_2 \delta \cdot \mathbb{E}\left[\max_{j \in [p]} |x_{i_1 j} x_{i_2 j}| I_{\overline{\Omega}_3}\right]/p^2$$

$$\leq L_2 \delta p^{-3/2},$$

where we used that $\max_j |x_{i_1 j} x_{i_2 j}| I_{\overline{\Omega}_3} \leq p^{1/2}$. So that $|q(\boldsymbol{G})| = O(p^{1/2})$. Combining the above we have our proposition. $\square$

### D.1.8. *Proof of Proposition C.2.7.*

**Proposition C.2.7.** *For all vector $\boldsymbol{v}_1, \boldsymbol{v}_2 \in \mathbb{R}^p$ such that the derivatives below exist,*

$$\mathbb{E}_G\left[\partial_{j_1} G(\boldsymbol{v}_1) \partial_{j_2} G(\boldsymbol{v}_2)\right] = \Sigma_p''(\boldsymbol{v}_1^\top \boldsymbol{v}_2/p) v_{1j_2} v_{2j_1}/p^2 + \Sigma_p'(\boldsymbol{v}_1^\top \boldsymbol{v}_2/p) \delta(j_1 = j_2)/p.$$

*Proof.* Let $\boldsymbol{t}_1, \boldsymbol{t}_2 \in \mathbb{R}^p$. Let

$$\mathscr{G}(\boldsymbol{t}_1, \boldsymbol{t}_2) = \mathbb{E}_G\left[G(\boldsymbol{v}_1 + \boldsymbol{t}_1) G(\boldsymbol{v}_2 + \boldsymbol{t}_2)\right] = \Sigma_p((\boldsymbol{v}_1 + \boldsymbol{t}_1)^\top (\boldsymbol{v}_2 + \boldsymbol{t}_2)/p).$$

Let $j_1, j_2 \in [p]$. Let us assume that the derivatives below exists,

$$\partial_{t_{2j_2}} \partial_{t_{1j_1}} \mathscr{G}(\boldsymbol{t}_1, \boldsymbol{t}_2) = \mathbb{E}_G\left[\partial_{j_1} G(\boldsymbol{v}_1 + \boldsymbol{t}_1) \partial_{j_2} G(\boldsymbol{v}_2 + \boldsymbol{t}_2)\right],$$

$$\partial_{t_{2j_2}} \partial_{t_{1j_1}} \mathscr{G}(\boldsymbol{t}_1, \boldsymbol{t}_2) = \Sigma_p''((\boldsymbol{v}_1 + \boldsymbol{t}_1)^\top (\boldsymbol{v}_2 + \boldsymbol{t}_2)/p)(v_{1j_2} + t_{1j_2})(v_{2j_1} + t_{2j_1})/p^2$$

$$+ \Sigma_p'((\boldsymbol{v}_1 + \boldsymbol{t}_1)^\top (\boldsymbol{v}_2 + \boldsymbol{t}_2)/p) \delta(j_1 = j_2)/p.$$

Then

$$\mathbb{E}_G\left[\partial_{j_1} G(\boldsymbol{v}_1) \partial_{j_2} G(\boldsymbol{v}_2)\right] = \partial_{t_{2j_2}} \partial_{t_{1j_1}} \mathscr{G}(\boldsymbol{0}_p, \boldsymbol{0}_p)$$

$$= \Sigma_p''(\boldsymbol{v}_1^\top \boldsymbol{v}_2/p) v_{1j_2} v_{2j_1}/p^2 + \Sigma_p'(\boldsymbol{v}_1^\top \boldsymbol{v}_2/p) \delta(j_1 = j_2)/p.$$

$\square$

### D.2. **Proofs of the limits $V(\psi_d, \psi_p, \lambda, \rho, \sigma)$ and $R(\psi_p, \lambda, \rho, \sigma)$.**

### D.2.1. *Proof of Proposition C.3.1.*

**Proposition C.3.1.** *Let $L^2 := \|\boldsymbol{y} - \boldsymbol{A}\widehat{\boldsymbol{\alpha}}\|_2^2/(\mathrm{trace}(\boldsymbol{I}_n - \boldsymbol{H}))^2$. Under the model (5) in the asymptotic setting (6), Assumption C.3.1 and Definition C.3.1,*

$$\mathrm{Var}(y_1) \to \theta_{\boldsymbol{\beta}}^2 + \theta_{\boldsymbol{\varepsilon}}^2 + \theta_{\mathrm{NL}}^2, \qquad nL^2/\mathrm{Var}(y_1) \overset{\mathbb{P}_{\boldsymbol{X}, \boldsymbol{W}, \boldsymbol{\varepsilon}}^{G, \boldsymbol{\varepsilon}}}{\to} V.$$

*Proof.* In this proof we consider the model in Mei and Montanari [2019]. Under this model, the limit of $(1/n) \|\boldsymbol{y} - \boldsymbol{A}\widehat{\boldsymbol{\alpha}}\|_2^2$ is given in Theorem 6 in Mei and Montanari [2019] as

$$\lim_{n \to +\infty} \mathbb{E}\left|(1/n) \|\boldsymbol{y} - \boldsymbol{A}\widehat{\boldsymbol{\alpha}}\|_2^2 - (\theta_{\boldsymbol{\beta}}^2 + \theta_{\boldsymbol{\varepsilon}}^2 + \theta_{\mathrm{NL}}^2)\left(\mathscr{L} - \frac{\psi_d \lambda}{\psi_p \mu_*^2}\mathscr{A}\right)\right| = 0.$$

The limit of $\mathrm{Var}(y_1)$ follows by

$$\mathrm{Var}(y_1) = \mathbb{E}\left[y_1^2\right] - \beta_0^2$$

$$= \mathbb{E}\left[(\beta_0 + \boldsymbol{x}_1^\top \boldsymbol{\beta} + G(\boldsymbol{x}_1) + \varepsilon_1)^2\right] - \beta_0^2$$

$$= \mathbb{E}\left[(\boldsymbol{x}_1^\top \boldsymbol{\beta})^2\right] + \mathbb{E}\left[(G(\boldsymbol{x}_1))^2\right] + \mathbb{E}\left[\varepsilon_1^2\right]$$

$$\to \theta_{\boldsymbol{\beta}}^2 + \theta_{\mathrm{NL}}^2 + \theta_{\boldsymbol{\varepsilon}}^2.$$

The limit of $(1/n)\operatorname{trace}(\boldsymbol{I}_n - \boldsymbol{H})$ is implied in Mei and Montanari [2019] by the following facts. Let us first denote $\boldsymbol{Z} = (1/\sqrt{p})\sigma(\boldsymbol{X}\boldsymbol{W}^\top)$. Let

$$\overline{\boldsymbol{Z}}(t) = (1+t)\begin{bmatrix} 0 & \boldsymbol{Z}^\top \\ \boldsymbol{Z} & 0 \end{bmatrix} \in \mathbb{R}^{(n+d)\times(n+d)}.$$

Let $u = (\psi_1\psi_2\lambda)^{1/2} \in \mathbb{R}^+$ and let log denote the complex logarithm with branch cut on the negative real axis. Let $\lambda_i(\overline{\boldsymbol{Z}}(t))$ be the eigenvalues of $\overline{\boldsymbol{Z}}(t)$ in non-increasing order. Let

$$\mathscr{J}(u,t) = (1/p)\sum_{i\in[n+d]} \log(\lambda_i(\overline{\boldsymbol{Z}}(t)) - \boldsymbol{i}u).$$

From Proposition 7.3 in Mei and Montanari [2019],

$$\partial_t\mathscr{J}(u,0) = \frac{2}{p}\operatorname{trace}((u^2\boldsymbol{I}_d + \boldsymbol{Z}^\top\boldsymbol{Z})^{-1}\boldsymbol{Z}^\top\boldsymbol{Z}) = \frac{2}{p}\operatorname{trace}(\boldsymbol{H}).$$

From the fact that $\overline{\boldsymbol{Z}}(t) = (1+t)\overline{\boldsymbol{Z}}(0)$, we have from the chain rule and the definition of $\mathscr{J}$,

$$\partial_t\mathscr{J}(u,0) = (1/p)\sum_{i\in[n+d]} (\lambda_i(\overline{\boldsymbol{Z}}(0)) - \boldsymbol{i}u)^{-1}\cdot(\boldsymbol{i}u) + \psi_1 + \psi_2.$$

From Proposition 7.2 and Step 2 in Lemma C.1. in Mei and Montanari [2019],

$$\lim_{n\to+\infty}\mathbb{E}\left|(1/p)\sum_{i\in[n+d]} (\lambda_i(\overline{\boldsymbol{Z}}(0)) - \boldsymbol{i}u)^{-1} - (\nu_1 + \nu_2)/\mu_*\right| = 0.$$

So that we have

$$\lim_{n\to+\infty}\mathbb{E}\left|(1/n)\operatorname{trace}(\boldsymbol{I}_n - \boldsymbol{H}) - (1/2)(1 - \psi_p((\nu_1+\nu_2)/\mu_*)u\boldsymbol{i} - \psi_d)\right| = 0.$$

We notice by the definition of $u$ and by (C.6),

$$1 - \psi_p((\nu_1+\nu_2)/\mu_*)u\boldsymbol{i} - \psi_d = 1 + \psi_p(\nu_1+\nu_2)(-\bar{z}) - \psi_d$$

$$= 1 + \psi_p\left(\psi_1 + \psi_2 + 2\chi + \frac{2\varrho^2\chi}{1-\varrho^2\chi}\right) - \psi_d$$

$$= 2 + \psi_p(2\chi + \frac{2\varrho^2\chi}{1-\varrho^2\chi}).$$

This implies

$$\lim_{n\to+\infty}\mathbb{E}\left|(1/n)\operatorname{trace}(\boldsymbol{I}_n - \boldsymbol{H}) - \mathscr{Q}\right| = 0.$$

Combining the above, we have the limit of $nL^2/\operatorname{Var}(y_1)$. $\qquad\square$

D.2.2. *Sketch of Proof of Proposition C.3.2.*

**Proposition C.3.2.** *Under the pure linear model* (4) *in the asymptotic setting* (6), *Assumption 1 and 2 with (ii) and Definition C.3.1,*

$$\operatorname{Var}(y_1) \to \theta_{\boldsymbol{\beta}}^2 + \theta_{\boldsymbol{\varepsilon}}^2, \qquad nL^2/\operatorname{Var}(y_1) \xrightarrow{\mathbb{P}_{\boldsymbol{X},\boldsymbol{W},\boldsymbol{\varepsilon}}} V.$$

*Proof.* In this sketch of proof, we consider the setting for the pure linear model,

$$\boldsymbol{y} = \boldsymbol{X}\boldsymbol{\beta} + \boldsymbol{\varepsilon},$$

with Gaussian $\boldsymbol{x}_i$ and $\boldsymbol{w}_k$ under Assumptions 1 and 2 with (ii). The limit of $(1/n)\operatorname{trace}(\boldsymbol{I}_n - \boldsymbol{H})$ satisfies

$$\lim_{n\to+\infty}\mathbb{E}\left|(1/n)\operatorname{trace}(\boldsymbol{I}_n - \boldsymbol{H}) - \mathscr{Q}\right| = 0,$$

under the same reasoning in Section D.2.1, which also holds for $\boldsymbol{x}_i, \boldsymbol{w}_k$ being Gaussian.

So it suffices to show that the limit of $(1/n)\|\boldsymbol{y} - \boldsymbol{A}\widehat{\boldsymbol{\alpha}}\|_2$ is the same as that in Section D.2.1, that is,

$$\lim_{n\to+\infty} \mathbb{E}\left|(1/n)\|\boldsymbol{y} - \boldsymbol{A}\widehat{\boldsymbol{\alpha}}\|_2^2 - (\theta_{\boldsymbol{\beta}}^2 + \theta_{\boldsymbol{\varepsilon}}^2)\left(\mathscr{L} - \frac{\psi_d\lambda}{\psi_p\mu_*^2}\mathscr{A}\right)\right| = 0.$$

First, by Remark 8 in Mei and Montanari [2019], we can consider $\boldsymbol{\beta} \in \mathbb{S}^{p-1}(\|\boldsymbol{\beta}\|_2^2)$ independent of $\boldsymbol{X}, \boldsymbol{\varepsilon}, \boldsymbol{W}$, instead of $\boldsymbol{\beta}$ being deterministic, since the training error $\|\boldsymbol{y} - \boldsymbol{A}\widehat{\boldsymbol{\alpha}}\|_2^2$ as a function of $\boldsymbol{\beta}$ is invariant in distribution after orthogonal rotation of $\boldsymbol{\beta}$. So letting $\mathbb{E}$ denote $\mathbb{E}_{\boldsymbol{X},\boldsymbol{W},\boldsymbol{\varepsilon},\boldsymbol{\beta}\sim\mathrm{Unif}(\mathbb{S}^{p-1}(\|\boldsymbol{\beta}\|_2^2))}$, by the fact that $\boldsymbol{y} - \boldsymbol{A}\widehat{\boldsymbol{\alpha}} = (\boldsymbol{I}_n - \boldsymbol{H})\boldsymbol{y}$, we have

$$
\begin{aligned}
(1/n)\mathbb{E}\left[\|\boldsymbol{y} - \boldsymbol{A}\widehat{\boldsymbol{\alpha}}\|_2^2\right] &= (1/n)\mathbb{E}\left[\|(\boldsymbol{I}_n - \boldsymbol{H})\boldsymbol{y}\|_2^2\right] \\
&= (1/n)\mathbb{E}\left[\|(\boldsymbol{I}_n - \boldsymbol{H})(\boldsymbol{X}\boldsymbol{\beta} + \boldsymbol{\varepsilon})\|_2^2\right] \\
&= (1/n)\left[\mathbb{E}\left[\mathrm{trace}(\widetilde{\boldsymbol{H}}\boldsymbol{X}\boldsymbol{\beta}\boldsymbol{\beta}^\top\boldsymbol{X}^\top)\right] + \mathbb{E}\left[\mathrm{trace}(\widetilde{\boldsymbol{H}}\boldsymbol{\varepsilon}\boldsymbol{\varepsilon}^\top)\right]\right] \\
&= (1/n)\left[\theta_{\boldsymbol{\beta}}^2\mathbb{E}\left[\mathrm{trace}\left(\widetilde{\boldsymbol{H}}\boldsymbol{X}\boldsymbol{X}^\top/p\right)\right] + \theta_{\boldsymbol{\varepsilon}}^2\mathbb{E}\left[\mathrm{trace}\left(\widetilde{\boldsymbol{H}}\right)\right]\right]
\end{aligned}
$$

where

$$\widetilde{\boldsymbol{H}} := (\boldsymbol{I}_n - \boldsymbol{H})^2 = (n\tau)^2(\boldsymbol{A}\boldsymbol{A}^\top + n\tau\boldsymbol{I}_n)^{-2}.$$

We notice that the expected value of traces above can be calculated by the quantities in (228) in Mei and Montanari [2019]. We also claim that the calculations of the traces in the (228) are the same for the random vectors on the spheres and the Gaussian random vectors, cf. Section C in the Appendix of Mei and Montanari [2019], especially Lemma C.1, Proposition 7.2 and Lemma C.7 there. So that $\|\boldsymbol{y} - \boldsymbol{A}\widehat{\boldsymbol{\alpha}}\|_2^2$ has the same limit as that in Theorem 6 in Mei and Montanari [2019]. □

D.2.3. *Proof of Proposition C.3.3.*

**Proposition C.3.3.** *Let $V, R$ be in Definitions C.3.1 and C.3.2. Then*

$$\lim_{\psi_d\to+\infty} V(\psi_d, \psi_p, \lambda, \rho, \sigma) = R(\psi_p, \lambda, \rho, \sigma).$$

*Proof.* In this proof we let $\to$ denote the convergence when $\psi_d \to +\infty$. We recall all the notations in Definition C.3.1 and C.3.2. We notice that $|\chi| \leq |\nu_1| \cdot |\nu_2| \leq 1/\bar{\lambda}^2$ is always bounded when $\psi_d \to +\infty$. From (C.6),

(D.2)
$$
\begin{aligned}
\nu_1(-\bar{z}) &= \psi_1 + \chi + \frac{\varrho^2\chi}{1 - \varrho^2\chi}, \\
\nu_2(-\bar{z}) &= \psi_2 + \chi + \frac{\varrho^2\chi}{1 - \varrho^2\chi}.
\end{aligned}
$$

The quantities $\nu_1, \nu_2$ are purely imaginary with positive imaginary part, so that $\chi$ is a negative real. In fact,

$$\nu_1 - \nu_2 = (\psi_1 - \psi_2)/(-\bar{z}) = \frac{\psi_1 - \psi_2}{(\psi_1\psi_2\lambda)^{1/2}}\mu_*\boldsymbol{i}$$

is purely imaginary. We can specify the real and the imaginary parts of $\nu_1$, $\nu_2$,

$$\nu_1 = a + (b+c)\boldsymbol{i}, \qquad \nu_2 = a + b\boldsymbol{i}, \qquad c = \frac{\psi_1 - \psi_2}{(\psi_1\psi_2\lambda)^{1/2}}\mu_*, \qquad \chi = (a^2 - b^2 - bc) + (2ab + ac)\boldsymbol{i}.$$

From the fact that $\mathscr{L}, \mathscr{Q}, \mathscr{A}$ are real numbers, we deduce that $\chi$ is real number. So that $a(2b+c) = 0$. From the fact that $\nu_1, \nu_2 \in \mathbb{C}_+$, we have $2b + c > 0$ so that $a = 0$. This shows that $\nu_1, \nu_2$ are purely

imaginary numbers and $\chi < 0$. Next, from (D.2), we have

$$\chi(-\psi_1\psi_2\overline{\lambda}) = \left(\psi_1 + \chi + \frac{\varrho^2\chi}{1 - \varrho^2\chi}\right)\left(\psi_2 + \chi + \frac{\varrho^2\chi}{1 - \varrho^2\chi}\right),$$

$$\implies \psi_1 = \frac{-\left(\chi + \frac{\varrho^2\chi}{1-\varrho^2\chi}\right)\left(\psi_2 + \chi + \frac{\varrho^2\chi}{1-\varrho^2\chi}\right)}{\psi_2 + \chi + \frac{\varrho^2\chi}{1-\varrho^2\chi} + \chi\psi_2\overline{\lambda}}.$$

When $\psi_1 \to +\infty$ and $\varrho \neq 0$ and $\psi_2 \neq 0$ fixed and $\chi$ is bounded, negative, we have

$$\psi_2 + \chi + \frac{\varrho^2\chi}{1 - \varrho^2\chi} + \chi\psi_2\overline{\lambda} \to 0,$$

$$\implies \chi^2 + \left(\frac{\psi_2 - 1}{1 + \psi_2\overline{\lambda}} - \varrho^{-2}\right)\chi - \frac{\psi_2}{1 + \psi_2\overline{\lambda}}\varrho^{-2} \to 0,$$

$$\implies \chi \to \overline{\chi} := \frac{\left(\varrho^{-2} - \frac{\psi_2-1}{1+\psi_2\overline{\lambda}}\right) - \sqrt{\left(\varrho^{-2} - \frac{\psi_2-1}{1+\psi_2\overline{\lambda}}\right)^2 + 4\frac{\psi_2}{1+\psi_2\overline{\lambda}}\varrho^{-2}}}{2}.$$

When $\varrho = 0$, we define $\overline{\chi} := -\frac{\psi_2}{1+\psi_2\overline{\lambda}}$ and we have $\chi \to \overline{\chi}$ still holds.

The quantities have limits

$$\mathcal{Q} \to \overline{\mathcal{Q}} := \psi_2^{-1}(-\overline{\chi}\psi_2\overline{\lambda}) = -\overline{\chi}\overline{\lambda},$$

$$\mathcal{L} \to \overline{\mathcal{L}} := \left(-\overline{\chi}\overline{\lambda}\right)\left[\frac{\rho}{1 + \rho}\frac{1}{1 - \overline{\chi}\varrho^2} + \frac{1}{1 + \rho}\right],$$

$$\mathcal{A}_1 \to \overline{\mathcal{A}_1} := \frac{\rho}{1 + \rho}\left[-\overline{\chi}^2\left(\overline{\chi}\varrho^4 - \overline{\chi}\varrho^2 + \psi_2\varrho^2 + \varrho^2 - \overline{\chi}\psi_2\varrho^4 + 1\right)\right]$$

$$+ \frac{1}{1 + \rho}\left[\overline{\chi}^2\left(\overline{\chi}\varrho^2 - 1\right)\left(\overline{\chi}^2\varrho^4 - 2\overline{\chi}\varrho^2 + \varrho^2 + 1\right)\right],$$

$$\mathcal{A}_0/\psi_1 \to \overline{\mathcal{A}_*} := (\psi_2 - 1)\overline{\chi}^3\varrho^6 + (1 - 3\psi_2)\overline{\chi}^2\varrho^4 + 3\psi_2\overline{\chi}\varrho^2 - \psi_2.$$

So that

$$V = (\mathcal{L} - \psi_1\overline{\lambda}\mathcal{A})/\mathcal{Q}^2 \to R = (\overline{\mathcal{L}} - \overline{\lambda\mathcal{A}_1/\mathcal{A}_*})/\overline{\mathcal{Q}}^2.$$

Furthermore, by simple algebra, we have that when $\varrho = 0$, $(\overline{\mathcal{L}} - \overline{\lambda\mathcal{A}_1/\mathcal{A}_*})/\overline{\mathcal{Q}}^2 = 1$. $\square$

### D.3. Proof of Theorem C.4.1.

**Theorem C.4.1.** *Let $t \in \mathbb{R}$. Under model (5), Assumption 1, 2, 3 and 4, Definition 1 and a further assumption that $\Sigma_p'(0) = O(1/p)$, we have*

$$(C.7) \qquad \sup_{\boldsymbol{u}_0 \in S_p}\left|\mathbb{P}_{\boldsymbol{X},\boldsymbol{W},\boldsymbol{\varepsilon},G|\boldsymbol{u}_0}\left(\frac{\zeta_L(\boldsymbol{u}_0)}{\|\boldsymbol{y} - \boldsymbol{A}\widehat{\boldsymbol{\alpha}}\|_2} \leq t\right) - \Phi(t)\right| \to 0$$

*for some $S_p \subset \mathbb{S}^{p-1}(1)$ satisfying $|S_p|/|\mathbb{S}^{p-1}(1)| \geq 1 - \log(p)/p \to 1$.*

D.3.1. *Proof of Theorem C.4.1.*

*Proof.* Let $\zeta$ be as in Definition C.2.1. Provided with Propositions C.4.1 and C.4.2, we will show that, for a large proportion of $\boldsymbol{u}_0 \in \mathbb{S}^{p-1}$, given $\boldsymbol{u}_0$,

(i) $\frac{\zeta(\boldsymbol{u}_0)}{(\mathrm{Var}_0[\zeta(\boldsymbol{u}_0)])^{1/2}} \xrightarrow{d} N(0, 1).$

(ii) $\frac{\|\boldsymbol{y} - \boldsymbol{A}\widehat{\boldsymbol{\alpha}}\|_2}{(\mathrm{Var}_0[\zeta(\boldsymbol{u}_0)])^{1/2}} \xrightarrow{\mathbb{P}} 1.$

(iii) $\frac{\zeta(\boldsymbol{u}_0) - \zeta_L(\boldsymbol{u}_0)}{\|\boldsymbol{y} - \boldsymbol{A}\widehat{\boldsymbol{\alpha}}\|_2} \xrightarrow{\mathbb{P}} 0.$

The above convergence are uniform over $\boldsymbol{u}_0 \in S_p$, where $S_p$ is a large subset of $\mathbb{S}^{p-1}(1)$. So by Slutsky's Theorem we have our proposition.

We specify $S_p$ as follows. Let $\boldsymbol{v}_n = \mathbf{1}_n - (H_{11}, H_{22}, \cdots, H_{nn})^\top$, $\boldsymbol{r} := \boldsymbol{y} - \boldsymbol{A}\widehat{\boldsymbol{\alpha}}$ and $q_0 :=$ $\mathrm{trace}(\boldsymbol{T}_{\mathrm{NL}}(\boldsymbol{u}_0)) = \mathrm{trace}\left[(\boldsymbol{I}_n - \boldsymbol{H})\,\mathrm{diag}(\boldsymbol{G}'\boldsymbol{u}_0)\right] = \boldsymbol{v}_n^\top \boldsymbol{G}'\boldsymbol{u}_0$. Notice that (iii) in Proposition C.4.1 and Proposition C.4.2 provide the existence of a constant $c_9 > 0$ independent of $n, p, d$ such that

$$\max\left(\mathbb{E}_{\boldsymbol{u}_0,\boldsymbol{X},\boldsymbol{W},G,\boldsymbol{\varepsilon}}\left[\epsilon_0^2\right], \mathbb{E}_{\boldsymbol{u}_0,\boldsymbol{X},\boldsymbol{W},G,\boldsymbol{\varepsilon}}\left[q_0^2/\|\boldsymbol{r}\|_2^2\, I_{\Omega_3}\right]\right) \le c_9/p$$

We specify the large volume index set $S_p \subset \mathbb{S}^{p-1}(1)$ as

$$S_p := \left\{\boldsymbol{u}_0 \in \mathbb{S}^{p-1} : \max\left(\mathbb{E}_{\boldsymbol{X},\boldsymbol{W},\boldsymbol{\varepsilon},G|\boldsymbol{u}_0}\left[\epsilon_0^2\right], \mathbb{E}_{\boldsymbol{X},\boldsymbol{W},\boldsymbol{\varepsilon},G|\boldsymbol{u}_0}\left[q_0^2/\|\boldsymbol{r}\|_2^2\, I_{\Omega_3}\right]\right) \le \frac{2c_9}{\log(p)}\right\}.$$

Since

$$\mathbb{P}_{\boldsymbol{u}_0}(S_p^c) = \mathbb{P}_{\boldsymbol{u}_0}\left(\mathbb{E}_{\boldsymbol{X},\boldsymbol{W},\boldsymbol{\varepsilon},G|\boldsymbol{u}_0}\left[\epsilon_0^2\right] \ge \frac{2c_9}{\log(p)} \text{ or } \mathbb{E}_{\boldsymbol{X},\boldsymbol{W},\boldsymbol{\varepsilon},G|\boldsymbol{u}_0}\left[q_0^2/\|\boldsymbol{r}\|_2^2\, I_{\Omega_3}\right] \ge \frac{2c_9}{\log(p)}\right)$$

$$\le \frac{\mathbb{E}_{\boldsymbol{u}_0,\boldsymbol{X},\boldsymbol{W},\boldsymbol{\varepsilon},G}\left[\epsilon_0^2\right] + \mathbb{E}_{\boldsymbol{u}_0,\boldsymbol{X},\boldsymbol{W},\boldsymbol{\varepsilon},G}\left[q_0^2/\|\boldsymbol{r}\|_2^2\, I_{\Omega_3}\right]}{2c_9}\log(p) \le \log(p)/p,$$

the relative volume $|S_p|/|\mathbb{S}^{p-1}(1)| \ge 1 - \log(p)/p \to 1$ as $p \to +\infty$.

We notice that (i) can be directly obtained from Proposition C.4.1 (i), by noticing that $(\boldsymbol{X}\boldsymbol{u}_0)^\top \mathbb{E}_0\left[\boldsymbol{r}\right]/\|\mathbb{E}_0\left[\boldsymbol{r}\right]\|_2 \sim N(0,1)$.

For (iii), we notice that for any $\epsilon > 0$,

$$\sup_{\boldsymbol{u}_0 \in S_p} \mathbb{P}_{\boldsymbol{X},\boldsymbol{W},\boldsymbol{\varepsilon},G|\boldsymbol{u}_0}\left(\left|\frac{\zeta(\boldsymbol{u}_0) - \zeta_{\mathrm{L}}(\boldsymbol{u}_0)}{\|\boldsymbol{y} - \boldsymbol{A}\widehat{\boldsymbol{\alpha}}\|_2}\right| > \epsilon\right) = \sup_{\boldsymbol{u}_0 \in S_p} \mathbb{P}_{\boldsymbol{X},\boldsymbol{W},\boldsymbol{\varepsilon},G|\boldsymbol{u}_0}\left(\left|\frac{q_0}{\|\boldsymbol{r}\|_2}\right| > \epsilon\right)$$

$$\le \sup_{\boldsymbol{u}_0 \in S_p} \mathbb{P}_{\boldsymbol{X},\boldsymbol{W},\boldsymbol{\varepsilon},G|\boldsymbol{u}_0}\left(\left|\frac{q_0}{\|\boldsymbol{r}\|_2}\right| I_{\Omega_3} > \epsilon\right)$$

$$\quad + \mathbb{P}_{\boldsymbol{X},\boldsymbol{W},\boldsymbol{\varepsilon},G}(I_{\Omega_3}^c)$$

$$\le \frac{2c_9}{\log(p)\epsilon^2} + o(-\exp(-\min(c_6, c_{11})n))$$

$$= o(1).$$

For (ii), we first recall Proposition C.4.1 (ii). Let $\epsilon > 0$ be a fixed number. From the definition of $S_p$ combined with Chebyshev's inequality, we can see that for all $\bar{\epsilon} > 0$,

$$\sup_{\boldsymbol{u}_0 \in S_p} \mathbb{P}_{\boldsymbol{X},\boldsymbol{W},\boldsymbol{\varepsilon},G|\boldsymbol{u}_0}(|\epsilon_0| > \bar{\epsilon}) \le \frac{2c_9}{\bar{\epsilon}^2 \log(p)}.$$

Let $\boldsymbol{u}_0 \in S_p$. Letting $\boldsymbol{r} = \boldsymbol{A}\widehat{\boldsymbol{\alpha}} - \boldsymbol{y}$, $V_0 := \mathrm{Var}_0\left[\zeta(\boldsymbol{u}_0)\right]$ and $U_0 := \left|\|\boldsymbol{r}\|_2/V_0^{1/2} - 1\right|$, we have that if $\epsilon_0 \le \frac{1}{2}$,

$$\mathbb{E}_0\left[U_0\right] \le \left(1 + \sqrt{2}\right)\epsilon_0/(1 - 2\epsilon_0^2)_+^{1/2} \le (2 + \sqrt{2})\epsilon_0.$$

Now let us focus on $\bar{\epsilon} < 1/2$. We let $\Omega_0(\bar{\epsilon}) := \left\{\mathbb{E}_0\left[U_0\right] < (2 + \sqrt{2})\bar{\epsilon}\right\}$. Then

$$\mathbb{P}_{\boldsymbol{X},\boldsymbol{W},\boldsymbol{\varepsilon},G|\boldsymbol{u}_0}(\Omega_0(\bar{\epsilon})) = \mathbb{P}_{\boldsymbol{X},\boldsymbol{W},\boldsymbol{\varepsilon},G|\boldsymbol{u}_0}\left(\mathbb{E}_0\left[U_0\right] < (2 + \sqrt{2})\bar{\epsilon}\right) \ge \mathbb{P}_{\boldsymbol{X},\boldsymbol{W},\boldsymbol{\varepsilon},G|\boldsymbol{u}_0}(|\epsilon_0| \le \bar{\epsilon}) \ge 1 - \frac{2c_9}{\bar{\epsilon}^2 \log(p)}.$$

Then, letting $\mathbb{I}(\cdot) := I_{\{\cdot\}}$ be the indicator function, we have

$$\mathbb{P}_{\boldsymbol{X},\boldsymbol{W},\boldsymbol{\varepsilon},G|\boldsymbol{u}_0}(U_0 > \epsilon) := \mathbb{E}_{\boldsymbol{X},\boldsymbol{W},\boldsymbol{\varepsilon},G|\boldsymbol{u}_0}\left[\mathbb{I}(U_0 > \epsilon)\right]$$

$$= \mathbb{E}_{\boldsymbol{X},\boldsymbol{W},\boldsymbol{\varepsilon},G|\boldsymbol{u}_0}\left[\mathbb{E}_0\left[\mathbb{I}(U_0 > \epsilon)\right] I_{\Omega_0(\bar{\epsilon})}\right] + \mathbb{E}_{\boldsymbol{X},\boldsymbol{W},\boldsymbol{\varepsilon},G|\boldsymbol{u}_0}\left[\mathbb{E}_0\left[\mathbb{I}(U_0 > \epsilon)\right] I_{\Omega_0^c(\bar{\epsilon})}\right]$$

$$\leq \mathbb{E}_{\boldsymbol{X},\boldsymbol{W},\boldsymbol{\varepsilon},G|\boldsymbol{u}_0}\left[\frac{\mathbb{E}_0\left[U_0\right]}{\epsilon} I_{\Omega_0(\bar{\epsilon})}\right] + \mathbb{P}_{\boldsymbol{X},\boldsymbol{W},\boldsymbol{\varepsilon},G|\boldsymbol{u}_0}(\Omega_0^c(\bar{\epsilon}))$$

$$\leq \left(2 + \sqrt{2}\right) \bar{\epsilon}/\epsilon + \frac{2c_9}{\bar{\epsilon}^2 \log(p)}.$$

Choosing $\bar{\epsilon} := \min\left(\frac{1}{\log\log(p)}, \frac{1}{2}\right)$, we have that, for all $\epsilon > 0$,

$$\lim_{p\to+\infty} \sup_{\boldsymbol{u}_0 \in S_p} \mathbb{P}_{\boldsymbol{X},\boldsymbol{W},\boldsymbol{\varepsilon},G|\boldsymbol{u}_0}(U_0 > \epsilon) = 0.$$

Thus we have (ii). $\qquad\square$

D.3.2. *Proof of Proposition C.4.1.*

**Proposition C.4.1.** *Let*

    *(i) $\boldsymbol{u}_0 \sim \mathrm{Unif}\left(\mathbb{S}^{p-1}(1)\right)$ independent with $\boldsymbol{X}, \boldsymbol{W}, G, \boldsymbol{\varepsilon}$.*
    *(ii) $\boldsymbol{X}_0 := \boldsymbol{X}\boldsymbol{u}_0$.*
    *(iii) $\zeta(\boldsymbol{u}_0) := \boldsymbol{u}_0^\top \left(\zeta(\boldsymbol{e}_j)\right)^{j\in[p]}$ where $\zeta(\boldsymbol{e}_j)$ is defined in Definition C.2.1.*
    *(iv) $\boldsymbol{r} = \boldsymbol{y} - \boldsymbol{A}\widehat{\boldsymbol{\alpha}}$.*
    *(v) $\epsilon_0^2 := \mathbb{E}_0\left[\|\nabla_{\boldsymbol{X}_0}\boldsymbol{r}\|_F^2\right] / \left(\mathbb{E}_0\left[\|\boldsymbol{r}\|_2^2\right] + \mathbb{E}_0\left[\|\nabla_{\boldsymbol{X}_0}\boldsymbol{r}\|_F^2\right]\right).$*

*Then*

    *(i) $\mathbb{E}_0\left[\left(\frac{\zeta(\boldsymbol{u}_0)}{(\mathrm{Var}_0\zeta(\boldsymbol{u}_0))^{1/2}} - \frac{\boldsymbol{X}_0^\top \mathbb{E}_0[\boldsymbol{r}]}{\|\mathbb{E}_0[\boldsymbol{r}]\|_2}\right)^2\right] \leq 6\epsilon_0^2.$*
    *(ii) $\mathbb{E}_0\left[\left|\frac{\|\boldsymbol{r}\|_2}{(\mathrm{Var}_0[\zeta(\boldsymbol{u}_0)])^{1/2}} - 1\right|\right] \leq \left(1 + \sqrt{2}\right) \epsilon_0/(1 - 2\epsilon_0^2)_+^{1/2}.$*
    *(iii) $\mathbb{E}_{\boldsymbol{u}_0,\boldsymbol{X},\boldsymbol{W},\boldsymbol{\varepsilon},G}\left[\epsilon_0^2\right] = O(1/p).$*

*Proof.*     (i) The proof is the same as that of Proposition C.2.5 (i).
    (ii) The proof is the same as that of Proposition C.2.5 (ii).
    (iii) For a general vector $\boldsymbol{u}_0$ satisfying $\|\boldsymbol{u}_0\|_2 = 1$, the propositions discussed before for canonical basis $\boldsymbol{e}_j$ can be updated as follows.
        **Step 1.** We show that Proposition C.2.2 (iii) and (iv) can be replaced with

$$p\mathbb{E}_{\boldsymbol{u}_0}\|\boldsymbol{T}_0(\boldsymbol{u}_0) + \boldsymbol{T}_{\mathrm{L}}(\boldsymbol{u}_0) + \boldsymbol{T}_{\mathrm{NL}}(\boldsymbol{u}_0)\|_F^2 \leq 2c_1^2 L^2 n\|\widehat{\boldsymbol{\alpha}}\|_2^2 + 2\|\boldsymbol{f}'\|_F^2.$$

    and

$$\|\boldsymbol{T}_1(\boldsymbol{u}_0)\|_F^2 \leq L^2 c_1^2/(4n\tau) \cdot \|\boldsymbol{y} - \boldsymbol{A}\widehat{\boldsymbol{\alpha}}\|_2^2.$$

    We will use the fact that $\mathbb{E}_{\boldsymbol{u}_0}\boldsymbol{u}_0\boldsymbol{u}_0^\top = (1/p)\boldsymbol{I}_p$ and $\|\boldsymbol{u}_0\|_2 = 1$.

$$p\mathbb{E}_{\boldsymbol{u}_0}\|\boldsymbol{T}_0(\boldsymbol{u}_0) + \boldsymbol{T}_{\mathrm{L}}(\boldsymbol{u}_0) + \boldsymbol{T}_{\mathrm{NL}}(\boldsymbol{u}_0)\|_F^2 \leq p\mathbb{E}_{\boldsymbol{u}_0}\|\boldsymbol{I}_n - \boldsymbol{H}\|_{\mathrm{op}}^2 \left\|\left(\sigma'(\boldsymbol{X}\boldsymbol{W}^\top)\operatorname{diag}(\widehat{\boldsymbol{\alpha}})\boldsymbol{W} - \boldsymbol{f}'\right)\boldsymbol{u}_0\right\|_2^2$$

$$\leq p\mathbb{E}_{\boldsymbol{u}_0}\left\|\left(\sigma'(\boldsymbol{X}\boldsymbol{W}^\top)\operatorname{diag}(\widehat{\boldsymbol{\alpha}})\boldsymbol{W} - \boldsymbol{f}'\right)\boldsymbol{u}_0\right\|_2^2$$

$$= \left\|\sigma'(\boldsymbol{X}\boldsymbol{W}^\top)\operatorname{diag}(\widehat{\boldsymbol{\alpha}})\boldsymbol{W} - \boldsymbol{f}'\right\|_F^2$$

$$\leq 2\left\|\sigma'(\boldsymbol{X}\boldsymbol{W}^\top)\operatorname{diag}(\widehat{\boldsymbol{\alpha}})\boldsymbol{W}\right\|_F^2 + 2\left\|\boldsymbol{f}'\right\|_F^2.$$

$$\leq 2\|\boldsymbol{W}\|_{\mathrm{op}}^2 \left\|\sigma'(\boldsymbol{X}\boldsymbol{W}^\top)\operatorname{diag}(\widehat{\boldsymbol{\alpha}})\right\|_F^2 + 2\left\|\boldsymbol{f}'\right\|_F^2$$

$$\leq 2c_1^2 L^2 n \|\widehat{\boldsymbol{\alpha}}\|_2^2 + 2\left\|\boldsymbol{f}'\right\|_F^2.$$

We used $\|I_n - H\|_{\mathrm{op}} \leq 1$ in the second inequality in the above display. In the equality above, We used the fact that $\mathbb{E}_{u_0} u_0 u_0^\top = (1/p)I_p$ and that

$$\mathbb{E}_{u_0} \left\| \left( \sigma'(XW^\top) \operatorname{diag}(\widehat{\alpha})W - f' \right) u_0 \right\|_2^2$$

$$= \mathbb{E}_{u_0} \operatorname{trace} \left( \left( \sigma'(XW^\top) \operatorname{diag}(\widehat{\alpha})W - f' \right) u_0 u_0^\top \left( \sigma'(XW^\top) \operatorname{diag}(\widehat{\alpha})W - f' \right)^\top \right)$$

$$= \operatorname{trace} \left( \left( \sigma'(XW^\top) \operatorname{diag}(\widehat{\alpha})W - f' \right) \mathbb{E}_{u_0} \left[ u_0 u_0^\top \right] \left( \sigma'(XW^\top) \operatorname{diag}(\widehat{\alpha})W - f' \right)^\top \right)$$

$$= (1/p) \left\| \sigma'(XW^\top) \operatorname{diag}(\widehat{\alpha})W - f' \right\|_F^2 .$$

Noticing that $\|u_0\|_2 = 1$, we have

$$\begin{aligned}
\|T_1(u_0)\|_F^2 &\leq \|A(n\tau I_d + A^\top A)^{-1}\|_{\mathrm{op}}^2 \cdot \| \operatorname{diag}(Wu_0)\sigma'(WX^\top) \operatorname{diag}(y - A\widehat{\alpha})\|_F^2 \\
&\leq 1/(4n\tau) \cdot L^2 \cdot \|(Wu_0)(y - A\widehat{\alpha})^\top\|_F^2 \\
&= L^2/(4n\tau) \cdot \|Wu_0\|_2^2 \|y - A\widehat{\alpha}\|_2^2 \\
&\leq L^2/(4n\tau) \cdot \|W\|_{\mathrm{op}}^2 \|y - A\widehat{\alpha}\|_2^2 . \\
&\leq L^2 c_1^2/(4n\tau) \cdot \|y - A\widehat{\alpha}\|_2^2 .
\end{aligned}$$

**Step 2.** By the proof of Proposition C.2.3, there exists a class of large events $\{\Omega(u_0)\}_{u_0 \in \mathbb{S}^{p-1}(1)}$ such that $\mathbb{E}_{X,\varepsilon|W,G,u_0}(I_{\Omega(u_0)}) \geq 1 - o(\exp(-c_{12}n))$ for some constant $c_{12} > 0$ independent of $u_0$ and that on $\Omega(u_0)$,

$$\mathbb{E}_0 \|y - A\widehat{\alpha}\|_2^2 \geq n \cdot c_{2,n}.$$

The starting point of the construction is as follows: Since the mapping $\varepsilon \mapsto \mathbb{E}_0 \|y - A\widehat{\alpha}\|_2$ is 1-Lipschitz, by Theorem 5.2.2 in Vershynin [2018], for some universal constant $c_5 > 0$ and for all $t > 0$, there exists an event $\Omega_2(u_0, t)$ such that:
   (i) On $\Omega_2(u_0, t)$, $\mathbb{E}_0 \|y - A\widehat{\alpha}\|_2 \geq \mathbb{E}_\varepsilon \mathbb{E}_0 \|y - A\widehat{\alpha}\|_2 - \sqrt{n}\theta_\varepsilon t$
   (ii) $\mathbb{P}(\Omega_2(u_0, t)) \geq 1 - 2\exp(-c_5 nt^2)$.
By the reasoning in the proof of Proposition C.2.3,

$$\mathbb{E}_\varepsilon \mathbb{E}_0 \|y - A\widehat{\alpha}\|_2 - \sqrt{n}\theta_\varepsilon t \geq \theta_\varepsilon \left[ \mathbb{E}_0 \|I_n - H\|_F - \sqrt{n}t - 1 \right].$$

From Proposition C.2.4 and its proof,

$$\mathbb{E}_0 \|I_n - H\|_F / \sqrt{n} \geq \mathbb{E}_0 (1 + F_n/\tau)^{-1} \geq (1 + \left[ 2c_1^2 L^2 \mathbb{E}_0 \|X\|_F^2 /n^2 + 2\psi_{d,n}(\sigma(0))^2 \right] /\tau)^{-1}.$$

Since $I_n = u_0 u_0^\top + Q_0$, $u_0^\top = u_0^\top + u_0^\top Q_0$, $u_0^\top Q_0 = 0_n^\top$.

$$\begin{aligned}
\mathbb{E}_0 \|X\|_F^2 &= \mathbb{E}_0 \left\| X_0 u_0^\top + XQ_0 \right\|_F^2 \\
&= \mathbb{E}_0 \left[ \left\| X_0 u_0^\top \right\|_F^2 \right] + \|XQ_0\|_F^2 + 2\mathbb{E}_0 \operatorname{trace} \left( X_0 u_0^\top Q_0^\top X^\top \right) \\
&= n + \|XQ_0\|_F^2 + 0 \leq n + \|X\|_F^2 .
\end{aligned}$$

So that we have

$$\mathbb{E}_0 \|I_n - H\|_F / \sqrt{n} \geq (1 + \bar{F}_n/\tau)^{-1}$$

where $\bar{F}_n$ is as given in Proposition C.2.4. Following the rest of the proof of Proposition C.2.3, we will have a desired large event $\Omega(u_0)$ for $u_0 \in \mathbb{S}^{p-1}(1)$.

We notice that, letting $\mathbb{E} := \mathbb{E}_{\boldsymbol{u}_0, \boldsymbol{X}, \boldsymbol{W}, \boldsymbol{\varepsilon}, G}$, we have

$$
\mathbb{E}\left[\epsilon_0^2\right] = \mathbb{E}\left[\frac{\mathbb{E}_0\left[\|\nabla_{\boldsymbol{X}_0}\boldsymbol{r}\|_F^2\right]}{\mathbb{E}_0\left[\|\boldsymbol{r}\|_2^2\right] + \mathbb{E}_0\left[\|\nabla_{\boldsymbol{X}_0}\boldsymbol{r}\|_F^2\right]}\right]
$$

$$
\leq \mathbb{E}\left[\frac{\mathbb{E}_0\left[\|\nabla_{\boldsymbol{X}_0}\boldsymbol{r}\|_F^2\right]}{\mathbb{E}_0\left[\|\boldsymbol{r}\|_2^2\right] + \mathbb{E}_0\left[\|\nabla_{\boldsymbol{X}_0}\boldsymbol{r}\|_F^2\right]} I_{\Omega(\boldsymbol{u}_0)}\right] + \mathbb{P}\left(\Omega(\boldsymbol{u}_0)^c\right)
$$

$$
\leq \mathbb{E}\left[\frac{\mathbb{E}_0\left[\|\nabla_{\boldsymbol{X}_0}\boldsymbol{r}\|_F^2\right]}{\mathbb{E}_0\left[\|\boldsymbol{r}\|_2^2\right] + \mathbb{E}_0\left[\|\nabla_{\boldsymbol{X}_0}\boldsymbol{r}\|_F^2\right]} I_{\Omega(\boldsymbol{u}_0)}\right] + o(\exp(-c_{12}n))
$$

$$
\leq \mathbb{E}\left[\frac{\mathbb{E}_0\left[2\left\|\boldsymbol{T}_0(\boldsymbol{u}_0) + \boldsymbol{T}_{\mathrm{L}}(\boldsymbol{u}_0) + \boldsymbol{T}_{\mathrm{NL}}(\boldsymbol{u}_0)\right\|_F^2\right]}{\mathbb{E}_0\left[\|\boldsymbol{r}\|_2^2\right] + \mathbb{E}_0\left[\|\nabla_{\boldsymbol{X}_0}\boldsymbol{r}\|_F^2\right]} I_{\Omega(\boldsymbol{u}_0)}\right]
$$

$$
+ \mathbb{E}\left[\frac{\mathbb{E}_0\left[2\left\|\boldsymbol{T}_1(\boldsymbol{u}_0)\right\|_F^2\right]}{\mathbb{E}_0\left[\|\boldsymbol{r}\|_2^2\right] + \mathbb{E}_0\left[\|\nabla_{\boldsymbol{X}_0}\boldsymbol{r}\|_F^2\right]} I_{\Omega(\boldsymbol{u}_0)}\right] + o(\exp(-c_{12}n))
$$

$$
\leq (2/n)c_{2,n}^{-1}\mathbb{E}\left[\left\|\boldsymbol{T}_0(\boldsymbol{u}_0) + \boldsymbol{T}_{\mathrm{L}}(\boldsymbol{u}_0) + \boldsymbol{T}_{\mathrm{NL}}(\boldsymbol{u}_0)\right\|_F^2\right]
$$

$$
+ L^2 c_1^2/(2n\tau) + o(\exp(-c_{12}n))
$$

$$
\leq (2/n)c_{2,n}^{-1}(1/p)\left(2c_1^2 L^2 n\mathbb{E}\left[\|\widehat{\boldsymbol{\alpha}}\|_2^2\right] + 2\mathbb{E}\left[\|\boldsymbol{f}'\|_F^2\right]\right)
$$

$$
+ L^2 c_1^2/(2n\tau) + o(\exp(-c_{12}n))
$$

$$
= O(1/p).
$$

The last two inequalities above are due to Step 1. The fact that $\mathbb{E}\left[\|\widehat{\boldsymbol{\alpha}}\|_2^2\right] = O(1)$ is provided in Proposition C.2.2. Assumption 1 and 3 provide us $\mathbb{E}\left[\|\boldsymbol{f}'\|_F^2\right]/n = \mathbb{E}\left[\|\boldsymbol{\beta} + \nabla G(\boldsymbol{x}_1)\|_2^2\right] = O(1)$. □

### D.3.3. Proof of Proposition C.4.2.

**Proposition C.4.2.** Let $\Omega$ be as in Proposition C.2.3. Let $\delta_{i,j} = 1$ if $i = j$, 0 otherwise. Let $\overline{\Omega}_3 := \Omega \cap \left\{\max_{i_1, i_2 \in [n]} |\boldsymbol{x}_{i_1}^T \boldsymbol{x}_{i_2}/p - \delta_{i_1, i_2}| < \delta\right\}$ where $\delta$ is a fixed positive defined in Assumption 4. Let $\boldsymbol{u}_0 \sim \mathrm{Unif}\left(\mathbb{S}^{p-1}(1)\right)$ be independent of $\boldsymbol{X}, \boldsymbol{W}, \boldsymbol{\varepsilon}, G$. Under Assumptions 1, 2, 3 and 4 and a further assumption that $\Sigma_p'(0) = O(1/p)$, we have that

$$
\text{(C.8)} \qquad \mathbb{E}_{\boldsymbol{u}_0, \boldsymbol{X}, \boldsymbol{W}, \boldsymbol{\varepsilon}, G}\left[\left(\frac{\mathrm{trace}\left((\boldsymbol{I}_n - \boldsymbol{H})\mathrm{diag}(\boldsymbol{G}'\boldsymbol{u}_0)\right)}{\|\boldsymbol{y} - \boldsymbol{A}\widehat{\boldsymbol{\alpha}}\|_2}\right)^2 I_{\Omega_3}\right] = O(1/p).
$$

*Proof.* Let us first define
   (i) $\boldsymbol{v}_n = \mathbf{1}_n - (H_{11}, H_{22}, \cdots, H_{nn})^\top$.
   (ii) $q_0 := \mathrm{trace}(\boldsymbol{T}_{\mathrm{NL}}(\boldsymbol{u}_0)) = \mathrm{trace}\left[(\boldsymbol{I}_n - \boldsymbol{H})\mathrm{diag}(\boldsymbol{G}'\boldsymbol{u}_0)\right] = \boldsymbol{v}_n^\top \boldsymbol{G}' \boldsymbol{u}_0$.
   (iii) $\boldsymbol{r} := \boldsymbol{y} - \boldsymbol{A}\widehat{\boldsymbol{\alpha}}$.
   (iv) $\Omega$ be in Proposition C.2.3 such that
       (i) $\Omega := \left\{1/n \cdot \|\boldsymbol{y} - \boldsymbol{A}\widehat{\boldsymbol{\alpha}}\|_2^2 \geq c_{2,n}\right\}$.
       (ii) $\mathbb{P}(\Omega^c) \leq o(\exp(-c_6 n))$ for some constant $c_6 > 0$.
   (v) $\overline{\Omega}_3$ be such that (cf. Corollary 2.8.3 in Vershynin [2018])
       (i)
$$
\overline{\Omega}_3 = \left\{\max_{i_1, i_2 \in [n]} |\boldsymbol{x}_{i_1}^T \boldsymbol{x}_{i_2}/p - \delta_{i_1, i_2}| < \delta\right\}
$$

(ii) $\mathbb{P}(\overline{\Omega}_3^c) \leq o(\exp(-c_{10}n))$ for some universal constant $c_{10} > 0$.

(vi) $\Omega_3 := \Omega \cap \overline{\Omega}_3$.

We abbreviate $\mathbb{E} := \mathbb{E}_{u_0, X, W, G, \varepsilon}$. The order of $\|r\|_2$ is specified on event $\Omega$ so that

$$\mathbb{E}\left[q_0^2 / \|r\|_2^2 \cdot I_{\Omega_3}\right] \leq (1/n)c_{2,n}^{-1}\mathbb{E}\left[q_0^2 I_{\overline{\Omega}_3}\right].$$

By some algebra,

$$\begin{aligned}
\mathbb{E}\left[q_0^2 I_{\overline{\Omega}_3}\right] &= \mathbb{E}\left[\left(\text{trace}\left[(\boldsymbol{I}_n - \boldsymbol{H})\,\text{diag}(\boldsymbol{G}'\boldsymbol{u}_0)\right]\right)^2 I_{\overline{\Omega}_3}\right] \\
&= \mathbb{E}\left[\left(\boldsymbol{v}_n^\top \boldsymbol{G}'\boldsymbol{u}_0\right)^2 I_{\overline{\Omega}_3}\right] \\
&= \mathbb{E}\left[\boldsymbol{v}_n^\top \mathbb{E}_G\left[\boldsymbol{G}'\mathbb{E}_{\boldsymbol{u}_0}[\boldsymbol{u}_0\boldsymbol{u}_0^\top]\boldsymbol{G}'^\top\right]\boldsymbol{v}_n I_{\overline{\Omega}_3}\right] \\
&= (1/p)\mathbb{E}\left[\boldsymbol{v}_n^\top \mathbb{E}_G\left[\boldsymbol{G}'\boldsymbol{G}'^\top\right]\boldsymbol{v}_n I_{\overline{\Omega}_3}\right],
\end{aligned}$$

where we used $\mathbb{E}_{\boldsymbol{u}_0}\left[\boldsymbol{u}_0\boldsymbol{u}_0^\top\right] = (1/p)\boldsymbol{I}_p$. So it suffices to show that

$$\mathbb{E}\left[\boldsymbol{v}_n^\top \mathbb{E}_G\left[\boldsymbol{G}'\boldsymbol{G}'^\top\right]\boldsymbol{v}_n I_{\overline{\Omega}_3}\right] = O(p).$$

Let $\boldsymbol{M} := \mathbb{E}_G\left[\boldsymbol{G}'\boldsymbol{G}'^\top\right]$. From Proposition C.2.7, on $\overline{\Omega}_3$, the $(i_1, i_2)$-th element of the above matrix is

$$m_{i_1,i_2} = \Sigma_p''(\boldsymbol{x}_{i_1}^\top \boldsymbol{x}_{i_2}/p)(\boldsymbol{x}_{i_1}^\top \boldsymbol{x}_{i_2}/p)/p + \Sigma_p'(\boldsymbol{x}_{i_1}^\top \boldsymbol{x}_{i_2}/p).$$

We look at Taylor expansions around $\delta_{i_1,i_2}$, and do some arrangement as following: for $i_1, i_2 \in [n]$ and $\left|\boldsymbol{x}_{i_1}^\top \boldsymbol{x}_{i_2}/p - \delta_{i_1,i_2}\right| \leq \delta$,

$$\begin{aligned}
\Sigma_p'(\boldsymbol{x}_{i_1}^\top \boldsymbol{x}_{i_2}/p) &= \Sigma_p'(\delta_{i_1,i_2}) + \Sigma_p''(\kappa_{i_1,i_2})(\boldsymbol{x}_{i_1}^\top \boldsymbol{x}_{i_2}/p - \delta_{i_1,i_2}), \\
&= \Sigma_p'(\delta_{i_1,i_2}) + \Sigma_p''(0)(\boldsymbol{x}_{i_1}^\top \boldsymbol{x}_{i_2}/p - \delta_{i_1,i_2}) \\
&\quad + \left(\Sigma_p''(\kappa_{i_1,i_2}) - \Sigma_p''(0)\right)(\boldsymbol{x}_{i_1}^\top \boldsymbol{x}_{i_2}/p - \delta_{i_1,i_2})(1 - \delta_{i_1,i_2}) \\
&\quad + \left(\Sigma_p''(\kappa_{i_1,i_2}) - \Sigma_p''(0)\right)(\boldsymbol{x}_{i_1}^\top \boldsymbol{x}_{i_2}/p - \delta_{i_1,i_2})\delta_{i_1,i_2},
\end{aligned}$$

$$\begin{aligned}
\Sigma_p''(\boldsymbol{x}_{i_1}^\top \boldsymbol{x}_{i_2}/p)(\boldsymbol{x}_{i_1}^\top \boldsymbol{x}_{i_2}/p)/p &= \Sigma_p''(0)(\boldsymbol{x}_{i_1}^\top \boldsymbol{x}_{i_2}/p)/p + \left(\Sigma_p''(\boldsymbol{x}_{i_1}^\top \boldsymbol{x}_{i_2}/p) - \Sigma_p''(0)\right)(\boldsymbol{x}_{i_1}^\top \boldsymbol{x}_{i_2}/p)(1 - \delta_{i_1,i_2})/p \\
&\quad + \left(\Sigma_p''(\boldsymbol{x}_{i_1}^\top \boldsymbol{x}_{i_2}/p) - \Sigma_p''(0)\right)(\boldsymbol{x}_{i_1}^\top \boldsymbol{x}_{i_2}/p)\delta_{i_1,i_2}/p,
\end{aligned}$$

where $\kappa_{i_1,i_2}$ satisfies $|\kappa_{i_1,i_2} - \delta_{i_1,i_2}| \leq \left|\boldsymbol{x}_{i_1}^\top \boldsymbol{x}_{i_2}/p - \delta_{i_1,i_2}\right|$. From this we have decomposition of $\boldsymbol{M} := \mathbb{E}_G\left[\boldsymbol{G}'\boldsymbol{G}'^\top\right]$ into several matrices with small operator norm easy to calculate. With a slight abuse of notations $\boldsymbol{A}, \boldsymbol{B}, \boldsymbol{C}, \boldsymbol{D}, \boldsymbol{E}, \boldsymbol{F}, \boldsymbol{G}, \boldsymbol{H}$, we have

$$\boldsymbol{M} = \boldsymbol{A} + \boldsymbol{B} + \boldsymbol{C} + \boldsymbol{D} + \boldsymbol{E} + \boldsymbol{F} + \boldsymbol{G} + \boldsymbol{H},$$

where

$$\begin{aligned}
\boldsymbol{A} &= \left(\Sigma_p'(1) - \Sigma_p'(0)\right)\boldsymbol{I}_n, \\
\boldsymbol{B} &= \Sigma_p'(0)\boldsymbol{1}_n\boldsymbol{1}_n^\top, \\
\boldsymbol{C} &= \Sigma_p''(0)(\boldsymbol{X}\boldsymbol{X}^T/p), \\
\boldsymbol{D}_{i_1,i_2} &= \left(\Sigma_p''(\kappa_{i_1,i_2}) - \Sigma_p''(0)\right)(\boldsymbol{x}_{i_1}^\top \boldsymbol{x}_{i_2}/p)\delta_{i_1 \neq i_2}, \\
\boldsymbol{E}_{i_1,i_2} &= \left[\Sigma_p''(\kappa_{i_1,i_2})\boldsymbol{x}_{i_1}^\top \boldsymbol{x}_{i_2}/p - \Sigma_p''(0)\boldsymbol{x}_{i_1}^\top \boldsymbol{x}_{i_2}/p - \Sigma_p''(\kappa_{i_1,i_2})\right]\delta_{i_1 = i_2}, \\
\boldsymbol{F} &= \Sigma_p''(0)(\boldsymbol{X}\boldsymbol{X}^T/p)/p, \\
\boldsymbol{G}_{i_1,i_2} &= \left(\Sigma_p''(\boldsymbol{x}_{i_1}^\top \boldsymbol{x}_{i_2}/p) - \Sigma_p''(0)\right)(\boldsymbol{x}_{i_1}^\top \boldsymbol{x}_{i_2}/p)\delta_{i_1 \neq i_2}/p, \\
\boldsymbol{H}_{i_1,i_2} &= \left(\Sigma_p''(\boldsymbol{x}_{i_1}^\top \boldsymbol{x}_{i_2}/p) - \Sigma_p''(0)\right)(\boldsymbol{x}_{i_1}^\top \boldsymbol{x}_{i_2}/p)\delta_{i_1 = i_2}/p.
\end{aligned}$$

It suffices to show that $q(\boldsymbol{N}) := \mathbb{E}\left[\boldsymbol{v}_n^\top \boldsymbol{N} \boldsymbol{v}_n I_{\overline{\Omega}_3}\right] = O(p)$ for $\boldsymbol{N}$ being from $\boldsymbol{A}$ to $\boldsymbol{H}$. We notice that $\|\boldsymbol{I}_n - \boldsymbol{H}\|_{\text{op}} \leq 1$ implies $|v_{n,i}| \leq 1$ for all $i \in [n]$. Then we have

(i) $q(\boldsymbol{A}) = O(p)$ provided that $\Sigma_p'(1), \Sigma_p'(0) = O(1)$.

(ii) $q(\boldsymbol{B}) = O(p)$ provided that $\Sigma_p'(0) = O(1/p)$.

(iii) $q(\boldsymbol{C}) = O(p)$ provided that $\mathbb{E}\left[\left\|\boldsymbol{X}/\sqrt{p}\right\|_{\text{op}}\right] = O(1)$ and $\Sigma_p''(0) = O(1)$.

(iv) $q(\boldsymbol{E}) = O(p)$ provided that $\sup_{x \in [1-\delta, 1+\delta]} \Sigma_p''(x), \Sigma_p''(0) = O(1)$ .

(v) $q(\boldsymbol{F}) = O(p)$ provided that $\Sigma_p''(0) = O(1)$.

(vi) $q(\boldsymbol{H}) = O(p)$ provided that $\sup_{x \in [1-\delta, 1+\delta]} \Sigma_p''(x), \Sigma_p''(0) = O(p)$.

We notice that the above are true by assumptions on $\Sigma_p$ and $\boldsymbol{X}$. For $\boldsymbol{D}$ and $\boldsymbol{G}$, we notice the following:

$$|q(\boldsymbol{D})| := \left|\mathbb{E}\left[\boldsymbol{v}_n^\top \boldsymbol{D} \boldsymbol{v}_n I_{\overline{\Omega}_3}\right]\right| \leq \mathbb{E}\left[|\boldsymbol{v}_n|^\top |\boldsymbol{D}| |\boldsymbol{v}_n| I_{\overline{\Omega}_3}\right] \leq \mathbf{1}_n^\top \mathbb{E}[|\boldsymbol{D}| I_{\overline{\Omega}_3}] \mathbf{1}_n,$$

where the absolute value operation is taken element-wise for the vector $\boldsymbol{v}_n$ and matrix $\boldsymbol{D}$. By the Lipschitz assumption of $\Sigma_p''$ around 0, for $i_1 \neq i_2$,

$$\mathbb{E}\left[|d_{i_1,i_2}| I_{\overline{\Omega}_3}\right] \leq \mathbb{E}\left[\left|\left(\Sigma_p''(\kappa_{i_1,i_2}) - \Sigma_p''(0)\right)\right| \left|\boldsymbol{x}_{i_1}^\top \boldsymbol{x}_{i_2}/p\right| I_{\overline{\Omega}_3}\right] \leq L_2 \cdot \mathbb{E}\left[(\boldsymbol{x}_{i_1}^\top \boldsymbol{x}_{i_2}/p)^2\right] = L_2/p.$$

This implies $|q(\boldsymbol{D})| = O(p)$. For $|q(\boldsymbol{G})|$, we notice that

$$|q(\boldsymbol{G})| := \left|\mathbb{E}\left[\boldsymbol{v}_n^\top \boldsymbol{G} \boldsymbol{v}_n I_{\overline{\Omega}_3}\right]\right| \leq \mathbb{E}\left[|\boldsymbol{v}_n|^\top |\boldsymbol{G}| |\boldsymbol{v}_n| I_{\overline{\Omega}_3}\right] \leq \mathbf{1}_n^\top \mathbb{E}\left[|\boldsymbol{G}| I_{\overline{\Omega}_3}\right] \mathbf{1}_n,$$

where the absolute value operation is taken element-wise for the vector $\boldsymbol{v}_n$ and matrix $\boldsymbol{G}$. By the Lipschitz assumption of $\Sigma_p''$ around 0, for $i_1 \neq i_2$,

$$\mathbb{E}\left[|g_{i_1,i_2}| I_{\overline{\Omega}_3}\right] \leq L_2 \mathbb{E}\left[(\boldsymbol{x}_{i_1}^\top \boldsymbol{x}_{i_2}/p)^2\right]/p \leq L_2/p^2.$$

So that $|q(\boldsymbol{G})| = O(1)$. Combining the above we have our proposition. $\qquad\square$