[Reviews · NeurIPS 2020]

Review 1

Summary and Contributions: Constructed confidence interval (for the linear components of the target function) in a two-layer random features model optimized with squared loss and ridge regularization. The result is based on establishing a certain CLT involving the derivative of the model. Theorems are supported by empirical findings.

Strengths: 1. To my knowledge this is the first work on establishing CI for random features model in the proportional asymptotic limit. This result is potentially useful in uncertainty quantification. 2. The paper contains a number of observations that I believe would be interesting to the NeurIPS community: (i) the "double descent" phenomenon in the CI length; (ii) the role of different activation functions; (iii) the benefit of interpolation (weak regularization). 3. The theoretical findings are nicely illustrated by figures and also supported by empirical results.

Weaknesses: 1. The writing felt a bit rushed, with a few typos and confusing notations (see charity below). 2. I'm not entirely sure if I understand the motivation of certain parts of the analysis. In particular, the CI is only established for the linear part of the target (which kind of makes sense since in this asymptotic limit, it is the only thing that RF model can learn); therefore, I do not know the value of including a nonlinear component in addition. See the following sections for additional comments and questions. 3. While I am not an expert in the area, my feeling is that a considerable part of the analysis is known if not rather standard. For instance the setup for CLT resembles Bellec and Zhang 2019, the risk of the RF model is provided in Mei and Montanari 2019, and the utilized linearization in high dimensions is relatively well-known (e.g. previous analysis of the kernel Gram matrix).

Correctness: To my knowledge yes.

Clarity: The structure of the paper is not quite easy to follow; there are times I would need to search back and forth for definitions and explanations. The notation is a bit confusing at times. To list a few, 1) in page 4 and 5, I don't know why the derivative of f(x) is taken w.r.t. $\xi$, which is previously defined as future observation. 2) $\Phi$ in Theorem 1 (which can be inferred as the Gaussian CDF) is not defined beforehand. 3) Caption for Figure 2 (left) is a bit hard to parse.

Relation to Prior Work: (minor) A few related works: On high-dimensional characterization of two-layer nets: Louart, Cosme, Zhenyu Liao, and Romain Couillet. "A random matrix approach to neural networks." The Annals of Applied Probability 28.2 (2018): 1190-1248. Ba, Jimmy, et al. "Generalization of two-layer neural networks: An asymptotic viewpoint." International Conference on Learning Representations. 2019. A work that has similar implications under Lipschitz feature transformation (i.e. weight matrix with bounded spectral norm): Seddik, Mohamed El Amine, et al. "Random matrix theory proves that deep learning representations of gan-data behave as gaussian mixtures." arXiv preprint arXiv:2001.08370 (2020).

Reproducibility: Yes

Additional Feedback: I will increase my score if my concerns are addressed and if the authors could correct my potential misunderstanding. 1. I find the "double descent" phenomenon in the CL length to be interesting. Intuitively, the uncertainty of the model could relate to the variance of the prediction, which we know might blow up at the interpolation threshold due to the variance from label noise or from initialization. Can the author comment on the plausible mechanism of this observation? 2. The asymptotic normality only applies to the linear component of the target function. In this case what would be the motivation of considering a nonlinear perturbation, which would basically be adding noise? 3. The result in Section 2.4 (based on Mei and Montanari 2019) seems to be under the assumption of iid weight matrix W. I might have missed something, but is there a place the authors discussed that this characterization also holds for arbitrary W (independent of X) with bounded spectral norm? 4. (minor) Does the characterization also holds for the ridgeless limit (\lambda=0)? 5. (minor) On Figure 2 Left, why is there a discrepancy between the predicted and simulated boxplot? 6. (minor) Although this is not the motivation of the work, the mentioned connection between NN and RF model typically requires significant overparameterization, and thus the current proportional scaling of n and d might not be the right setup. ----------------------------------- Post-rebuttal: The authors addressed some of my concerns; I have thus increased my score. This being said, I do feel that the usefulness of this setup (i.e. considering a nonlinear model when CLT can only be established for the linear component) requires a bit more elaboration.


Review 2

Summary and Contributions: The paper provides a way to calculate 1-\alpha CI's for the input gradients of a special class of two-layer networks called the Random Features model. === After rebuttal === Thanks for the clarifying rebuttal. Increasing original score by +1.

Strengths: The paper provides an analysis of the confidence intervals of a special class of 2 layer neural networks. This is important from the point of view of calibration of neural networks and other properties of the predictions such as sensitivity to inputs.

Weaknesses: The model of random features is very simplistic and does not provide a clear picture about the behaviour into the empirically observed behaviour of 2 layer neural networks trained using SGD. Neural Networks, when used for classification, are known to suffer from poor calibration. This paper does not touch upon any of these issues prevalent in 2-layer neural networks. It does not address how one would consider extending the present results to understand this phenomenon.

Correctness: I was unable to verify the correctness of the proofs.

Clarity: There is scope for improvement. Section 2 needs better organization of results and concepts. In the the discussion proceeding Theorem 1, it is not clear what the quantity \xi_L has to do with the derivatives of the predictor \hat{f} or its derivative. The quantity nL^2 remains elusive and springs out of nowhere in the discussion. What is its significance. It appears that the training loss appears everywhere. Is this consistent with the population loss or is there a bias introduced due to the regularization. The weighing in (8) seems to be a matter of analytic convenience. What if someone is interested in the unweighted average, what does one do in that case. This seems like a strange constraint from the point of view of using the Asymptotic normality in (8). The paper does not provide any intuition into the proof of the results and all the discussion is relegated to the supplementary section. This is important information that the reader needs to have some idea about to trust the results without having to look at the proofs. I think an overview into the proof techniques needs to be provided into the main text. A comment on the generality of this framework to other similar models such as the NTK (See Mei and Montanari) and models with other regularizers such as \ell_1 in (1) would also be useful.

Relation to Prior Work: Most of the papers discussed are not related to the topic of CI estimation. I think the authors should provide some insight into the problem of CI estimation in the classical setting and how it is done for linear models and what is known.

Reproducibility: Yes

Additional Feedback: It remains unclear why the subscript L is used in quantities defined in (11). Quantity H in equation 7 is not defined


Review 3

Summary and Contributions: In this paper, the authors study a random feature (or two-layer random neural net) model, and show that a weighted average of the derivatives of the trained model output is asymptotically normal in the large n,p,d limit, where the sample size n, input data dimension p and the first layer width d diverge to infinity at the same pace. Technically, the input data are assumed to be i.i.d. standard Gaussian and both linear and nonlinear response regression models are discussed. Built upon this asymptotic normality, confidence intervals are established for components of the (unknown) regression vector. The impact of different dimension ratios p/n and d/n, as well as of the nonlinear activation function, are discussed.

Strengths: The claims are, as far as I can check, technically coorect, and the empirical evaluation on finite-dimensional problems looks compelling. This is in general a nice piece of work studying the confidence intervals of random feature models from the perspective of frequentist uncertainty quantification and is a significant addition to the existing literature.

Weaknesses: The presentation of this paper can be improved. Also, since the assumption on the weight matrix W is rather mild (e.g., to have bounded operator norm), by extending the data beyond the current i.i.d. Gaussian setting, even at the price of imposing i.i.d . Gaussian weights, this paper could have a greater impact.

Correctness: The claims and technical details are, as far as I can check, correct.

Clarity: The presentation of this paper can be improved: there are some typos in the notations (see below) and the connection between (8) and the main Theorem 1 should be clarified (see below).

Relation to Prior Work: The related works are clearly discussed.

Reproducibility: Yes

Additional Feedback: * line 30: the penalty parameter tau is chosen to be tau = (d/p)*lambda, is this scaling (by d/p) technically important? Would most conclusions and insights still hold true if tau is fixed? * Eq (2) and above, notations for transpose should be aligned * Eq (3) a transpose is missing for W in the gradient * line 51 and then in Assumption 1: the independence between feature and additive noise needs clarification. * line 150-152 and Eq (8): the partial derivative is with respect to xi or x? since hat f is evaluated at x_i. This is then related to the presentation of Theorem 1. The connection between Eq (8) and Theorem 1 needs to be clarified, for instance, is the gradient term essentially (the entries) of T_0 in (12)? * Theorem 1: the notation Phi(t) should be introduced explicitly * Figure 3: the leaky ReLU is hardly distinguishable from Swish activation, perhaps using a zoom here or choosing max(x, 0.01x) could help? ===================== After rebuttal: The authors addressed some of my concerns and I have increased my score accordingly. As pointed out by other reviewers, this work can have a larger impact if the model and the setup are better motivated.


Review 4

Summary and Contributions: This paper addresses the important problem of providing confidence intervals for derivatives of a two layer neural net with fixed random weights in the first layer. The paper states a CLT for the derivatives of the trained neural network in equation (8). They go on to introduce a linear model and a linear model with non-linear perturbation function. Under the linear model, thm 1 states that a term which is affine in \beta_j, and all other components can be calculated, has an asymptotic normal distribution. Thm 2 states a similar result but for the more general linear model with non-linear perturbation function. The key point of theoretical interest is that a double descent phenomenon occurs for the confidence interval width as the number of hidden layers increases relative to input dimension and sample size. ------------------------ I have read the review and am satisfied by their explanation of how (8) follows from the results. This paper is a solid addition to the work on theory for neural nets, and will raise my assessment accordingly.

Strengths: This is the first result that I know of that presents a double descent phenomenon for CI widths. Typically, the double descent is on the true risk which is another functional of the trained NNet. This result implies that the variation on the gradients diminishes as width increases, making this functional more stable. It is certainly enlightening, and it indicates that we can expect this behavior from CIs of other functionals.

Weaknesses: It is unclear of what the practical usefulness of this result might be. CLTs are very useful because you can both use them to establish levels for hypothesis tests as well as CIs for population level quantities. The main issue is that this CLT holds under a linear model (or with non-linear perturbation). The coefficients within a linear model are not a natural quantity to be concerned with for the neural network. It seems that if we believed that a linear model held, we would fit a linear model, not a neural net. I’m not sure how these theorems might generalize to models that are more in-line with the neural net architecture itself, such as the mean of the response following a neural net of matching architecture. Perhaps the importance of this case has been suppressed for space considerations.

Correctness: What I read in detail of the proofs look correct, and they are highly non-trivial applications of random matrix theory, the “de-biasing” technique, and tools used in recent related works. However, I cannot find anything that actually substantiates the claim in equation (8). While it certainly seems that the gradient in question lies within the pivotal quantity in theorems 1 and 2, it is confusing to see that the results address the linear models and their betas. Perhaps the authors can clear up this confusion, by answering how (8) exactly follows.

Clarity: The paper is well written, with an excellent related work section. As I discuss there is insufficient motivation for the results, given the assumptions. There are some grammatical mistakes, so the authors should have someone read it over to identify these typos.

Relation to Prior Work: It is, the relation between this work and prior works that display double descent phenomena is clearly outlined.

Reproducibility: Yes

Additional Feedback:

[Author Response · NeurIPS 2020]

We thank all reviewers for their time and understanding of the results, and we are glad that all reviewers appreciate the
introduction of the first CIs that feature a double-descent behavior. We first address 2 points raised by several referees
and then address comments by individual reviewers. All typos/writing issues will be fixed as suggested in the reviews.
We believe the rebuttal addresses all actionable concerns, and we invite the referees to revisit their scores in light of this.

• **Why does (8) hold?** By (3), the weighted average in (8) is equal to $-\mathrm{trace}[\boldsymbol{T}_0(\boldsymbol{e}_j)]/\|\boldsymbol{y} - \boldsymbol{A}\widehat{\boldsymbol{\alpha}}\|_2$ in Theorem 1. The
correctness of (8) then follows by Theorem 1. The term [additive correction] in (8) includes traces of $\boldsymbol{T}_1, \boldsymbol{T}_L$ and the
first term in (11). We'll clarify in the camera-ready version. **This addresses a major concern from Reviewers 2, 3, 4.**
• **Why vector $\boldsymbol{\xi}$ is the argument of $\hat{f}$ in (3)?** The variable name $\boldsymbol{\xi}$ in (3) was picked to avoid confusion with the
observed feature vectors $(\boldsymbol{x}_i)_{i\in[n]}$, since the function $\hat{f}$ itself in (3) depends implicitly on $(\boldsymbol{x}_i)_{i\in[n]}$ through $\widehat{\boldsymbol{\alpha}}$. It is
thus important to avoid $\boldsymbol{x}_i$ as argument for the definition of the function $\hat{f} : \mathbb{R}^p \to \mathbb{R}$ in (3); for clarity we avoided $\boldsymbol{x}$ as
well. Applied to $\boldsymbol{x}_i$, quantity $\hat{f}(\boldsymbol{x}_i)$ depends on $\boldsymbol{x}_i$ directly through the input given to $\hat{f}$ and implicitly through $\widehat{\boldsymbol{\alpha}}$ in the
definition of $\hat{f}$. The implicit dependence of $\hat{f}$ on $\boldsymbol{x}_i$ through $\widehat{\boldsymbol{\alpha}}$ is the reason why classical CLT would not apply to
$n^{-1/2}\sum_i \hat{f}(\boldsymbol{x}_i)$. Since $\hat{f}$ is the prediction function after training, $\hat{f}(\boldsymbol{\xi})$ can be thought of as the prediction at a point of
interest such as an unlabeled future observation, though this is only a thought experiment and $\boldsymbol{\xi}$ is only used as the
argument of $\hat{f}$ to set the notation for partial derivatives. The quantity $\xi_L$ in Theorem 1 is unrelated to the vector $\boldsymbol{\xi}$ in (3);
to avoid confusion we will clarify and use $\zeta_L$ instead. **This addresses a major concern from Reviewers 1, 2 and 3.**

**Reviewer 1:** *"the linear part of the target is the only thing that RF model can learn"* thanks–we'll add this comment
with a reference as a motivation for the $\beta_j$s being the targets of the CIs. *" (...) the CI is only established for the linear*
*part of the target (...) In this case what would be the motivation of considering a nonlinear perturbation "* $\Rightarrow$ The
nonlinear perturbation adds noise correlated with the features, which is possibly more hostile than the independent
additive noise $\boldsymbol{\varepsilon}$. This model of nonlinear perturbation wasn't introduced in the submission but in [Mei and Montanari
2019] to study the double-descent curves for the risk. It is not an ad hoc model to obtain CIs. *"I find the "double*
*descent" phenomenon in the CI length to be interesting.(...) Can the author comment on the plausible mechanism of this*
*observation?"* $\Rightarrow$ A possible mechanism is the conjecture that the length of the CIs is an increasing function of the risk.
Proving this appears highly non-trivial. Such phenomena were observed for M-estimators in [Celentano and Montanari
2019, Prop. 4.3(iii)]. *"The result in Section 2.4 (based on Mei and Montanari 2019) seems to be under the assumption*
*of iid weight matrix W"* $\Rightarrow$ Results in § 2.4 build upon results of Mei and Montanari. Hence, for that section only we
require $\boldsymbol{W}$ with iid entries, but, e.g., Theorems 1 and 2 hold for bounded $\|\boldsymbol{W}\|_{op}$ with no iid assumption. *"(minor)*
*A few related works(...)"* $\Rightarrow$ Thanks–we'll add those. *"(minor) Does the characterization also hold for the ridgeless*
*limit $(\lambda = 0)$?"* $\Rightarrow$ Not currently. A lead to study $\lambda = 0$ is to first take $n, p \to +\infty$ (our result), then take $\lambda \to 0$ and
study whether interchangeability of limits applies. This appears non-trivial. *"(minor) On Figure 2 Left, why is there a*
*discrepancy between the predicted and simulated boxplot?"* $\Rightarrow$ The predicted theory is the thick blue line only. The
discrepancy is mild and expected for some boxplots when plotting that many boxplots. We'll increase number of runs.
**Reviewer 2:** *"Neural Networks, when used for classification, are known to suffer from poor calibration. This paper*
*does not touch upon any of these issues prevalent in 2-layer neural networks."* $\Rightarrow$ We understand the referee's concerns
but it does not appear reasonable to solve, in a single 8-pages submission, both CIs for regression, classification and
solve the mentioned calibration issues. *"The quantity $nL^2$ remains elusive (...)"* $\Rightarrow$ In our results $L \asymp n^{-1/2}$, as for the
length of CIs in classical statistics. $nL^2$ is of constant order after multiplication by $n$. That's why we study this quantity
in figures. *" weighing in (8) seems to be a matter of analytic convenience (...) "* $\Rightarrow$ This is a contribution of the paper to
pinpoint this specific surprising weighting that leads to a pivotal quantity in Theorem 1. This allows the construction of
CIs. *" It appears that the training loss appears everywhere. Is this consistent with (...)"* $\Rightarrow$ Training loss, $\mathrm{trace}[\boldsymbol{H}]$ and
population loss are linked through subtle nonlinear relationships studied [Mei and Montanari 2019] and our § 2.4. *"(...)*
*overview into the proof techniques needs to be provided"* $\Rightarrow$ A proof outline will be added (using the 9th page allowed).
**Reviewer 3:** *" (...)$\tau$ is chosen to be $\tau = (d/p)\lambda$, is this scaling (by $d/p$) technically important?(...) "* $\Rightarrow$ No. Since $d/p$
has a finite limit, parametrization of the tuning parameter can be performed through either $\tau$ or $\lambda$ without changing the
conclusions. *"Eq (2) and above, notations for transpose should be aligned (...) needs clarification. (...) "* $\Rightarrow$ Thanks!
We'll clarify the writing and fix the typos as suggested. *" (...) The connection between Eq 8 and Theorem 1 needs to*
*be clarified (...) "* $\Rightarrow$ cf. line 5-7 above. *" Figure 3:(...)or choosing max(x, 0.01x) could help? "* $\Rightarrow$ Thx! We'll use it.
**Reviewer 4:** *"The main issue is that this CLT holds under a linear model(...) The coefficients within a linear model are*
*not a natural quantity to be concerned with for the neural network(...) if we believed that a linear model held, we would*
*fit a linear model, not a neural net"* $\Rightarrow$ Thanks. Developing similar CIs where $\mathbb{E}[y|\boldsymbol{x}]$ is a nonlinear function of $\boldsymbol{x}$ is an
interesting future direction. One reason for starting with a linear model is that the targets for the CIs are canonically
defined (the unknown coefficients $\beta_j$), while for nonlinear models, it is unclear what to consider as canonical population
targets (functionals of $\mathbb{E}[y|\boldsymbol{x}]$) for the CIs. Studying linear models has been fruitful to understand double descent curves
in numerous recent works (cf. related work section or e.g., [Mei and Montanari]), and this submission is the first work
to provide CIs within this line of research. *" (...)it is confusing to see that the results address the linear models and*
*their betas.(...) Perhaps the authors can clear up this confusion, by answering how (8) exactly follows. "* $\Rightarrow$ cf. line 5-7
above. The relationship between Theorem 1 and the linear coefficients $\beta_j$s are addressed on line 187-188 around (14).

[Meta-Review · NeurIPS 2020]

The reviewers point out that this is a borderline submission. They reasonably questions several things in the paper: - it is not clear why the coefficients for which the CLT holds for are important; - assumptions are restrictive; - the paper studies too simplistic of a model; - parts of the analysis are unclear; - writing is done hastily with typos lingering around. After my own reading, I agree with these comments. On the other hand, the reviewers also point out that there are certain aspects of double descent that are not previously explored, which are of more interest compared to the confidence intervals. My opinion is that the paper would be much stronger if the cons were addressed in a revised manuscript.